# Colocalized, bidirectional optogenetic modulations in freely behaving mice with a wireless dual-color optoelectronic probe

Lizhu Li[1,8], Lihui Lu[2,3,8], Yuqi Ren[2,3,8], Guo Tang[1,8], Yu Zhao[1], Xue Cai[1], Zhao Shi[1], He Ding [4], Changbo Liu[5], Dali Cheng[1], Yang Xie[1], Huachun Wang[1], Xin Fu[6], Lan Yin [6], Minmin Luo [2,3,7✉] & Xing Sheng [1,7✉]

Optogenetic methods provide efficient cell-specific modulations, and the ability of simultaneous neural activation and inhibition in the same brain region of freely moving animals is highly desirable. Here we report bidirectional neuronal activity manipulation accomplished by a wireless, dual-color optogenetic probe in synergy with the co-expression of two spectrally distinct opsins (ChrimsonR and stGtACR2) in a rodent model. The flexible probe comprises vertically assembled, thin-film microscale light-emitting diodes with a lateral dimension of $125 \times 180\ \mu m^2$, showing colocalized red and blue emissions and enabling chronic in vivo operations with desirable biocompatibilities. Red or blue irradiations deterministically evoke or silence neurons co-expressing the two opsins. The probe interferes with dopaminergic neurons in the ventral tegmental area of mice, increasing or decreasing dopamine levels. Such bidirectional regulations further generate rewarding and aversive behaviors and interrogate social interactions among multiple mice. These technologies create numerous opportunities and implications for brain research.

[1] Department of Electronic Engineering, Beijing National Research Center for Information Science and Technology, Center for Flexible Electronics Technology, Tsinghua University, Beijing, China. [2] Chinese Institute for Brain Research, Beijing, China. [3] National Institute of Biological Sciences and School of Life Sciences, Tsinghua University, Beijing, China. [4] School of Optics and Photonics, Beijing Institute of Technology, Beijing, China. [5] School of Materials Science and Engineering, Beihang University, Beijing, China. [6] School of Materials Science and Engineering, Tsinghua University, Beijing, China. [7] IDG/McGovern Institute for Brain Research, Tsinghua University, Beijing, China. [8] These authors contributed equally: Lizhu Li, Lihui Lu, Yuqi Ren, Guo Tang. ✉email: luominmin@cibr.ac.cn; xingsheng@tsinghua.edu.cn

Understanding brain functions and treating neurological disorders rely on the continuous development of advanced technologies to interrogate complex nervous systems[1,2]. Over the past decades, optogenetic methods have been emerging as a powerful toolset for effective and precise neural manipulation, owing to their capability of specific cell targeting[3,4]. Until now, a range of genetically encoded, light-sensitive proteins (opsins) with distinct spectral responses have been engineered to regulate flows of different cations and anions[5–15], triggering various biological responses, from cells and *C. elegans* to rodents and non-human primates[16–19]. Accompanying the development of opsins, advanced technologies for light delivery into the brain have also been progressing rapidly. Apart from various waveguide-based emitters interfacing with external light sources[20–22], recently developed implantable probes based on thin-film, microscale optoelectronic devices have offered a viable solution to versatile neural modulations in untethered animals, when incorporated with various wirelessly operating systems based on radio frequency (RF) antennas[23–25], near-field communication (NFC)[26], Bluetooth chips[27], and infrared receivers[28]. In addition to these achievements, bidirectional neural modulations, specifically, by activating or suppressing the same or different populations of cells in behaving animals with the light of different wavelengths, are highly demanded. Such capabilities would enable more precise control of neural activities to advance both neuroscience explorations and disease medications[29]. Certain efforts have been attempted, including explorations based on experiments by expressing excitatory and inhibitory opsins in separate animals[30,31] or distinct cells in different neural regions[17,32]. There are also reports on dual-color optogenetic activation and inhibition by co-expressing different opsins in the same cells[33–36]. Recent efforts also remarkably demonstrate bidirectional optogenetic modulations with implantable waveguides coupled to extracranial dual-color laser sources and their combination with in vivo electrophysiological recordings[23,36,37], but the wireless operation has not been achieved. Therefore, we envision that wirelessly operated light sources with polychromatic emissions, which collaborate with co-expressed opsins, can be implemented for untethered use in freely moving animals and enable previously inaccessible applications.

This study presents synergic optoelectronic and biological strategies to overcome challenges of existing techniques, and realize colocalized, bidirectional neural manipulations in untethered behaving mice. With heterogeneously integrated thin-film, microscale light-emitting diodes (micro-LEDs), a wirelessly operated, implantable dual-color probe allows independently controlled red and blue emissions in the same region. We also discover that the co-expression of two channelrhodopsins (ChrimsonR and stGtACR2) in the same group of neurons enables efficient light-evoked cell depolarization and hyperpolarization with controlled cation and anion flows under red and blue illuminations, respectively. With the wireless, dual-color micro-LED probes implanted into the ventral tegmental area (VTA) of freely moving mice, preference and aversion tests of individual and social behaviors clearly demonstrate the bidirectional optogenetic excitation and inhibition capability. Furthermore, in vivo behavioral results associated with the red and blue stimulations in the VTA are validated by increased or decreased dopamine levels detected in the nucleus accumbens (NAc) in real time.

## Results

### Wirelessly operated, dual-color micro-LED probes and circuit systems.
We design and implement a wireless, dual-color micro-LED probe that generates red and blue emissions in the same brain region for bidirectional optogenetic modulations in vivo, as schematically illustrated in Fig. 1a. Our previous works establish wirelessly operated microscale optoelectronic device strategy for in vivo, single-color optogenetic stimulations[37–40], here these concepts are applied for dual-color, bidirectional modulations. Specifically, with the co-expression of spectrally distinct ion channels, ChrimsonR[7] and stGtACR2[9], in the same population of cells, red and blue light regulates the inflow of cations and anions, causing cell depolarization and hyperpolarization, respectively (Fig. 1b). Figure 1c displays an enlarged view of the dual-color probe, with design and fabrication details provided in "Methods" and Supplementary Fig. S1. From bottom to top, the probe structure comprises of a copper (Cu)-coated polyimide (PI) thin-film substrate, an indium gallium phosphide (InGaP) red LED, a silicon oxide ($SiO_2$)/titanium oxide ($TiO_2$)-based dielectric filter, and an indium gallium nitride (InGaN) blue LED. The Cu coating on PI serves as an efficient heat spreader to reduce the probe's operation temperature[37]. All the devices (red LEDs, blue LEDs, and filters) are formed in thin-film, microscale formats (lateral dimension ~$125 \times 180$ $\mu m^2$, thickness ~5–7 $\mu m$) through epitaxial lift-off, and vertically assembled via transfer printing[37,39,40] (Fig. 1d). The dimensions of our LEDs are designed to target a large nucleus like VTA. Although smaller devices down to 10–20 $\mu m$ can be fabricated[41,42], reducing LED size decreases luminescence efficiencies[43,44] and may cause additional tissue heating. The intrinsic optical transparency of active layers in the InGaN blue micro-LED allows emissions from the red micro-LED to pass through with minimal losses. In addition, the designed filter selectively reflects blue light and transmits red light, enhancing the emissive efficiencies of both micro-LEDs.

Flexible PI substrates with vertically stacked micro-LEDs are laser milled and encapsulated with a waterproof coating, forming a needle-shaped probe with a dimension (thickness ~120 $\mu m$ and width ~320 $\mu m$) similar to a conventional silica fiber (diameter ~220 $\mu m$) used in optogenetic stimulations. The Cu coated PI substrate has a measured Young's modulus of ~15 GPa, softer than silicon (~180 GPa) and tungsten (~400 GPa), but still much harder than the brain tissue (~1 kPa). The formed probe has a bending stiffness of ~$9 \times 10^4$ pN·$m^2$, similar to metal electrodes used for electrophysiological studies[45]. Separate metallization and electrical insulation ensure that red and blue micro-LEDs can be independently lighted up and display spectrally varied illuminations (red, blue, or combined) in the same location (Fig. 1e and Supplementary Movie S1). Figure 1f plots measured electroluminescence (EL) spectra. The red and blue micro-LEDs exhibit emission peaks at 630 nm and 480 nm, which are in good accordance with the excitation spectra of ChrimsonR and stGtACR2 (data extracted from refs. [7,8]), respectively. Moreover, the spectral separation between the two opsins, guarantee precise optogenetic activation and inhibition with minimal optical interferences. More optoelectronic properties of the dual-color micro-LED probe, including current–voltage characteristics, quantum efficiencies, power irradiance, and angular-dependent emissive profiles are presented in Supplementary Fig. S2. Under injection currents of 10 mA, the red and the blue micro-LEDs can reach irradiances up to 50 mW/$mm^2$ and 200 mW/$mm^2$ on the device surface, which are sufficient for optogenetic modulations. In addition, both devices exhibit a uniform, near Lambertian emission profile.

To illustrate the optical performance of the devices, we insert a dual-color probe into a brain phantom (Fig. 1g, top). For comparison, established models calculate optical profiles based on the optical properties (absorption and scattering) of brain tissues (Fig. 1g, bottom). Both experimental and simulation results reveal that vertically stacked dual-color micro-LEDs show colocalized red and blue emission profiles, ensuring effective activation and

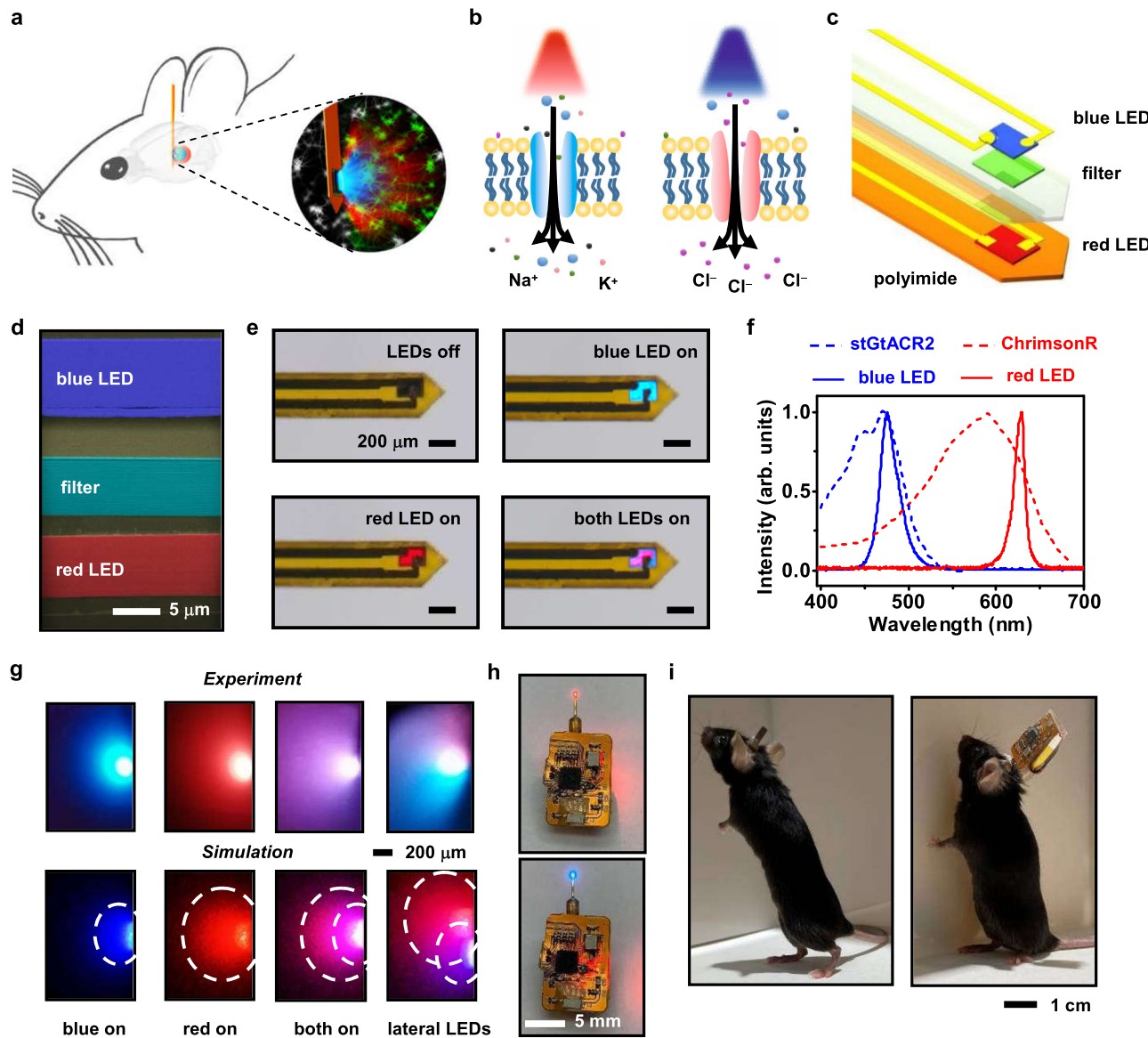

**Fig. 1 Wirelessly operated, dual-color microscale LED (micro-LED) probes for bidirectional, colocalized optogenetic control of neural activities.**
**a** Schematic view of the probe inserted into the target brain area to perform dual-color neural activation and inhibition. **b** Schematic, cellular scale depiction of the modulation function: red light regulates the cation channel ChrimsonR for depolarization, while blue light regulates the anion channel stGtACR2 for hyperpolarization. **c** Exploded view of the dual-color LED probe made from vertically stacked blue and red thin-film micro-LEDs assembled on a flexible polyimide (PI) substrate, with a thin-film filter interface for spectrally selective reflection and transmission. **d** Cross-sectional scanning electron microscopic (SEM) image of the probe structure, with false colors highlighting (from bottom to top) the indium gallium arsenide (InGaP) red micro-LED, the multilayer silicon dioxide/titanium dioxide ($SiO_2/TiO_2$) filter, and the indium gallium nitride (InGaN) blue micro-LED. Layers of SU-8 photoresists serve as bonding materials. **e** Optical images of a micro-LED probe (with a thickness of ~120 μm and a width of ~320 μm), showing independent control of blue and red emissions under current injections. **f** Emission spectra of the blue and red micro-LEDs, in comparison with the action spectra of ion channels stGtACR2 and ChrimsonR. **g** Experimental photographs and simulated results illustrating blue and red light propagations for a micro-LED probe embedded into a brain phantom (columns 1–3). Results for a probe with laterally assemble blue and red LEDs are also presented (column 4), showing misplaced blue and red emissions. Iso-intensity lines show 10% of the maximum power. **h** Images of a micro-LED probe integrated with a wireless circuit for independent control of dual-color emissions. **i** Photos of a behaving mouse after intracranial implantation of a probe, with (right) and without (left) the wireless circuit module. Source data are provided as a Source Data file.

inhibition in the same brain region. Power irradiances for blue and red emissions decrease to 10% of the original power at propagating distances of ~400 μm and ~800 μm, respectively. Compared to a probe with laterally arranged red and blue micro-LEDs displaying misaligned emissions with a relatively small overlapping volume of light, the vertically assembled device provides a smaller footprint and generates an optimal overlapping profile for dual-color stimulations.

Thermal behaviors of the micro-LED probe are also systematically analyzed, with detailed results presented in Supplementary Figs. S3 and S4. Probes are inserted into the brain phantom as well as the living mouse brain, with temperature rises measured by an infrared camera and a thermocouple. Both experiments obtain similar results and are in agreement with the thermal models established based on finite-element analysis. Under a pulsed operation (20 Hz frequency, 10-ms pulse duration, current

0–10 mA), the tissue temperature rises caused by the red micro-LED can be restricted within 1 °C. The operation of the blue micro-LED induces a higher thermal effect because practical neural inhibition for stGtACR2 requires continuous current injection; nonetheless, the tissue heating can be controlled within 2 °C by limiting the current within 5 mA. Such operation modes for red and blue micro-LEDs will be employed throughout in vivo experiments, to mitigate undesired neural responses and possible tissue damage due to overheating.

We also design a customized, miniaturized flexible circuit module to wirelessly operate the dual-color micro-LED probe. Compared to previous reports[22,36] on tethered fiber or laser coupled devices for dual-color stimulations, the wireless operation here enables the study of complex behaviors of freely moving animals more conveniently. Controlled by radio frequency (RF) communication at 2.4 GHz, the battery-powered circuit (weight ~1.9 g) independently addresses the red and the blue micro-LEDs, by adjusting their pulse frequencies, durations, and injection currents for versatile neural modulations in vivo (Fig. 1h, and details in "Methods" and Supplementary Figs. S5–S9). Designed programming commands also allow the independent operation of multiple micro-LED probes in synchronized or non-synchronized modes (Supplementary Movie S2). After subcranial implanting the micro-LED probe into a behaving mouse, the light-weighted circuit mounted onto the mouse head provides the capability for optogenetic control in an untethered manner (Fig. 1i).

**Bidirectional optogenetic manipulations of neurons in vitro, with dual-color stimulations.** We first evaluate the feasibility of dual-color, bidirectional optogenetic modulations for neurons co-expressing ChrimsonR and stGtACR2 in brain slices with electrophysiological tests. Mixed vectors of AAV-CAG-DIO-ChrimsonR-mCherry and AAV-EF1a-DIO-stGtACR2-EGFP are injected into the ventral tegmental area (VTA) of *DAT-Cre* mice (Fig. 2a). The deployment of the soma-targeted variant stGtACR2 exhibits performance superior to previously reported anion channel GtACR2, by enhancing the membrane trafficking and reducing the axonal excitation[8,9]. Figure 2b shows the fluorescence images of stained cells 3 weeks after viral injection in the VTA, including 4',6-diamidino-2-phenylindole (DAPI, blue), stGtACR2 (green), and ChrimsonR (red). The merged graph and zoomed-in views (Fig. 2c) clearly reveal the desirable co-expression of the two opsins in target cells. Statistical results show that ~30% of cells are labeled with stGtACR2 or ChrimsonR, and ~83% of these labeled cells co-express both opsins on average (Fig. 2d).

Whole-cell patch recordings capture intracellular signals of these cells under optical stimulations (Fig. 2e). Other than using our designed micro-LED probe, here we employ a collimated light spot with coupled red (633 nm) and blue (480 nm) laser sources for in vitro experiments, to achieve uniform illuminations for better power calibrations. Figure 2f, g validates that pulsed red stimulations (20-Hz frequency, 10-ms pulse duration) induce action potentials (APs or spikes) by depolarizing these cells, and spike activation probability increases at elevated red irradiances. In these same cells, continuous blue stimulations suppress activities by hyperpolarization (Fig. 2h–j). Voltage-clamp results illustrate that blue light evokes photocurrent reversion when the membrane potential becomes more positive (Fig. 2h), and the probability for spike inhibition also increases at elevated blue irradiances (Fig. 2i, j). Threshold irradiances for complete spike activation and inhibition are around 10 mW/mm$^2$ and 1 mW/mm$^2$ for red and blue illuminations, respectively. The results are consistent with previous reports on cells expressing only ChrimsonR or stGtACR2[7,9].

Notably, we do not only demonstrate that the co-expression of both opsins does not affect the efficacy of cell activity excitation or suppression, but also accomplish bidirectional activation and inhibition in the same cell by instantaneously altered red and blue irradiations (Fig. 2k). In accordance with the activation spectra of two opsins (Fig. 1f), red light does not interact with stGtACR2, which exhibits nearly zero responses at wavelengths longer than 600 nm. On the other hand, ChrimsonR should have responded to blue light, which could raise a concern about possible optical crosstalk. Nevertheless, spike activations caused by blue light do not occur in cells co-expressing ChrimsonR and stGtACR2, since cells are suppressed by turning on stGtACR2 channels at a much lower blue irradiance. This fact can be confirmed by patching the cells under combined red and blue illuminations (Fig. 2l, m). Example traces in Fig. 2l show that red light-evoked spikes are suppressed under simultaneous blue illuminations even with very low irradiances. More traces are reported in Supplementary Fig. S10, and Fig. 2m summarizes the spike probability with combined red and blue illuminations. Introducing blue irradiances orders of magnitude lower than red irradiances can effectively suppress cell spikes and cause activity inhibition.

**Bidirectional optogenetic modulation and simultaneous electrophysiological recording in vivo.** Neural activities associated with dual-color stimulations are further examined with simultaneous electrophysiological recordings in vivo (Supplementary Fig. S11). Two opsins (ChrimsonR and stGtACR2) are co-expressed in the primary somatosensory cortex of *CaMKII-Cre* mice (Supplementary Fig. S11a). Followed by 2 weeks of recovery and viral expression, extracellular electrodes, and dual-color micro-LED probes are inserted in the head-fixed mice, with a separation of ~500 μm (Supplementary Fig. S11b). Spontaneous and red light stimulated waveforms are recorded and sorted, showing typical extracellular spike events (Supplementary Fig. S11c). Multiple trials collected for a sample unit indicate that cell spike rates can be enhanced or decreased by applying red or blue illuminations (Supplementary Fig. S11d, e). Results summarized for multiple cells show that the basal spike firing rate (13 Hz) increases to 28 Hz during red illumination and decreases to 2 Hz during blue illumination (Supplementary Fig. S11f, g). These results showcase the capability for in vivo bidirectional modulations and are in good accordance with the previous report based on a different opsin combination (ChR2 and Jaws)[36].

**Bidirectional optogenetic modulation of preference and aversion behaviors in vivo, with wireless dual-color micro-LED probes.** Activities of the dopaminergic neurons in the VTA we manage to manipulate in Fig. 2 significantly correlate with the reward and aversion-based behaviors[46–48]. In Fig. 3, we further exploit the capability of bidirectional optogenetic manipulations by implanting dual-color micro-LED probes into behaving mice. After the injection of AAV-CAG-DIO-ChrimsonR-mCherry and AAV-EF1a-DIO-stGtACR2-EGFP vectors, the dual-color micro-LED probe is implanted into the VTA of *DAT-cre* mice via standard stereotaxic surgery (Fig. 3a and Supplementary Fig. S12). Figure 3b presents fluorescence images of a brain slice with a micro-LED probe underneath it, showing the co-expression of ChrimsonR and stGtACR2 in the VTA, combined with the luminescence of the red and the blue micro-LEDs under corresponding optical excitations. To assess the devices' biocompatibility in vivo, histological examinations of brain slices show that inflammatory reactions of the micro-LED probe are similar to a conventional silica fiber (diameter ~220 μm) 1 day or 3 weeks' post-implantation (Supplementary Fig. S13), owing to its thin-film geometry, mechanical flexibility, and bio-friendly

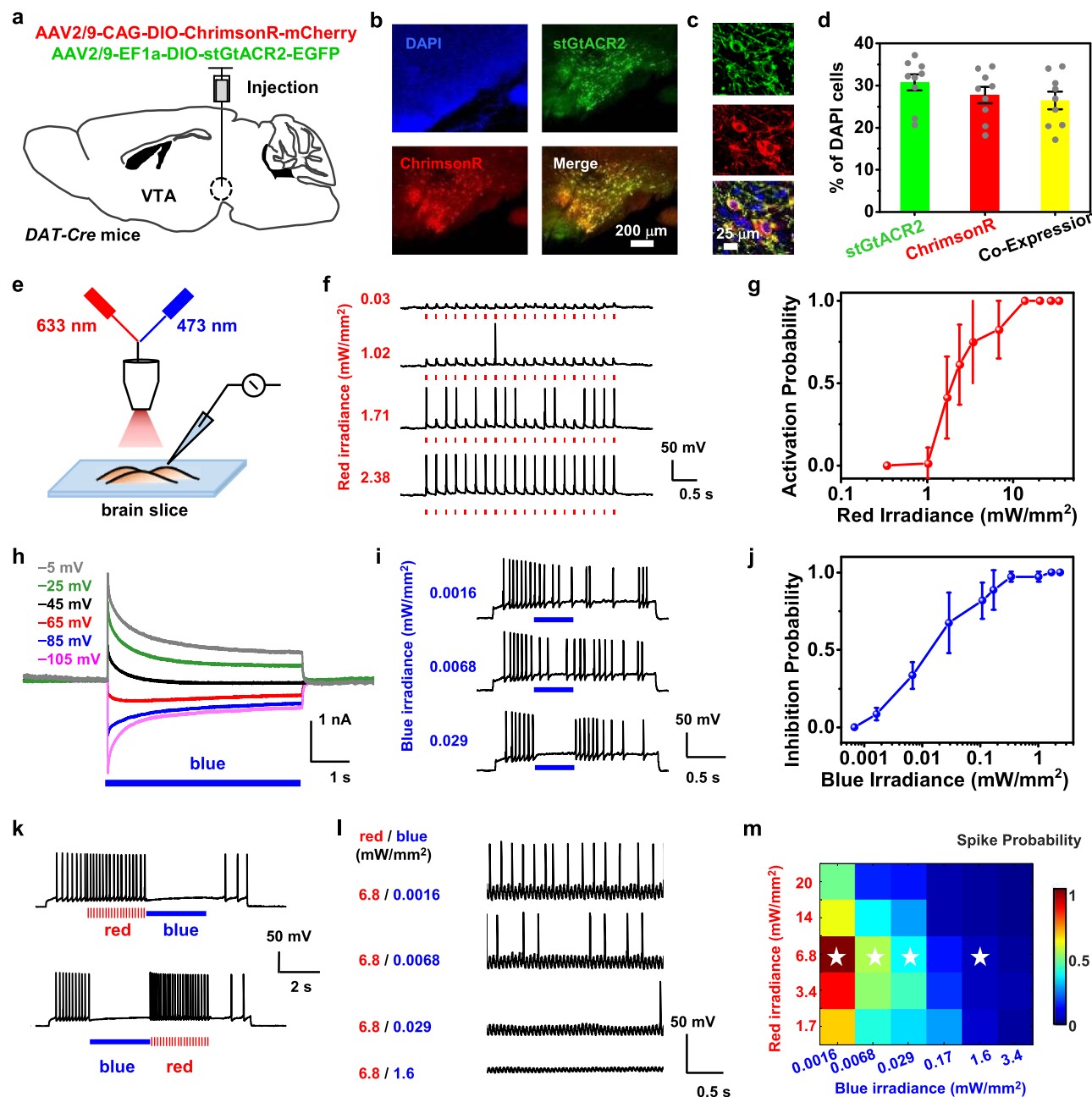

**Fig. 2 Bidirectional optogenetic activation and inhibition of neurons with red and blue light stimulation. a** Schematic strategy for co-expressing ChrimsonR and stGtACR2 in the ventral tegmental area (VTA) of *DAT-cre* mice. **b** Immunostained fluorescence images of the VTA region expressing DAPI (blue), stGtACR2 (green), and ChrimsonR (red) and a merged image, 3 weeks after viral injection. **c** Zoomed view showing cells co-expressing stGtACR2 (green) and ChrimsonR (red). **d** Percentages of stGtACR2-, ChrimsonR- and co-expressing neurons among DAPI cells in the VTA (mean ± s.e.m., 3 slices per mouse, $n = 3$ mice). **e** Illustration of the experiment setup with combined blue (480 nm) and red (633 nm) laser beams (spot diameter ~1 mm) for optogenetic stimulating brain slices during patch-clamp recordings in vitro. **f** Example traces showing neural activation by pulsed red light at different irradiances (20 Hz, 10-ms pulse, 20-pulse train). **g** Activated spike probability as a function of red irradiances (mean ± s.e.m., $n = 4$ cells). **h** Example photocurrents of neurons at membrane potentials varied from −105 to −5 mV (bottom to top), under continuous blue light illumination (duration 5 s, irradiance 8 mW/mm²). **i** Example traces of neural inhibition by continuous blue light at different irradiances (duration 0.5 s). **j** Inhibited spike probability as a function of blue irradiances (mean ± s.e.m., $n = 5$ cells). **k** Traces presenting bidirectional spike activation and inhibition with alternating red and blue illuminations (Patching current +90 pA; Red: duration 3 s, 20 Hz, 10-ms pulse, 2.2 mW/mm²; Blue: duration 3 s, 8.5 mW/mm²). **l** Example traces of neural activities with simultaneous red and blue illuminations at different irradiances (duration 6 s). **m** Summarized plot of measured spike probability (normalized mean values) as a function of red and blue irradiances ($n = 5$ cells). The stars indicate the position of the four example traces shown in (**l**). Source data are provided as a Source Data file.

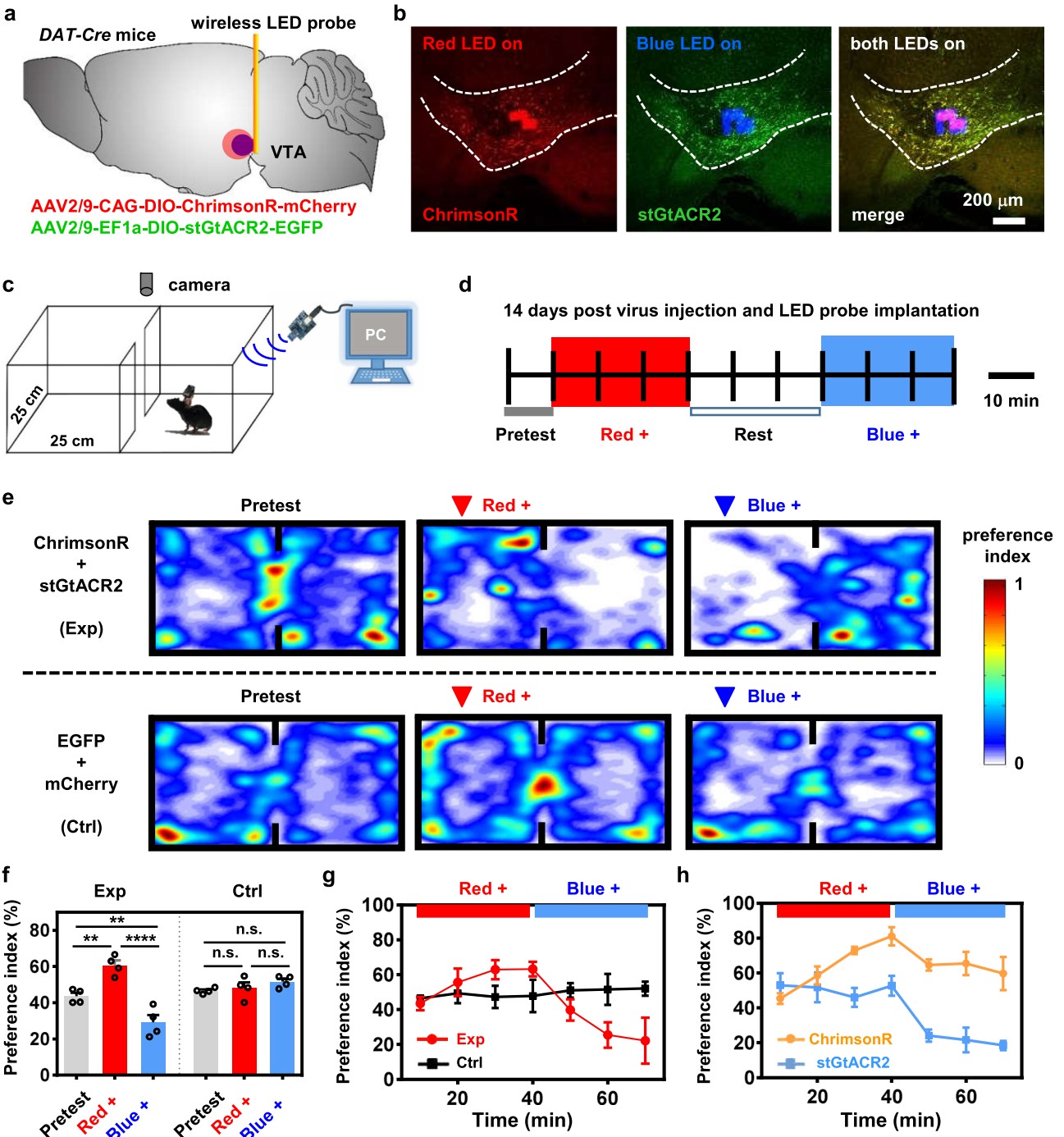

**Fig. 3 Bidirectional, in vivo optogenetic regulation of mice preference or aversion behaviors with wirelessly operated, dual-color microscale LED (micro-LED) probes. a** Diagram illustrating the combined viral injection (AAV2/9-CAG-DIO-ChrimsonR-mCherry and AAV2/9-EF1a-DIO-stGtACR2-EGFP) followed by implanting the micro-LED probe into the VTA of *DAT-cre* mice. **b** Micrographs of the VTA showing co-expression of stGtACR2 and ChrimsonR, with a dual-color micro-LED probe embedded into the tissue. **c** Illustration of the real-time place preference test with a two-compartment arena, where a mouse receives signals for dual-color control and its locomotion is recorded by a camera. **d** Patterns used for optogenetic modulations, including a 10-min pretest, a 30-min red LED stimulation (20 Hz, 10-ms pulse, current 7 mA), a 30-min rest, and a 30-min blue LED stimulation (continuous, current 5 mA). **e** Representative heatmaps showing real-time preference and aversion behavior following red or blue stimulation for mice co-expressing stGtACR2 + ChrimsonR (experiment, or Exp) or EGFP + mCherry (control, or Ctrl). **f** Summary of preference indices (the ratio of the time that mice spend in the left chamber to the whole recorded time) under red and blue stimulations for Exp and Ctrl groups ($n = 4$ mice each). Two-way ANOVA, Sidak's multiple-comparisons test (baseline versus stim), *$P < 0.05$, **$P < 0.01$, ***$P < 0.001$, n.s. $P > 0.05$. **g** Preference indices measured at different times under red and blue stimulations for Exp and Ctrl groups ($n = 4$ mice each). **h** Preference indices measured at different times under red and blue stimulations for groups expressing only ChrimsonR or stGtACR2 ($n = 3$ mice each). All data are represented as mean ± s.e.m. See Supplementary Table 1 for detailed statistical analysis. Source data are provided as a Source Data file.

encapsulation. Also, mass spectrometric analysis shows that the brain tissue has a minimal accumulation of dissolved Cu element 5 weeks' post-implantation (Supplementary Fig. S14). In addition, we perform dynamic calcium imaging on acute brain slices for AAV-hsyn-GCaMP6m expressing mice 2 weeks' post-implantation (Supplementary Fig. S15). Cell activities can be effectively recorded in neurons ~100 μm away from the lesion region, in agreement with the histological analysis. Moreover, the parylene-based, waterproof encapsulation maintains the micro-LEDs' chronic and stable operation for more than 200 days after implantation (Supplementary Fig. S16).

In vivo experiments are performed within a real-time place preference/avoidance (RTPP) paradigm. After 14 days of virus and micro-LED probe implantation, we place mice in a two-compartment apparatus and monitor their behaviors with a camera during optogenetic modulations (Fig. 3c and Supplementary Fig. S17). Figure 3d provides the test protocol, including a baseline pretest (10 min), red LED stimulation (pulsed wave, 30 min), a rest phase (30 min), and blue LED stimulation (continuous wave, 30 min). The red or blue LED stimulations are only provided once the mice appear in the left chamber. Representative heatmaps in Fig. 3e show the change of real-time behaviors during optogenetic stimulations, for mice expressing different combinations of opsins. Summarized results in Fig. 3f clearly show that mice co-expressing ChimsonR and stGtACR2 present strong preference and avoidance behaviors following red and blue stimulations, respectively. In addition, the application of both red and blue stimulations in opposite chambers further enhances the preference indices of mice compared to those only experiencing single-color stimulations as well as the control group (co-expressing EGFP and mCherry), demonstrating the efficacy of bidirectional modulation (Supplementary Fig. S18).

By contrast, the control groups without photosensitive opsins (expressing only EGFP and mCherry) show little difference of behaviors under optical stimulations. More results are provided in Supplementary Figs. S19 and S20, comparing these mice with other groups expressing only ChrimsonR or stGtACR2. When introducing only ChrimsonR, mice exhibit place preference under both red and blue stimulations. For the group with only stGtACR2, blue light effectively evokes place aversion while red light generates little influence on mouse behaviors. Fig. 3g, h present the dynamic behavior change at different time courses. These results are consistent with the spectral characteristics of the opsins as well as the in vitro neural responses at the cellular level (Fig. 2), elucidating the utility of bidirectional modulations with wireless, dual-color micro-LED probes. In conclusion, the implantable wireless dual-color LED probe combined with two spectrally distinct channelrhodopsins can realize bidirectional control of the place preference or aversion in freely behaving mice.

**Bidirectional optogenetic modulation of social interactions among multiple mice, with wireless dual-color micro-LED probes**. Modulating dopaminergic neurons in the VTA region also influences the social preference, in particular, affective and aggressive behaviors[49,50]. Compared with tethered stimulation systems, our wirelessly operated micro-LED probes with independently addressable illuminations present prominent advantages in studying social interactions among multiple objects (Supplementary Movie S3). In addition, the colocalized, dual-color emission capability allows the bidirectional regulation of social activities in real-time, which is difficult to access with single-color stimulations[49,51]. In paired-object interactions (Fig. 4a), two mice co-expressing ChrimsonR and stGtACR2 in the VTA are placed in an open-field arena, and synchronized red

or blue stimulations are provided for 5 min each. Mice behaviors are video recorded throughout the test period, with social activities (including sniffing and attacking) labeled and scored (Fig. 4b). The total time spent in these activities is summarized and compared among different scenarios (red light on, blue light on, and resting) in Fig. 4c. In comparison to the resting state, red or blue LED stimulations strongly promote or suppress social interactions between two objects in the experiment group, while no significant effects are observed in the control group (expressing EGFP and mCherry).

The wireless micro-LED probes can further be applied to interrogate more complicated interaction behaviors among multiple (>2) objects. In Fig. 4d, we place three mice in the same social chamber and provide these animals with alternating red or blue LED stimulations. In such a testing paradigm, two mice receive red stimulations while the third one receives blue stimulation, then vice versa (Fig. 4e). The mice with red stimulations spend more time interacting with each other compared to the individual under blue illuminations, and the social preference can be reversed by alternating the stimulation patterns among these three mice in real-time (Fig. 4f). By contrast, summarized data among control groups present no significant difference under red or blue stimulations. Collectively, these social interaction explorations clearly demonstrate the unique advantage of our wireless and bidirectional LED stimulators in the interrogation of complex animal behaviors.

**Bidirectional optogenetic regulation of dopamine release in vivo, with wireless dual-color micro-LED probes**. The effective bidirectional behavior manipulations demonstrated in Figs. 3 and 4 can be ascribed to the light-induced activation and inhibition of dopaminergic neurons in the VTA. We further investigate the effect of optogenetic stimulations on dopamine release using in vivo experiments described in Fig. 5a. Similarly, combined opsins (ChrimsonR and stGtACR2) are expressed and followed by implanting the dual-color micro-LED probe in the VTA of *DAT-cre* mice. While stimulations are performed in the VTA, dopamine signals are simultaneously monitored in one of its main projection areas, the nucleus accumbens (NAc)[52–54]. A genetically encoded indicator GRAB$_{DA2m}$[55] is expressed in the NAc, where the fluorescence signals are recorded with a fiber photometer[30,31,48,49,56,57]. The combination of micro-LED-based optogenetic stimulations and fiber-based photometric recordings realizes all-optical interrogation of neural circuits. During the surgery, we carefully select the insertion direction of the micro-LED probe, to obtain desirable stimulation efficacy in the VTA and minimize the optical crosstalk that could disturb photometric recording in the NAc (Supplementary Fig. S21). Fluorescence images in Fig. 5b show the expressions of GRAB$_{DA2m}$ in the NAc as well as ChrimsonR and stGtACR2 in the VTA.

Figure 5c plots temporally resolved dopamine release in response to stimulations caused by red and blue micro-LEDs in the same mice. Other representative curves are presented in Supplementary Fig. S22, with results summarized in Fig. 5d. Illuminating the VTA by red or blue micro-LEDs (with injection currents from 0 to 10 mA) clearly induces increased or decreased dopamine signals in the NAc, which correlates with neuron responses at the cellular level (Fig. 2). Experiments operated among mouse groups expressing different opsins are also performed, with results compared in Fig. 5e. In contrast to the group expressing both opsins (ChrimsonR and stGtACR2), mice with only ChrimsonR show increased dopamine release under both red and blue illuminations, and dopamine suppression is observed under blue illumination for the mice with only stGtACR2. The control group (with EGFP and mCherry) exhibits

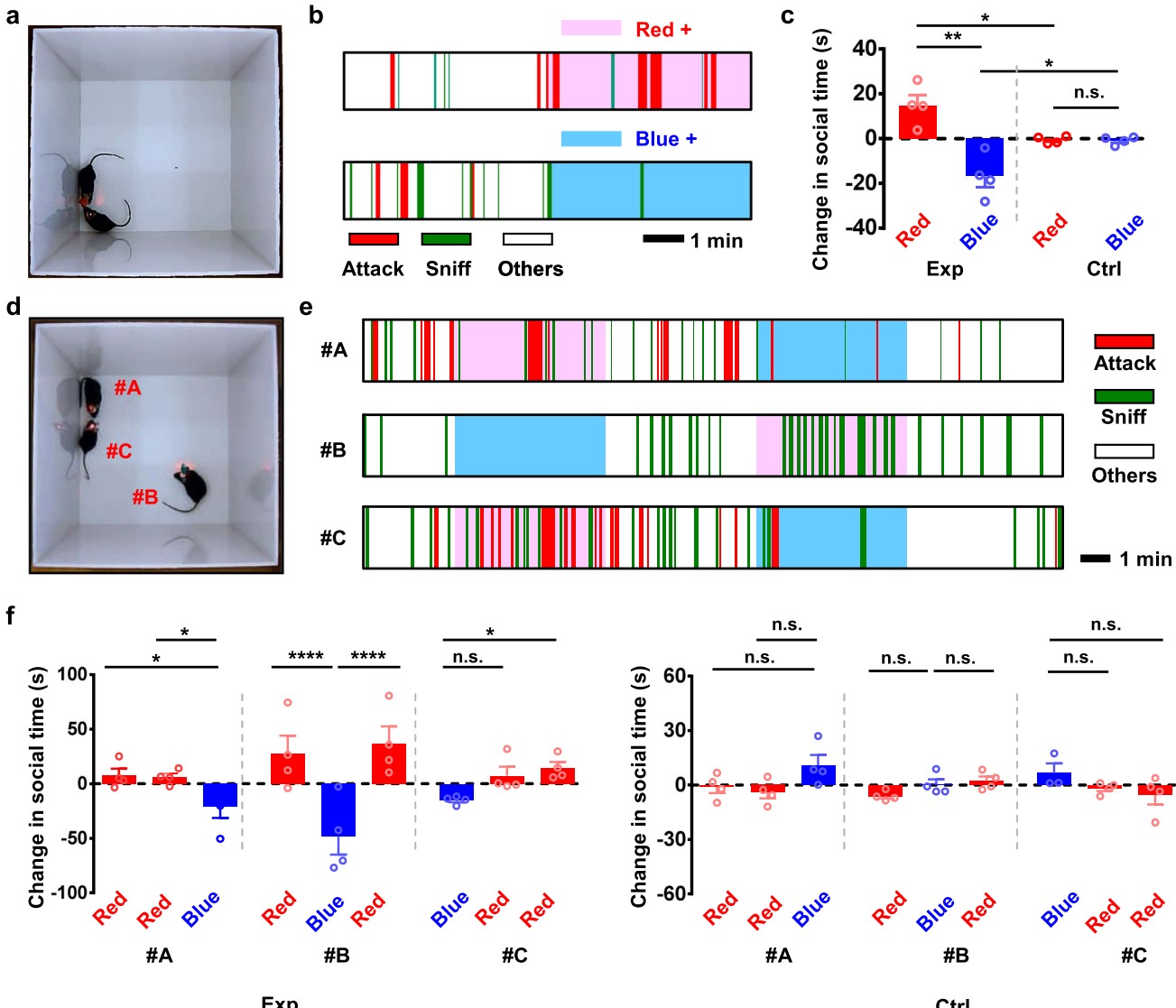

**Fig. 4 Bidirectional, in vivo optogenetic control of mice social interactions with wirelessly operated, dual-color micro-LED probes. a** Photograph of two *DAT-cre* mice with wireless probes in an arena (30 × 30 cm²). Experiment groups (Exp) express stGtACR2 + ChrimsonR, and control groups (Ctrl) express EGFP + mCherry. **b** Representative behavioral sequences recorded for one mouse receiving red or blue stimulations during social interactions. Here sniffing and attacking behaviors are highlighted and labeled, and other behaviors include locomotion, resting and escaping. **c** Summary of changes in social interactions under red or blue stimulations. Here, we compare the time spent in sniff and attack during the 5 min light stimulation with the 5 min resting state. n = 4 groups for both Exp. and Ctrl. Two-way ANOVA, Sidak's multiple-comparisons test (experiment versus control). **d** Photograph of three *DAT-cre* mice with wireless probes in an arena. **e** Representative behavioral sequences recorded for the three mice (#A, #B, #C) receiving red or blue stimulation during social interactions. **f** Summary of changes in social interactions under red or blue stimulations for each mouse. n = 4 groups for both Exp (left) and Ctrl (right). Two-way ANOVA, uncorrected Fisher's LSD (red versus blue). *P < 0.05, **P < 0.01, ***P < 0.001, ****P < 0.0001, n.s. P > 0.05. All data are represented as mean ± s.e.m. See Supplementary Table 1 for detailed statistical analysis. Source data are provided as a Source Data file.

no response under optical stimulations. These results evidently explain in vivo preference and avoidance behaviors presented in Fig. 3, showcasing the capability for precise control of neural activities based on bidirectional, dual-color modulations.

We further investigate the level of the released dopamine with coincident red and blue LED radiations. Figure 5f plots example traces of dopamine transients, in response to the red and blue micro-LEDs under pulse and continuous operations (6-s duration) with varied currents. Statistical data for accumulative dopamine concentration variations (unit: area under the curve during stimulation, or AUC) are summarized in Fig. 5g and Supplementary Fig. S23. Decreasing the power of the red LED and increasing the power of the blue LED to cause the decline of dopamine release. In addition, the blue LED current required to

suppress the dopamine release is much lower than that of the red LED required for dopamine level rises, which further conforms with the cellular activities reported in Fig. 2. It should be noted that the in vivo results in Fig. 5g are not quantitatively identical to the in vitro electrophysiological ones in Fig. 2m. In fact, higher irradiances for blue LEDs are needed for effective inhibition in vivo. This may be due to the fact that blue light has higher absorption in tissue and stronger illumination is required to affect enough cells in the VTA.

Figure 5h (and more examples in Supplementary Fig. S24) presents dopamine dynamics in response to alternated or combined red and blue LED stimulations, for mice expressing ChrimsonR and stGtACR2. The red illumination triggers the elevated dopamine release, while the use of blue light can

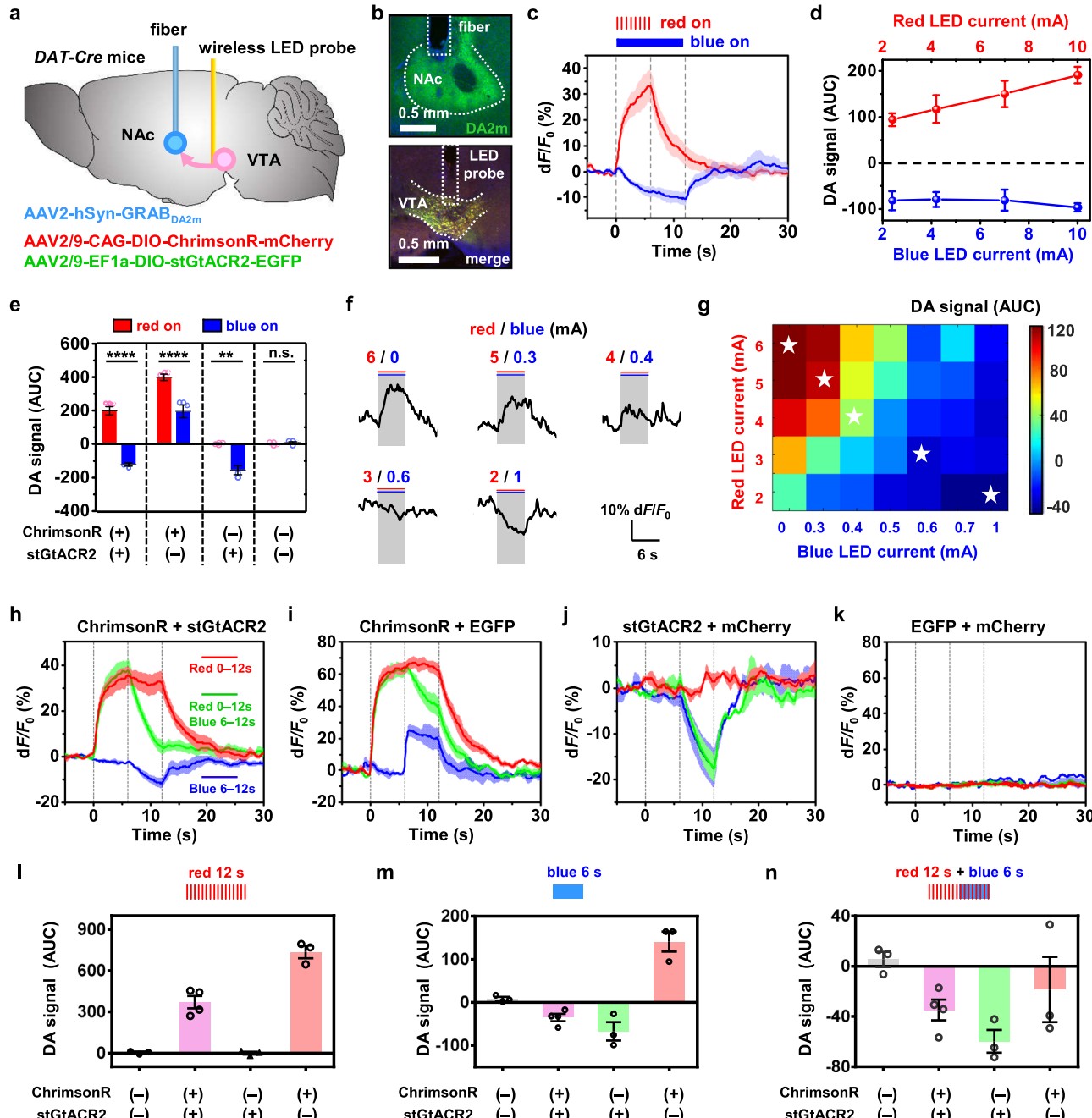

**Fig. 5 Bidirectional, in vivo optogenetic control of dopamine (DA) release recorded in the nucleus accumbens (NAc) during dual-color stimulation in the VTA. a** Diagram illustrating the viral injection followed by implanting the micro-LED probe and optical fiber of *DAT-cre* mice. **b** Micrographs of (top) expression of DA2m and the fiber track, (bottom) the co-expression of stGtACR2 and ChrimsonR (merged color) and the probe track. **c** Traces of DA signals responding to stimulations by the red LED (*n* = 4 mice) and the blue LED (*n* = 4 mice). **d** Accumulative DA signals responding to red and blue stimulations under different LED currents (*n* = 4 mice). AUC: area under the curve during stimulation. **e** Summary of measured DA signals responding to red or blue stimulations (Red: 6 s; Blue: 12 s) for different mouse groups. Two-way ANOVA, Sidak's multiple-comparisons test (red versus blue). **f** Example traces and **g** summarized plot of DA signals under combined red and blue stimulations with varied LED currents (*n* = 3 mice). The stars indicate results for five traces shown in (**f**). **h–k** Plots of optogenetically evoked DA transients responding to different stimulation patterns (Red: red LED, 0–12 s; Blue: blue LED, 6–12 s; Green: red LED 0–12 s and blue LED 6–12 s), for different mouse groups expressing different opsins. **l–n** Summary of measured DA signals (AUC) for different mouse groups in (**h–k**) ChrimsonR + stGtACR2 (*n* = 4 mice), ChrimsonR + EGFP (*n* = 3 mice), stGtACR2 + mCherry (*n* = 3 mice), EGFP + mCherry (*n* = 3 mice). All data are represented as mean ± s.e.m. The solid lines and shaded areas (**c**, **h–k**) indicate the mean and s.e.m. See Supplementary Table 1 for detailed statistical analysis. Source data are provided as a Source Data file.

effectively and promptly suppress the signal. Similar illumination patterns are implemented on mice expressing different opsins (Fig. 5i–k) and results are summarized and compared in Fig. 5l–n. Evidently, red or blue illuminations bidirectionally regulate

dopamine signals in the experiment group (with both ChrimsonR and stGtACR2). On the other hand, mice expressing only ChrimsonR can be excited by both red and blue light, and those with only stGtACR2 can only be inhibited by blue light and show

no reaction under red light. Finally, the control group (with EGFP and mCherry) does not respond to any optical stimulation patterns. Taken together, these results clearly demonstrate that wireless dual-color LED combined with optogenetics enable efficient, bidirectional control of neuronal activity in the VTA and DA release of the terminal in the NAc. It is also noted that fluorescence signals experience a drop when the blue LED is operating during red stimulation for the mice only expressing ChrimsonR (Fig. 5i). Such a response is unexpected considering the fact that blue light also activates ChrimsonR. One possible explanation is that when both pulsed red light and continuous blue light are imposed simultaneously, their activation effects on ChrimsonR are not additive but competitive. On the other hand, the application of both LEDs possibly induces additional heating effects, complicating the optical and thermal responses of ChrimsonR. Nevertheless, understanding spectral and temporal properties for these opsins requires further explorations in the future work.

## Discussion

Optogenetic modulations offer a viable solution for cell-type-specific targeting, but still demand biological and engineering tools for versatile functionalities. In this work, we report that bidirectional optogenetic activation and inhibition in behaving mice can be achieved by co-expressing spectrally distinct excitatory and inhibitory channelrhodopsins, cooperating with a wireless, dual-color micro-LED probe for colocalized red and blue stimulations. With collective results at the cellular, circuit, behavioral and social levels, successful proof-of-concept demonstrations in the VTA-NAc circuitry in freely moving mice clearly establish the utility of such a synergistic methodology.

The unique characteristics of thin-film, microscale optoelectronic devices, and their integration strategies implicate prominent advantages over conventional waveguide-coupled or single-color LED-based sources. In terms of technology developments, more directions can be explored and some preliminary efforts are presented below: (1) Multiple red and blue micro-LEDs can be assembled on the implantable probe via the stacking approach, realizing multi-channel, bidirectional modulations imposed on multiple brain regions (Supplementary Movie S4). It is noted that the efficacy of preference behavior regulations demonstrated in Fig. 3 would also be enhanced by implementing dual-channel stimulations bilaterally in the VTA[58–61]. (2) The use of miniaturized wireless control circuits allows independently addressing multiple, untethered animal objects, making the interrogations of more complex social behaviors feasible[62–65]. (3) Micro-LEDs with other colors can also be included and target specific opsins more efficiently. For example, a tri-color probe can be achieved by similarly stacking red, green, and blue micro-LEDs in a vertical structure (Supplementary Movie S5)[66]. Independently controlled micro-LEDs can interact with different opsins that express in the same or different populations of cells. (4) Micro-LED-based optogenetics can also be combined with previously reported electrical, optical, and chemical sensors as well as microchannel devices for drug administrations[67–70], achieving closed-loop neural activity stimulations and detections. While arrays of blue InGaN micro-LEDs can be directly grown on silicon and integrated with recording electrodes[41], the combination of simultaneous dual-color stimulations and electrophysiological recordings in the deep brain region like VTA require more sophisticated device design and fabrication and have not been attempted here. Considering the system dimensions, wireless circuits can be further miniaturized by employing inductive coupling-based power transfer strategies[24,26]. In terms of biocompatibility, tissue lesions can be further reduced by using thinner substrates with much

lower stiffness[45]. Taken together, we envision a multi-spectral, multi-channel, multi-objective, and multi-functional optical neural interface, realizing wireless, real-time, bidirectional, close-loop, and simultaneous interrogations of neural signals in living animals with desirable system stability and biocompatibility.

On the other hand, the efficacy of optogenetic modulations can be further optimized by advanced biological engineering. Other combinations of opsins, like ChR2 and Jaws[36], have also been proved to be feasible. A recent effort fuses excitatory and inhibitory channelrhodopsins (ChrimsonR and GtACR2) in a single, trafficking-optimized tandem protein (BiPOLES), improving the efficiency of opsin co-expression[71]. Moreover, the combination of super sensitive opsins (Jaws[15], ChRmine[72], ReaChR[73], SOUL[13], etc.) could be explored for bidirectional modulations via extracranial illumination without implants. Since relatively large irradiance (>200 mW/mm²) is required to penetrate into the deep tissue, corresponding light sources and control systems have to be miniaturized for wireless operation in mice. In addition to the bidirectional modulation of the same cell population, the dual-color (or even tri-color) micro-LED probe can also selectively activate genetically distinct and spatially intermingled cells expressing spectrally resolved opsins (for example, blue for ChR2 or GtACR2, green for C1V1 or GtACR1 and red for ChrimsonR or halorhodopsin)[3,7,8,21] and provide prolonged activation of neuronal activity by regulating step-function opsins (for example, SSFO, SOUL)[11,13,74], which is also of critical importance for understanding the circuit structures and functions. It should be also noted that these dual-color or multi-color modulations should be carefully designed, and their efficacies are highly dependent on the spectral overlap among opsins, the irradiance of different colors, levels of viral expressions, etc. Possible applications would include bidirectional modulations of neural activities by leveraging electrophysiological or neurotransmitter signals for medications of seizures or Parkinson's disease[29,75]. Besides the brain system, the spinal cord and peripheral neural circuits can also be targeted for medical treatments[76–78]. In summary, the results presented here provide a viable means for fundamental neuroscience studies and advanced neuroengineering applications.

## Methods

### Device fabrication

*Fabrication of red and blue micro-LEDs.* Details about the thin-film micro-LED structures and fabrication processes can be found in our previous work[37,39]. Via metal-organic chemical vapor deposition (MOCVD), red and blue LEDs are originally grown on gallium arsenide (GaAs) and sapphire substrates, respectively. The red LED active device contains an indium gallium phosphide (InGaP) based multiple quantum well structure (thickness ~5.6 μm) grown on GaAs with an $Al_{0.95}Ga_{0.05}As$ sacrificial interlayer. The blue LED active device contains an indium gallium nitride (InGaN) based multiple quantum well structure (thickness ~7 μm) grown on sapphire. Photolithographic process, wet/dry etching and metallic electrode deposition define the LED geometry (size ~180 μm × 125 μm). Freestanding, thin-film red LEDs are formed by selectively removing the $Al_{0.95}Ga_{0.05}As$ sacrificial layer in a hydrofluoric acid (HF)-based solution. Freestanding, thin-film blue LEDs are formed by a laser lift-off (with a KrF excimer laser at 248 nm). Poly(dimethylsiloxane) (PDMS) based stamps are employed to pick up released thin-film micro-LEDs and transfer them onto other foreign substrates.

*Fabrication of thin-film filters.* Details about the thin-film filter fabrication can be found in our previous work[40]. The wavelength selective thin-film filters is based on a multilayered titanium dioxide ($TiO_2$) and silicon dioxide ($SiO_2$) structure (thickness ~6.6 μm) comprising (from top to bottom): 19 nm $TiO_2$/15 periods (89 nm $SiO_2$ + 52 nm $TiO_2$)/63 nm $SiO_2$/66 nm $TiO_2$/22 periods of (73 nm $SiO_2$ + 42.5 nm $TiO_2$)/154 nm $SiO_2$. The structure is deposited on GaAs by ion beam-assisted sputter deposition. The filter shape (size ~180 μm × 125 μm) is defined by laser milling (Nd: $YVO_4$ laser, 1064 nm). After removing the GaAs substrate in solutions ($NH_4OH:H_2O_2:H_2O = 1:1:2$), freestanding thin-film filters can be picked up and transferred printed onto other foreign substrates by PDMS stamps.

*Fabrication of dual-color micro-LED probes.* The process flow to fabricate the dual-color micro-LED probe is described in Supplementary Fig. S1. The micro-LED probe is based on a flexible substrate made of composite polyimide (PI) and copper

(Cu) films (from top to bottom: Cu/PI/Cu = 18/25/18 μm, from DuPont). From bottom to top, a red LED, an optical filter and a blue LED are sequentially transferred printed on the film by PDMS in a vertical stack, via a customized alignment setup (with a lateral alignment accuracy of ~2 μm). Between each device layer, spin-coated SU-8-based epoxy (thickness ~2–5 μm) serves an optically transparent and electrically insulating bonding layer. Each individual LED is metalized with sputtered metal layers (Cr/Au/Cu/Au = 10/100/500/100 nm). Laser milling (ultraviolet laser 365 nm) defines the needle shape (width ~320 μm, length ~5 mm). A bilayer of PDMS (~20 μm, by dip-coating) and parylene (~15 μm, by CVD) is used for waterproof encapsulation. For comparison in Fig. 1g, laterally arranged red-blue micro-LED probes are also fabricated with similar procedures.

### Device characterization and modeling

*Structural and optoelectronic characterizations.* Scanning electron microscopic (SEM) images are taken with a ZEISS Auriga SEM/FIB Crossbeam System. Optical images are captured by a microscope MC-D800U(C). The LED emission spectra are measured with a spectrometer (HR2000 + , Ocean Optics). Current–voltage characteristics are recorded by a Keithley 2400 source meter. LED irradiances and external quantum efficiencies are measured using a spectroradiometer with an integrating sphere (LabSphere Inc.). The angular-dependent emission profiles are measured by a standard Si photodetector (DET36A, Thorlabs) with the device mounted on a goniometer. The probe's mechanical properties are measured with dynamic mechanical analysis (DMA Q800, TA Instruments). For in vitro tests, micro-LED probes are implanted into the brain phantom. The brain phantom is made of agarose (0.5% w/v), hemoglobin (0.2% w/v), and intralipid (1% w/v) mixed in phosphate buffer solutions (98.3% w/v). After stirring, the mixed liquid is heated to boiling and then naturally cooled to room temperature, forming the gels.

*Optical modeling.* A ray-tracing method based on Monte Carlo simulations (TracePro free trial version) is used to simulate the light propagation in the brain tissue. In the model, the tissue (size $5 \times 5 \times 5$ mm$^3$) has absorption coefficients of 0.2 /mm and 0.08 /mm, and scattering coefficients of 47 /mm and 35 /mm, for blue (475 nm) and red (630 nm) wavelengths, respectively. The tissue has an anisotropy factor of 0.85 and a refractive index of 1.36[79]. Planar sources similar to the surface of micro-LEDs (size $125 \times 180$ μm$^2$) emit $10^6$ rays for each monochromic wavelength (475 nm and 630 nm), assuming a Lambertian distribution. For the stacked micro-LEDs, the blue and red sources are overlaid. For the lateral case, the two sources are adjacent to each other.

*Thermal characterizations.* For in vitro experiments, the micro-LED probes are implanted into the brain phantom (~0.5 mm underneath the surface, at a depth of ~5 mm), and various currents are injected into the LEDs. The temperature distributions on the phantom surface are mapped with an infrared camera (FOTRIC 228, with software AnalyzIR). For comparison, the temperature inside the phantom is also measured by inserting an ultra-fine flexible micro thermocouple (Physitemp, IT-24P) (~0.3 mm above the LED probe tip). For in vivo experiments, the micro-LED probe is bonded with the same thermocouple and inserted into the anesthetized mouse brain. Steady-state temperature rises are recorded from the thermocouple with a thermometer (Physitemp, BAT-12).

*Thermal modeling.* 3D steady-state heat transfer models are established by finite-element analysis (Comsol Multiphysics). Materials and corresponding parameters (density, thermal conductivity and heat capacity) used in the model include: brain tissue (1.1 g/cm$^3$, 0.5 W/m/K, 3.7 J/g/K), parylene (1.2 g/cm$^3$, 0.082 W/m/K, 0.71 J/g/K), PDMS (0.98 g/cm$^3$, 0.16 W/m/K, 1.5 J/g/K), polyimide (1.4 g/cm$^3$, 0.15 W/m/K, 1.1 J/g/K), GaN (6.1 g/cm$^3$, 130 W/m/K, 0.49 J/g/K), Cu (9.0 g/cm$^3$, 400 W/m/K, 0.39 J/g/K), SU-8 (1.2 g/cm$^3$, 0.2 W/m/K, 1.2 J/g/K), GaP (4.4 g/cm$^3$, 110 W/m/K, 0.43 J/g/K). The boundary condition is natural heat convection to air at 20 °C. The micro-LED serves as the heat source, with the input thermal power estimated by $P = V \times I \times (1 - EQE)$, where $V$, $I$, and EQE are the measured voltage, current, and corresponding external quantum efficiencies for LEDs. The thermocouple is also included in the model for comparison, which is 0.3 mm above the LED probe tip. It consists of two metal wires: copper (diameter 31 μm) and constantan (45Ni-55Cu alloy, thermal conductivity 21 W/m/K, diameter 31 μm), which are encapsulated by polyester to form a wire with an outer diameter of 127 μm.

### Wireless circuit design

A customized circuit module is designed to wirelessly operate the implantable micro-LED probe by independently controlling the red and the blue micro-LEDs for bidirectional modulation. The schematic diagram is depicted in Supplementary Fig. S5, with a detailed circuit diagram, layouts, and components presented in Supplementary Figs. S6–S9. The core components include: a micro-controller (nRF24LE1, Nordic Semiconductor) operated at 2.4 GHz for data communication, and two LED drivers (ZLED7012, Renesas Electronics Corp.) with programmable current levels, frequencies, pulse widths for providing a constant current to micro-LEDs[38]. The driver provides programmable constant current levels ranging from 1.8 to 20 mA and also supports arbitrary waveforms. Two red LEDs are used on the circuit board outside the brain for signal indication, without affecting the mouse behaviors[80]. Control signals are wirelessly transmitted from the transmitting module connected to a laptop computer via an

antenna operating at a radio frequency of 2.4 GHz, with a communication distance up to 50 m. Different from our previous design[81], here a polyimide-based flexible circuit board is implemented for reduced weight. The wireless circuit has a footprint of ~$2.2 \times 1.3$ cm$^2$ and a weight of 1.9 g (including a 0.9-g rechargeable lithium-ion battery with a capacity of 35 mAh). Used in the experimental conditions in this paper, the battery life can reach up to 2 h. The circuit can be connected with the implanted micro-LED probes via a flexible printed circuit (FPC) connector before the animal experiment (Supplementary Fig. S8).

### Biological studies

*Ethical statement.* Animal care is in accordance with the institutional guidelines of the National Institute of Biological Sciences in Beijing (NIBS), with protocols proved by Institutional Animal Care and Use Committee (IACUC). Wildtype, male C57BL6/N mice are purchased from VitalRiver (Beijing, China). *DAT-Cre* (B6.SJLSlc6a3tm1.1(cre) Bkmn; JAX Strain 006660) and *CamkIIa-Cre* (B6.Cg-Tg(Camk2a-cre)T29-1Stl/J; JAX Strain 005359) male mice are at least six weeks old at the time of surgery. All animals are socially housed in a 12 h/12 h (lights on at 8 am) light/dark cycle (at room temperature ~25 °C, humidity 30–50%), with food and water ad alibitum.

*Virus production.* The AAV2/9-CAG-DIO-ChrimsonR-tdTomato (Plasmid #130909) vector[7] is from Addgene. The AAV2/9-EF1a-DIO-stGtACR2-EGFP-Kv2.1 plasmid is synthesized and constructed according to the original reports[8,9]. The AAV-hSyn-GRAB$_{DA2m}$ plasmid is provided by Prof. Yulong Li at Peking University[55]. AAV vectors are packaged into serotype2/9 vectors, which consist of AAV2 ITR genomes pseudotyped with AAV9 serotype capsid proteins. AAV vectors are replicated in HEK293 cells with the triple plasmid transfection system and purified by chloroform[82], resulting in AAV vector titers of about $5 \times 10^{12}$ particles/mL. Virus suspension is aliquoted and stored at −80 °C. AAV titers are measured using real-time qPCR. Adeno-associated viral particles are produced at the Vector Core Facility at the Chinese Institute for Brain Research, Beijing (CIBR).

*Stereotaxic surgery.* Adult *DAT-Cre* male mice are anesthetized with an intraperitoneal injection of Avertin (250 mg/kg) before surgery and then placed in a standard stereotaxic instrument for surgical implantation (Supplementary Fig. S12). After disinfection with 0.3% hydrogen peroxide, a small incision of the scalp is created to expose the skull. Then, 0.3% hydrogen peroxide is applied to clean the skull. A small craniotomy is made, followed by carefully removing the dura with a thin needle. A calibrated pulled-glass pipette (Sutter Instrument) is lowered to the VTA (coordinates 3.64 mm posterior to the bregma, 0.5 mm lateral to the midline, and 4.6 mm ventral to the skull surface). The virus is delivered through a small durotomy by a glass micropipette using a microsyringe pump (Nanoliter 2000 injector with the Micro4 controller, WPI).
Four different combinations of virus are used for in vivo experiments:

1. ChrimsonR + stGtACR2 (AAV2/9-CAG-DIO-ChrimsonR-tdTomato and AAV2/9-EF1a-DIO-stGtACR2-EGFP-Kv2.1, 1:1);
2. mCherry + EGFP (AAV2/9-EF1a-DIO-mCherry and AAV2/9-EF1a-DIO-EGFP, 1:1);
3. ChrimsonR + EGFP (AAV2/9-CAG-DIO-ChrimsonR-tdTomato and AAV2/9-EF1a-DIO-EGFP, 1:1);
4. stGtACR2 + mCherry (AAV2/9-EF1a-DIO-stGtACR2-EGFP-Kv2.1 and AAV2/9-EF1a-DIO-mCherry, 1:1)

A bolus of 0.4 μL of mixed virus is injected into the VTA at 46 nL/min. The pipette is held in place for 10 min after the injection and then slowly withdrawn. For the real-time place preference (RTPP) behavioral test, the dual-color LED probe implantation is carried out after virus injection and diffusion. The probe was slowly implanted at the side of the target brain areas with an angle of 45° to midline (coordinates 3.64 mm posterior to the bregma, 0.8 mm lateral to the midline, and 4.85 mm ventral to the skull surface). The probe is fixed to the skull with dental cement. For dopamine detection with fiber photometric recording, the AAV-hSyn-GRAB$_{DA2m}$ (0.2 μL) is injected into the NAc (coordinates: 1.18 mm front to the bregma, 0.75 mm lateral to the midline, and 3.75 mm ventral to the skull surface). Optical fiber implantation is carried out after virus injection and diffusion. A piece of optical fiber (FT200UMT, 200 μm O.D., 0.39 NA, Thorlabs) is fit into an LC-sized ceramic fiber ferrule (Shanghai Fiblaser, China) and implanted over the target brain areas with the tip 0.1 mm above the virus injection sites in the NAc. The ceramic ferrule is fixed to the skull with Cyanoacrylate adhesive (TONSAN 1454) and dental cement. After the virus injection and device implantation, mice are housed individually to prevent potential damage to the probe. All subsequent experiments are performed at least 2 weeks after injection to allow sufficient time for transgene expression and animal recovery.

*Immunohistochemistry.* The micro-LED probes are implanted into the VTA region and fixed to the skull with Cyanoacrylate adhesive (TONSAN 1454) and dental cement. After 1 day or 3 weeks following the surgery, mice are sacrificed with an overdose of pentobarbital and transcardially perfused with 0.1 M phosphate buffer saline and 4% paraformaldehyde. After pcryoprotection in 30% sucrose, brains are post fixed in 4% PFA for 4 h and 30% sucrose for 24 h, 40-μm-thick coronal sections are prepared by a cryostat (Leica CM1950). After 5-min phosphate-buffered saline (PBS) rinses for 3 times, the sections are blocked in PBST (PBS + 0.3%

Triton X-100) with 3% bovine serum albumin for 1 h, and incubated in rabbit anti-Iba1 (1:1000, Wako, 019-19741) and chicken anti-GFAP antibody (1:2000, Sigma, AB5541) dissolved in PBST at 4 °C for 24 h. After 5-min washes in PBS for five times, the sections are incubated with Alexa Cy3-conjugated goat anti-rabbit antibody (1:500, Jackson ImmunoResearch, 111-165-008) and Alexa 488 goat anti-chicken antibody (1:500, ThermoFisher Scientific, A11039) dissolved in PBST for 2 h at room temperature followed by 5-min washes in PBS for three times. Finally, the samples are coverslipped in 50% glycerol and slides and photographed with an automated slide scanner (VS120 Virtual Slide, Olympus) or a confocal scanning microscope (DigitalEclipse A1, Nikon). The numbers of DAPI, stGtACR2 and ChrimsonR-stained cells or the numbers of Iba1+ microglia and GFAP+ astrocytes are analyzed with Imaris (v9.5) and ImageJ (Fiji package) and their percentages in cell population indicated by DAPI are calculated.

*Element analysis.* To characterize the Cu concentration in the brain tissue, mice brains are extracted after perfusion with PBS, and then full dried by vacuum freeze dryer (ALPHA, Christ) for about 6 h after freezing at −80 °C overnight. After weighting, the dried brain tissues are added with nitro hydrochloric acid and placed overnight. After being microwave digested for 25 min (160 °C), the resolved brains solutions are analyzed with inductively coupled plasma mass spectrometry (Thermo ICP-MS iCAPQ, ThermoFisher, USA) to measure the Cu level.

*In vitro optogenetics evaluation.* Electrophysiological properties of the neurons in brain slices are measured with whole-cell patch-clamp recording technique following a previously described procedure[31]. After 2 weeks of virus expression (ChrimsonR + stGtACR2), mice are deeply anesthetized with pentobarbital (100 mg/kg i.p.) and intracardially perfused with 5 mL ice-cold oxygenated modified artificial cerebrospinal fluid (ACSF) at a rate of 2 mL/min. After perfusion with modified ACSF containing (in mM): 225 sucrose, 119 NaCl, 2.5 KCl, 0.1 CaCl₂, 4.9 MgCl₂, 1.0 NaH₂PO₄, 26.2 NaHCO₃, 1.25 glucose, 3 kynurenic acid, and 1 Na L-ascorbate (Sigma-Aldrich), brains are quickly removed and placed in ice-cold oxygenated ACSF containing (in mM): 110 choline chloride, 2.5 KCl, 0.5 CaCl₂, 7 MgCl₂, 1.3 NaH₂PO₄, 25 NaHCO₃, 20 glucose, 1.3 Na ascorbate, and 0.6 Na pyruvate. Sagittal sections (200 μm thick) are prepared using a Leica VT1200S vibratome. Brain slices are incubated for 1 h at 34 °C with ACSF saturated with 95% O₂/5% CO₂ and containing (in mM) 125 NaCl, 2.5 KCl, 2 CaCl₂, 1.3 MgCl₂, 1.3 NaH₂PO₄, 25 NaHCO₃, 10 glucose, 1.3 Na ascorbate, and 0.6 Na pyruvate. The internal solution within whole-cell recording pipettes (3–6 MΩ) contains (in mM): 130 K-gluconate, 10 HEPES, 0.6 EGTA, 5 KCl, 3 Na₂ATP, 0.3 Na₃GTP, 4 MgCl₂, and 10 Na₂-phosphocreatine (pH 7.2–7.4). Voltage-clamp recordings are performed using a MultiClamp 700B amplifier (Molecular Devices). Traces are lowpass filtered at 2.6 kHz and digitized at 10 kHz (DigiData 1440, Molecular Devices). The data are acquired and analyzed using Clampfit 10.0 software (Molecular Devices). Red and blue illuminations are provided by a fiber-coupled to two laser sources (633 nm and 473 nm).

*In vivo extracellular electrophysiology recordings.* We perform the extracellular single-unit recording as our previously described procedure[83]. Adult CamkIIa-Cre mice are anesthetized with an intraperitoneal injection of Avertin (250 mg/Kg). A bolus of 1 μL of virus (AAV2/9-CAG-DIO-ChrimsonR-tdTomato and AAV2/9-EF1a-DIO-stGtACR2-EGFP-Kv2.1, 1:1) is injected in neocortex and hippocampus at different depths (AP − 1.6 mm, ML 1.1 mm, DV 0.4 to 2.0 mm increments; 200 nL/site). A silver wire (127 μm diameter, A-M system) is attached to a skull-penetrating M1 screw above the olfactory bulb to serve as the ground reference. For head-fixed preparations, a custom-made titanium head-plate is secured to the skull with dental acrylic. Recordings are performed 2 weeks after viral injection. Mice are held in place with a header bar cemented to the skull. The dual-color micro-LED probe is lowered into the neocortex with a 15° angle from medial to lateral and another 10° angle from anterior to posterior. 8 stereotrodes with Pt-Ir wires (diameter 17.5 μm, platinum 10% iridium, California Fine Wire, USA) are used for extracellular electrophysiology recordings. The distance between the stereotrodes and the micro-LEDs is ~500 μm. To reduce the noise induced by the stray capacitance coupling from LEDs to electrodes, we use a tinfoil shield to encompass the electric circuit of the LED. Voltage signals are digitized and recorded by an Open-Ephys board (http://www.open-ephys.org/). Signals from each recording electrode are bandpass-filtered between 300 and 6000 Hz and sampled at 25 kHz. Single units are discriminated with principal component analysis (Offline sorter, spike2, CED, UK). Recorded units smaller than 2.5 times the noise band or in which interspike intervals are shorter than 2 ms are excluded from further analysis. Red light (pulse width 0.2 s, interval 3 s, 50–100 pulses, LED current 10 mA) and blue light (pulse width 1 s, interval 3 s, 50–100 pulses, LED current 1 mA) are used to activate or inhibit neurons. Data are analyzed and plotted with spike2 and a custom-developed MATLAB program[84].

*Calcium imaging in acute brain slice.* Mice are injected with 0.4 μL of Raav2/9-hsyn-GCaMP6m into the S1. After 2 weeks of recovery, acute brain slices are prepared. Ca²⁺ fluorescent signals in GCaMP6m expressing cells are captured with a 20× water immersion objective on a confocal microscope (FV1000, Olympus). For cell activation, 1 μM α-amino-3-hydroxy-5-methyl-4-isoxazolepropionic acid (AMPA, No. A6816, Sigma-Aldrich) is added in ACSF to enhance the Ca²⁺ signals. Control experiments are performed by applying pure ACSF. Fluorescence images are analyzed via ImageJ. Normalized fluorescence

changes are calculated as $dF/F = (F − F_0)/F_0$, where $F_0$ is the baseline intensity (before 10 s).

*Real-time place preference test.* After 2 weeks of recovery and AAV expression in *DAT-cre* mice, real-time place preference/avoidance (RTPP) tests are performed in a custom-made two-compartment apparatus. The RTPP apparatus comprises two white plastic chambers (size: 25 × 25 × 30, L × W × H in cm), separated by half-open opaque plastic walls in the middle (8 × 30, L × H cm). An RTPP test consists of three phases: 10-min baseline (pretest), 30-min red LED stimulation, 30-min rest (no stimulation), and 30-min blue LED simulation. During the pretest phase, individual mice are allowed to freely explore the entire apparatus for 10 min. Mice that exhibit a strong initial preference or avoidance for one of the side chambers (preference index >60% or <40%) are excluded from further experiments. During the red LED stimulation phase, mice that entered the left chamber received red light pulses (20 Hz, 20-ms pulse, 7 mA). During the blue light stimulation phase, mice that entered the left chamber receive constant blue light (continuous, 5 mA). The locomotion of mice is assessed from the video recording data using a custom-developed MATLAB program[85].

*Social interaction test.* After 2 weeks of recovery and AAV expression, social interaction tests are performed with *DAT-cre* male mice. Two or three mice are placed in an open-field arena (30 × 30 cm²), wireless dual-color micro-LED probes provide red (20 Hz, 20-ms pulse, 7 mA) and blue (continuous, 5 mA) stimulations, and their behaviors are recorded by a camera. In paired-mice experiments, the two mice receive the same 3-phase stimulations: (1) 5-min baseline (pretest, no stimulation), 5-min red LED stimulation, 5-min free interaction (no stimulation), and 5-min red LED stimulation; (2) 2-h rest in individual housing; (3) 5-min baseline (pretest, no stimulation), 5-min blue LED stimulation, 5-min free interaction (no stimulation), and 5-min blue LED stimulation. For triple-mice interactions, the three mice receive the following stimulations: 5-min baseline (pretest, no stimulation), 5-min LED stimulation (red, red, blue), 5-min free interaction (no stimulation), and 5-min LED stimulation (blue, blue, red). The tests are repeated three times, by alternating the order. Experiment groups express stGtACR2 + ChrimsonR, and control groups express EGFP + mCherry. The videos are analyzed by Etho Vision XT 15 and BORIS[86] social behaviors including sniff and attack are manually labeled and scored.

*Fiber photometry for dopamine (DA) detection.* Details for the fiber photometric system are described previously[81]. The excitation source is a 488 nm semiconductor laser (Coherent, Inc. OBIS 488 LS, tunable power up to 60 mW). A dichroic mirror with a 452–490 nm reflection band and a 505–800 nm transmission band (Thorlabs, Inc. MD498) is employed for wavelength selection. A multimode fiber (Thorlabs, Inc., 200 μm in diameter and 0.39 in numerical aperture) coupled to an objective lens (Olympus, ×10, NA 0.3) is used for optical transmission. The fluorescence signals are collected with a photomultiplier tube (PMT) (Hamamatsu, Inc. R3896) after filtering by a GFP bandpass filter (Thorlabs, MF525-39). An amplifier (C7319, Hamamatsu) is used to convert the current output from the PMT to voltage signals, which are further filtered through a lowpass filter (40 Hz cutoff; Brownlee 440, USA). The fluorescence signals are digitalized at 100 Hz and recorded by a Power 1401 digitizer and Spike2 software (CED, UK). Photometric data are exported and analyzed with MATLAB. The fluorescence signals ($\Delta F/F_0$) are processed by averaging the baseline signal $F_0$ over a 4.5-s long control time window and presented as heatmaps or per-event plots. AUC (area under the curve) of the DA signal is used for statistical analysis. AUC is the integral of a curve that describes the variation of DA signals during light stimulation.

**Statistics and reproducibility**. Immunostaining experiments are performed at least two times independently with similar results. The numbers of animals are provided as *n* value. No statistical methods are used to predetermine sample sizes, but our sample sizes are similar to those reported in the previous publications[87–89]. All samples are randomly assigned to different experimental groups. All data are presented as mean ± s.e.m. (the standard error of the mean) or as individual plots. Data distributions are assumed to be normal. For two-group comparisons, statistical significances are determined by two-tailed Student's *t* tests. For multiple-group comparisons, ANOVA tests are used. *$P < 0.05$, **$P < 0.01$, ***$P < 0.001$, ****$P < 0.0001$, n.s. (not significant) $P > 0.05$, for all statistical analyses presented in figures. Inferential statistical tests are carried out using GraphPad Prism (v7.0a), Origin 2020 or MATLAB (R2016a). Results of the statistical analyses are reported in Supplementary Table S1.

**Reporting summary**. Further information on research design is available in the Nature Research Reporting Summary linked to this article.

## Data availability
All data needed to evaluate the conclusions in the paper are provided in the main Article file, the Supplementary information file and the Source Data file.

## Code availability
The codes used in this study are available at https://github.com/shengxingstars/2022-dual-color-optogenetics.

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

## Acknowledgements

This work is supported by the National Natural Science Foundation of China (NSFC) (61874064, X.S.; 62005016, H.D.), Beijing Municipal Natural Science Foundation (4202032, X.S.), Tsinghua University Initiative Scientific Research Program (X.S.), Beijing National Research Center for Information Science and Technology (BNR2019ZS01005, X.S.). We thank Prof. Y. Li (Peking University) for support on GRAB$_{DA2m}$, Y. Zhou and Y. Ju (Chinese Institute for Brain Research, Beijing) for support on stGtACR2.

## Author contributions

X.S. and M.L. developed the concepts. L. Li, G.T., Y.Z., X.C., Z.S., H.D., C.L., Y.X., H.W., L.Y., and X.S. performed material, device and circuit design, fabrication, and characterization. L. Li, X.C., D.C., and X.S. performed the simulations. L. Li, L. Lu, Y.R., G.T., X.F., L.Y., M.L., and X.S. designed and performed biological experiments. L. Li, L. Lu, M.L., and X.S. wrote the paper in consultation with other authors.

## Competing interests

The authors declare no competing interests.
