## [Peer Review File · Nature Communications]

Reviewers' Comments:

Reviewer #1:

Remarks to the Author:

Optogenetics is an advanced neuromodulation technique that allows highly precise cell-type specific control of target neurons, opening new opportunities for basic neuroscience research and treatment of brain disorders and neurodegenerative diseases. Wireless and implantable optogenetic devices represent an important technological trend in this field to overcome critical limitations associated with the fiber-based tethered method. This manuscript reports a wireless optogenetic probe with colocalized dual-color microLEDs, which enable bidirectional optogenetic manipulation (which authors mean deterministic activation/inhibition of target neurons). Authors thoroughly and comprehensively explored the concept, fabrication strategies, proof-of-demonstration in vivo experiments to demonstrate unique capability of the proposed device. Overall, the technology is well developed and the application experiments demonstrate solid results, which show potentials for advanced in vivo optogenetics. The manuscript is timely and impactful. I would recommend publication, if authors can address/clarify the following questions:

1. Dual color optogenetic probe was developed previously by incorporating optical waveguides (Kampasi, et al., *Microsystems & Nanoengineering* 4:10 (2018)). The previous device can also provide colocalized delivery of light with two distinct wavelengths. Can authors clarify the novelty of the proposed design compared to the previous work?
2. Authors claimed a probe with laterally arranged red and blue micro-LEDs shows misaligned emissions and can severely limit its capability for bidirectional modulations (Fig 1g with iso-intensity lines indicating 10% of the max power). Although laterally arranged LEDs do not provide perfectly matched stimulation, light propagates in brain tissue (as shown in Fig 2g), therefore it can still make colocalized optogenetic stimulation as far as light power surpasses the minimum required optical power for optogenetic actuators. According to the data shown in Fig. S2, even with 10% or smaller percentage of the max power, optogenetic stimulation may be possible, indicating two laterally placed LEDs can still make a meaningful overlapping volume with light with two different colors. Please comment on this.
3. On page 3, authors introduced state-of-the-art wirelessly operated systems by citing ref. 24-26. These are based on different wireless schemes (RF, NFC). There are other recent work based on Bluetooth (Kim, et al., *Nature Communications* 12:535 (2021)) and infrared (Hashimoto, et al., *Neurophotonics* 1(1), 011002 (2014)). I'd like to suggest authors to cite these technologies as well to give readers information for various wireless technologies.
4. The probe was fabricated on a flexible substrate. How flexible is the probe? (bending stiffness?) Did authors use some kind of an assist needle to implant the probe into the brain? If so, please add some explanation about the surgical procedure related to this.
5. This is a minor comment. This manuscript doesn't have 'Methods' section. Authors may move supplementary contents to the Methods section of the main manuscript.

Reviewer #2:

Remarks to the Author:

Overview

In this manuscript, the authors report the commissioning of dual-color LED probes in behaving mice. Each device consists of a single dual-sided LED equipped with a wireless driver. Animals were injected with a combination of depolarizing and hyperpolarizing opsins, and functionality was assessed in brain slices. Each mouse was implanted with a single two-LED probe in the VTA and biasing of place preference was demonstrated. Finally, fiber photometry of accumbens dopamine was used to provide non-behavioral readout.

General

Dual-color probe technology holds great potential, but the technology already exists. Wireless

operation of optogenetic probes shows great promise, but this was previously established. Adding another color at nearly the same location allows bidirectional manipulation of the same neuronal population – yet this was also already shown. All experiments conducted in the MS could have been carried out using wired dual-color probes. This is due to the large size of the micro-LEDs and the fact that a single device was used in a single animal in all demonstrated applications.

Recommendation

Rejection (resubmission allowed) or major revision

Major comments

(1) Clarify the advantage relative to extra-cranial illumination. Since only a single diode of each color was used, it is unclear that an intra-cranial approach is required. Multiple recent developments have enabled extra-cranial activation of locally-injected opsins. Please suggest and demonstrate an application that cannot be realized using extra-cranial illumination.

(2) Application using simultaneous dual-color control. Figs. 3 and 4 show distinct effects of red vs. blue light, but it is not clear that any of the results require dual-color devices. Please suggest and demonstrate an application that cannot be realized using sequential single-color illumination.

(3) Damage to local circuitry. In cellular terms, the probe has huge dimensions: 320 x 120 micrometers, similar to the cross-section of 200 micrometer diameter fibers that have been used in some optogenetic studies. However, even from the authors own work (Figure S9) it is clear that the damage that such devices incur is enormous. Even 100 micrometer diameter fibers preclude neuronal recordings for at least 200 micrometers away from the tip (Kravitz et al., 2010, Nature). Thus, using a 200 micrometer fiber as a reference for comparison is inadequate. Furthermore, histological tools are insufficient, since local pressure may induce neuronal ischemia with no or minor structural changes. Please quantify the extent of damage functionally, for instance using ex-vivo recordings after implantation.

(4) Stimulation with arbitrary waveforms. The driving circuit seems to be a modification of a previously reported device (Supplementary information, reference 5). Does the circuitry support the use of non-square waveforms, e.g. sinusoids? If not, can an upgrade be designed and demonstrated?

(5) Independent channels. While it is clear that ChrimsonR responds to both red and blue light, and that stGtACR2 responds specifically to blue light, it is not clear that the combination of ChrimsonR and stGtACR2 is truly effective. Specifically, Fig. S18a and S18b show similar time courses for the combined protocols (green curves). Why is the fluorescence signal decreased when blue light is added to red light when only ChrimsonR is expressed? Regardless of the specific mechanism, this critical observation limits the interpretational power of some of the experiments.

(6) Combination with sensing electrodes. The choice of using fiber photometry from a remote region (Fig. 4) suggests that extracellular electrophysiology has either not been attempted or has been attempted and failed. If the former, please attempt; if the latter, please report the conclusions so device limitations would be crystal clear.

Minor points

(7) Background (lines 24-26) is incorrect. Previous work has shown activation and silencing of the same neuron in freely-moving rats about a decade ago, and more recently using mice (reference 33).

(8) The novelty of the present work with respect to the authors' own work should be clarified. The thin film technology has already been reported (references 34-36), and a single-color version of the micro-LED probe along with the control circuitry has already been reported (Supplementary Information, reference 5).

(9) The novelty of the present work with respect to the work of other authors should be clarified. The combination of optical stimulation with wireless control and power delivery has been reported much earlier than mentioned in the Introduction (Wentz et al., 2011, Journal of Neural

Engineering). Same-cell activation, silencing, and extracellular recordings in intact mice have been reported earlier than mentioned in the Introduction (Kampasi et al., 2016, Scientific Reports), and much earlier in freely-moving rats.

(10) The LEDs themselves are very large (125 x 180 micrometers) compared to a neuron. Is light emitted uniformly over the emitter, or are there "hot spots"? In particular, are any hot spots aligned between the red and blue LEDs?

(11) stGtACR2 can be activated with arbitrary waveforms, not only "pulsed" (lines 109-110).

(12) Electrophysiological assessment of dual-color control is incomplete. Figure 4c shows a dual-color protocol, but the electrophysiological experiments (Figure 2) only used one color at a time (Fig. 2e-k) or simultaneous application of two colors without a single-color internal control (Fig. 2l-m). Please show electrophysiological results for the exact protocol used in the intact mice (Fig. 4c).

(13) Clarify the advantage relative to extra-cranial light sources. One of the potential advantages of wireless devices is the capability to conduct experiments with multiple subjects simultaneously, as suggested by Movie S4. Can experiments be designed to that effect?

(14) The manuscript could benefit from English grammar editing. For instance, line 138 – "to wirelessly operate"; line 271, "which further confirms with the".

(15) Details about the optical model are missing (Supp. Lines 57-62).

(16) Differences between in vitro (electrophysiology, Fig. 2m) and in vivo (dopamine readout, Fig. 4g) should be minimized/discussed.

(17) "Animals" is a bit misleading, as only a mouse model was tested.

(18) Fig. 1d-1e – why is there a difference in the size of the blue and the red LEDs? This difference is not seen in fig 1c and is not mentioned in the given dimensions in line 94.

(19) Fig. 1g – it is not clear that the first three columns are the new presented technology. Is the last column another probe you tried? If so, I would like to see if this probe could not achieve all the results you have achieved with the presented technology, thus establishing the unique results the new probe offers.

(20) Fig. 1h – how does the chip connect to the probe? A photo would clarify this.

(21) Fig. 1i – very low resolution of the photos.

(22) Fig. 2d – it is written in the text (line 161) that "> 83% of cells co-express both opsins". I believe you meant "> 83% of the labeled cells". Is that the case? What are the precise percentages?

(23) Fig. 2 (and lines 176-179) – bidirectional activation in the same cell was shown in a previous paper (reference 33). This was done in a transgenic mouse injected with a mix of two viral vectors. Is the viral cocktail you used the novel procedure to be emphasized here? Calibration of the light intensity needed to use with these viral vectors? Otherwise, I am not sure how this figure is adding new information, especially when lasers are used, and not the probe your paper is about

(24) Fig. 3b – on line 200 it says that this panel illustrates the locations of the micro-LEDs in the VTA. Does this mean this is an illustration? I am not sure how were these photos created.

(25) Fig. S10 – what are "functional samples"? Are the five sample five mice? five micro-LEDs? five couples of probes (10 micro LEDs, five red and five blue)?

(26) Fig. 3f – I am not sure why the significance test was done once in reference to the pretest

and once in reference to the Red+. what is the difference between pretest and Blue +?

(27) Fig. 4g – what do the stars mean?

(28) Lines 213-215 - I suggest it would be added in the main text that the stimulation only occurred in the left chamber.

(29) Movie S3, S5 – does the driving chip support the multi-channel and multi-color modes?

(30) Line 88 – I suggest the word “enlarged” instead of “exploded”.

(31) Line 90 – I suggest to add “of” after “comprises”.

(32) Fig. 1f – I suggest to print the headlines in a different order: red LED and ChrimsonR one below the other on the right (above the corresponding curve), and blue LED and stGtACR2 on the left.

(33) Line 143 - I suggest “commands” instead of “commends”.

(34) Line 183 – I suggest “are suppressed” instead of “suppressions”.

(35) Fig. 2f, 2i, 2l – why were these values chosen?

(36) Fig. S8 – small DC response unnoticed.

(37) Line 203 – I suggest “assess” instead of “access”.

(38) Line 211 – I suggest “implantation” instead of “injection”.

(39) Fig. 3d – I suggest to fix shaded areas to match exactly 30 min in the graph.

(40) Line 251 – I suggest “Other” instead of “More”.

(41) Fig. 4h – I suggest to add a graphical legend so it would be easier to understand the panel without referring to the legend text.

(42) Supplementary line 171 – I suggest “finally” instead of “at last”.

(43) It seems that the response of the ChrimsonR in both the behavioral paradigm and the dopamine release experiment is a result of the action spectra presented in Fig. 1f, but this is not stated anywhere (the “optical crosstalk” possibility you mentioned was not directly connected to the last parts in the paper).

(44) Supplementary line 163 – what did you use for the 40 um slices?

Reviewer #3:

Remarks to the Author:

The manuscript reports on the application of dual-color optoelectronic probes in the VTA of freely behaving animals. The optrodes are used in place preference test controlling the μ LEDs using a smaller wireless headstage. The authors demonstrate by increasing or decreasing dopamine levels excited through colocalized red and blue stimulations rewarding and aversive behaviors of freely moving animals. The probes combine existing technologies to realize blue and red μ LEDs (cf. Refs 1&2 of the supplementary information) and – for the first time – their combination as a stacked μ LED sandwich with a filter placed in-between both μ LEDs. The μ LED probe is validated in vitro and in vivo with respect to electrical, optical, and thermal properties. The in vivo application targets dopaminergic neurons in the VTA – increasing and decreasing the dopamin level by optical stimuli at different wavelengths.

While the neuroscientific experiments are convincing, a couple of aspects with respect to the optoelectronic probe fabrication and characterization need to be clarified as further detailed.

Dual color optoelectronic probe fabrication

- o The μ LEDs are relatively large compared to the state of the art, cf. Yoon Lab at Univ. of Michigan, Ann Arbor, Mathieson Lab at Strathclyde Univ., Glasgow Scotland, and Schwarz Lab at Fraunhofer Institute IAF, Freiburg, Germany (Gossler et al.), Rogers Lab at Univ. of Illinois at Urbana-Champaign, USA: these probes are thinner (when realized on polymeric substrates provide an improved flexibility), provide smaller LEDs; the statement that the probes are flexible is a relative statement at a documented thickness of 120 μm – it has to be taken into account that the stiffness increases with the thickness to the power of 3.
- o It would be interesting to see how small the LEDs can be made; which smallest pitch can be achieved in case of multiple LED probes; how small the LED substrates can be made? How precisely can the μ LEDs be positioned on the polyimide substrate?
- o How are the μ LEDs/filter fixed on the polyimide substrate and on each other?
- o Are the 2 copper layers needed on the polyimide substrate? Aren't these layers increasing the probe stiffness too much? What about the biocompatibility of the applied copper? Can the authors guarantee that copper does not leak out?
- o How is the PDMS layer applied? By dip coating?
- o Have the lateral LED probes been applied in this study?
- o LED dimensions: A general question to be answered is on whether the larger LED size of 125x180 μm is requested to optically stimulate enough brain tissue to elicit behavioral changes; could the same effect be achieved with smaller LEDs as used for instance by the Yoon Lab?

* Thermal characterization

- o How have the authors calibrated the temperature measurements taking the emissivity of the LED materials into account? How do the measurements taken with the IR camera compare to the measurements using the thermocouples? This could be done in the tissue phantom. Are the extracted temperatures those of the LED surface or the surrounding tissue?
- o Are the copper layers on the polyimide substrate taken as a heat spreader?
- o In vitro tests with thermocouples: How do the thermocouples influence the temperature measurement as their wire represent a further path to conduct heat away from the LED?
- o Thermal modeling: which material parameters of the LED probe have been applied for the FE simulation of the temperature distribution?
- o In a recent paper by Zgierski-Johnston et al. (doi: 10.1016/j.pbiomolbio.2019.11.004), the temperature increase in cardiac tissue (presumably with similar thermal properties as brain tissue) was measured both for pulsed and continuous optical stimulation (in both cases 10 mA, 50% duty cycle at 10 kHz) using commercial CREE LEDs of comparable size. The extracted temperature increase was in the continuous operation mode up to 8 K after 1.67 sec. Please comment on the pronounced temperature increase extracted directly under the LEDs in comparison to your moderate temperature increase.

* Probe implantation

- o Do you need to remove the dura for optrode implantation? Is the stiffness high enough for a direct implantation through the dura without a shuttle as requested for thinner, polymeric recording probes (cf. Lan Luan et al.)

* Long-term optrode testing

- o How did the authors test the long-term performance of the LEDs and – in particular – the long-term stability of the PDMS / parylene C based encapsulation layer stack? Was the probe just exposed to the salt solution and tested from time to time or have the authors operated the LEDs continuously while exposing them to the electrolyte (or in vivo to brain tissue)?

* To be clarified:

- o Line 275-276: "The red illumination triggers the elevated dopamine release, while the use of blue light can effectively and promptly suppress the signal." From Figure 4h I conclude that red illumination results in an immediate response while the blue stimulus causes a slow reduction in DA concentration starting after 5s of blue light stimulation. The authors are asked to clarify.

* Typos (changes indicated by **)

o Line 42 – Change to: Understanding ** brain functions and treating neurological disorders rely on the continuous development of advanced technologies to interrogate with ** complex nervous systems.

o Line 222: “ ... More results are provided in Figures S12 and S13, comparing these animals with other group*s* expressing only ... ”

o Line 250-251: “Figure 4c plots temporally resolved dopamine release in respon*se* to stimulations caused by red and blue micro-LEDs in *the” same mouse.

* References

o Ref. 23 does not use fibers; the tool applies waveguides integrated on the probe shank; the authors are asked to be more specific

o The authors are further asked to compare their system with the state of the art (using transfer printing as well as wafer-level based bonding and laser-lift-off) in view of system dimensions, integration density (how many μ LEDs can realistically be integrated using their approach), mechanical stiffness, long-term stability

Reviewer #1

Optogenetics is an advanced neuromodulation technique that allows highly precise cell-type specific control of target neurons, opening new opportunities for basic neuroscience research and treatment of brain disorders and neurodegenerative diseases. Wireless and implantable optogenetic devices represent an important technological trend in this field to overcome critical limitations associated with the fiber-based tethered method. This manuscript reports a wireless optogenetic probe with colocalized dual-color microLEDs, which enable bidirectional optogenetic manipulation (which authors mean deterministic activation/inhibition of target neurons). Authors thoroughly and comprehensively explored the concept, fabrication strategies, proof-of-demonstration in vivo experiments to demonstrate unique capability of the proposed device. Overall, the technology is well developed and the application experiments demonstrate solid results, which show potentials for advanced in vivo optogenetics. The manuscript is timely and impactful. I would recommend publication, if authors can address/clarify the following questions:

Our response:

We thank the reviewer for these positive comments and valuable suggestions. The technical questions have been addressed in the following discussions.

Comment 1: “Dual color optogenetic probe was developed previously by incorporating optical waveguides (Kampasi, et al., *Microsystems & Nanoengineering* 4:10 (2018)). The previous device can also provide colocalized delivery of light with two distinct wavelengths. Can authors clarify the novelty of the proposed design compared to the previous work?”

Our response:

We agree that dual color optogenetic probe was developed previously by incorporating optical waveguides, as the reviewer mentioned in the reference by Kampasi, et al., *Microsystems & Nanoengineering* 4:10 (2018). Similar concepts have also been reported in Noked, et al., *IEEE Trans Biomed Eng* 68:416-427 (2021). Both systems can provide colocalized delivery of light with two distinct wavelengths. However, it is noted that these systems are still based on wired light sources that are connected to external control boards and power supplies. Wireless operation has not been realized for studies in freely moving animals.

Here we would like to articulate the technical and scientific advances in our work. Technically, for the first time, we report a wireless operated, dual-color optoelectronic probe for colocalized stimulations in freely moving animals. Scientifically, for the first time, we accomplish the simultaneous up-regulating and down-regulating dopamine levels, bidirectionally affecting behaviors of freely moving animals. These results have never been accomplished in prior works.

Our modification to the manuscript:

In Line 154, we add:

“...Compared to previous reports^{22, 36} on tethered fiber or laser coupled devices for dual-color control, the wireless operation enables the study of complex behaviors of freely moving animals more conveniently. ...”

Comment 2: “Authors claimed a probe with laterally arranged red and blue micro-LEDs shows misaligned emissions and can severely limit its capability for bidirectional modulations (Fig 1g with iso-intensity lines indicating 10% of the max power). Although laterally arranged LEDs do not provide perfectly matched stimulation, light propagates in brain tissue (as shown in Fig 2g), therefore it can still make colocalized optogenetic stimulation as far as light power surpasses the minimum required optical power for optogenetic actuators. According to the data shown in Fig. S2, even with 10% or smaller percentage of the max power, optogenetic stimulation may be possible, indicating two laterally placed LEDs can still make a meaningful overlapping volume with light with two different colors. Please comment on this.”

Our response:

We agree that a probe with laterally arranged red and blue micro-LEDs still shows overlapped dual-color emissions and can enable bidirectional optogenetic stimulations in certain applications. However, it is noted that the overlapping volume in such a design is much smaller than that in a vertically stacked micro-LED design (Fig. 1g). It means that more currents need to be injected into the LEDs to create a larger illuminating region, which inevitably increases tissue heating as well as the power consumption. By contrast, our unique vertical stacking design makes sure that the overall device has a smaller footprint and generates the optimal overlapping optical profile for dual-color stimulation.

Our modification to the manuscript:

In Line 138, we add:

“... Compared to a probe with laterally arranged red and blue micro-LEDs displaying highly misaligned emissions, the vertically assembled device provides a smaller footprint and generates the optimal overlapping profile for dual-color stimulation. ...”

Comment 3: “On page 3, authors introduced state-of-the-art wirelessly operated systems by citing ref. 24-26. These are based on different wireless schemes (RF, NFC). There are other recent work based on Bluetooth (Kim, et al., *Nature Communications* 12:535 (2021)) and infrared (Hashimoto, et al., *Neurophotonics* 1(1), 011002 (2014)). I’d like to suggest authors to cite these technologies as well to give readers information for various wireless technologies.”

Our response:

We thank the reviewer for pointing out these recent related publications. These works have been cited and discussed in the revised text.

Our modification to the manuscript:

In Line 54, Page 3, we modified:

“... when incorporated with various wirelessly operated systems based on radio frequency (RF) antennas^{23, 24, 25}, near-field communication (NFC)²⁶, Bluetooth chips²⁷, and infrared receivers²⁸. ...”

In reference list, we add these papers:

27. Kim CY, Ku MJ, Qazi R, Nam HJ, Park JW, Nam KS, et al. Soft subdermal implant capable of wireless battery charging and programmable controls for applications in optogenetics. *Nat. Commun.* 2021, 12(1): 535.

28. Hashimoto M, Hata A, Miyata T, Hirase H. Programmable wireless light-emitting diode stimulator for chronic stimulation of optogenetic molecules in freely moving mice. *Neurophotonics* 2014, 1: 011002.

Comment 4: “The probe was fabricated on a flexible substrate. How flexible is the probe? (bending stiffness?) Did authors use some kind of an assist needle to implant the probe into the brain? If so, please add some explanation about the surgical procedure related to this.”

Our response:

The thin-film probe has a measured Young’s modulus of ~15 GPa and a bending stiffness of $\sim 9 \times 10^4$ pNm². It is softer than silicon (~180 GPa) and tungsten (~400 GPa), but still much harder than the brain tissue (~1 kPa). The probe’s critical buckling load is ~35 mN, which is much larger than the load (~90 μN) required to puncture the brain. Therefore, no other supporting needles are needed to assist the implantation.

The probe is implanted into the mouse brain via standard stereotaxic surgery, similar to the process for fiber implantation for optogenetics. A figure is added into the supporting document to illustrate the surgical procedure.

Our modification to the manuscript:

In Line 114, we add:

“...The Cu coated PI substrate has a measured Young’s modulus of ~15 GPa, softer than silicon (~180 GPa) and tungsten (~400 GPa), but still much harder than the brain tissue (~1 kPa). The formed probe has a bending stiffness of $\sim 9 \times 10^4$ pNm², similar to metal electrodes used for electrophysiological studies⁴⁵. ...”

We add a new reference to compare the mechanical properties of probes made by different materials:

45. He F, Lycke R, Ganji M, Xie C, Luan L. Ultraflexible neural electrodes for long-lasting intracortical recording. *iScience* 2020, 23(8): 101387.

In Line 233, we modify:

“...the dual-color micro-LED probe is implanted into the VTA of *DAT-cre* mice via standard stereotaxic surgery (Figure 3a and Figure S12) ...”

In the supplement, we add a new figure S12, to illustrate the surgical procedure.

Figure S12

Figure S12. Photographs of the surgical procedure for probe implantation. A hole is created on the exposed skull of a mouse by drilling, and the dura is carefully removed by needle. The micro-LED probe is fixed on a holder controlled by a stereotaxic instrument, and slowly inserted into the targeted region.

Comment 5: *“This is a minor comment. This manuscript doesn’t have ‘Methods’ section. Authors may move supplementary contents to the Methods section of the main manuscript.”*

Our response:

We thank the reviewer for this suggestion. The manuscript has been revised accordingly and includes a section of Methods.

Reviewer #2

Overview

In this manuscript, the authors report the commissioning of dual-color LED probes in behaving mice. Each device consists of a single dual-sided LED equipped with a wireless driver. Animals were injected with a combination of depolarizing and hyperpolarizing opsins, and functionality was assessed in brain slices. Each mouse was implanted with a single two-LED probe in the VTA and biasing of place preference was demonstrated. Finally, fiber photometry of accumbens dopamine was used to provide non-behavioral readout.

General

Dual-color probe technology holds great potential, but the technology already exists. Wireless operation of optogenetic probes shows great promise, but this was previously established. Adding another color at nearly the same location allows bidirectional manipulation of the same neuronal population – yet this was also already shown. All experiments conducted in the MS could have been carried out using wired dual-color probes. This is due to the large size of the micro-LEDs and the fact that a single device was used in a single animal in all demonstrated applications.

Recommendation

Rejection (resubmission allowed) or major revision

Our response:

We thank the reviewer for the insightful comments and the recommendation for revision and resubmission.

We agree with the reviewer that dual color optogenetic probe was developed previously by incorporating optical waveguides, for example, in Kampasi, et al., *Microsystems & Nanoengineering* 4:10 (2018), and Noked, et al., *IEEE Trans Biomed Eng* 68:416-427 (2021). We have cited and highlighted these works in our manuscript.

Here we would like to articulate the technical and scientific advances in our work. Technically, for the first time, we report a wireless operated, dual-color optoelectronic probe for colocalized stimulations in freely moving animals. Scientifically, for the first time, we accomplish the simultaneous up-regulating and down-regulating dopamine levels, bidirectionally affecting behaviors of freely moving animals. These results have not been accomplished in prior works.

We also agree with the reviewer that it is possible to conduct some related experiments using a wired dual-color system in a single animal. Moreover, the reviewers suggest additional applications to further demonstrate the uniqueness and irreplaceable advantages of our technology. These are excellent suggestions. In fact, we have performed additional experiments to investigate interactive social activities among multiple animals, under the influence of bidirectional optogenetic modulations. We believe these results further strengthen the novelty of our work. These results have been included in the revised manuscript and described in the following discussions.

Major Comments

Comment 1: “Clarify the advantage relative to extra-cranial illumination. Since only a single diode of each color was used, it is unclear that an intra-cranial approach is required. Multiple recent developments have enabled extra-cranial activation of locally-injected opsins. Please suggest and demonstrate an application that cannot be realized using extra-cranial illumination.”

Our response:

We agree with the reviewer that it is possible to conduct optogenetic stimulation with laser diodes coupled waveguides or fibers via extra-cranial illumination. Remarkable results have been reported using this strategy, for example, in Kampasi, et al., *Microsystems & Nanoengineering* 4:10 (2018), and Noked, et al., *IEEE Trans Biomed Eng* 68:416-427 (2021). In these works, the dual-color systems are combined with electrophysiological recordings and demonstrate their utilities for bidirectional stimulations. We have cited and highlighted these works in our manuscript.

It should be noted that these systems are based on commercial laser or LED chips that have a relatively large footprint, consume more power and require tethered systems for energy supply. By using our micro-LEDs integrated on the implantable probe, the entire system can be further shrunk. One unique feature of our approach is the wireless operation with a miniaturized circuit module, which makes behavioral studies (e.g., place preference test) more convenient.

Another potential application with our wireless systems is to study interactive social activities among multiple animals. We have performed additional experiments and explore the social activities among two and three mice, and demonstrated bidirectional modulations of their social preference. These experiments have not been reported before and will be difficult to perform using a single-color emitter or a wired stimulation system.

Our modification to the manuscript:

In Line 63, Page 3, we add:

“... There are also reports on dual-color optogenetic activation and inhibition by co-expressing different opsins in the same cells^{33, 34, 35, 36}. Recent efforts also remarkably demonstrate bidirectional optogenetic modulations with implantable waveguides coupled to extra-cranial dual-color laser sources and their combination with *in vivo* electrophysiological recordings^{23, 36, 37}, but wireless operation has not been achieved. ...”

In reference list, we add these papers:

35. Kampasi K, Stark E, Seymour J, Na K, Winful HG, Buzsáki G, et al. Fiberless multicolor neural optoelectrode for in vivo circuit analysis. *Sci. Rep.* 2016, 6(1): 1-13.

In Line 276, we add a new section:

“

Bidirectional optogenetic modulation of social interactions among multiple mice, with wireless dual-color micro-LED probes

Modulating dopaminergic neurons in the VTA region also influences the social preference, in particular, affective and aggressive behaviors^{49, 50}. Compared with tethered stimulation systems, our wirelessly operated micro-LED probes with independently addressable illuminations present prominent advantages in studying social interactions among multiple objects (Movie S4). Additionally, the colocalized, dual-color emission capability allows the bidirectional regulation of social activities in real time, which is difficult to access with single color stimulations^{49, 51}. In paired-object interactions (Figure 4a), two mice co-expressing ChrimsonR and stGtACR2 in the VTA are placed in an open-field arena, and synchronized red or blue stimulations are provided for 5 mins each. Mice behaviors are video recorded throughout the test period, with social activities (including sniffing and attacking) labeled and scored (Figure 4b). The total time spent in these activities are summarized and compared among different scenarios (red light on, blue light on, and resting) in Figure 4c. In comparison to the resting state, red or blue LED stimulations strongly promote or suppress social interactions between two objects in the experiment group, while no significant effects are observed in the control group (expressing EGFP and mCherry).

The wireless micro-LED probes can further be applied to interrogate more complicated interaction behaviors among multiple (> 2) objects. In Figure 4d, we place three mice in the same social chamber, and provide these animals with alternating red or blue LED stimulations. In such a testing paradigm, two mice receive red stimulations while the third one receives blue stimulation, then vice versa (Figure 4e). The mice with red stimulations spend more time in interacting with each other compared to the individual under blue illuminations, and the social preference can be reversed by alternating the stimulation patterns among these three mice (Figure 4f). By contrast, summarized data among control groups present no significant difference under red or blue stimulations. Collectively, these social interaction explorations clearly demonstrate the unique advantage of our wireless and bidirectional LED stimulators in the interrogation of complex animal behaviors.

”

In reference list, we add these papers:

50. Yu Q, Teixeira CM, Mahadevia D, Huang Y, Balsam D, Mann JJ, *et al.* Dopamine and serotonin signaling during two sensitive developmental periods differentially impact adult aggressive and affective behaviors in mice. *Nat. Neurosci.* 2014, **19**(6): 688-698.

51. Yang Y, Wu M, Vázquez-Guardado A, Wegener AJ, Grajales-Reyes JG, Deng Y, *et al.* Wireless multilateral devices for optogenetic studies of individual and social behaviors. *Nat. Neurosci.* 2021, **24**(7): 1035-1045.

In Line 672, we add associated description of the method:

“

Social interaction test

After 2 weeks of recovery and AAV expression, social interaction tests are performed with *DAT-cre* mice. Two or three mice are placed in an open-field arena ($30 \times 30 \text{ cm}^2$), wireless dual-color micro-LED probes provide red (20 Hz, 20-ms pulse, 7 mA) and blue (continuous, 5 mA) stimulations, and their behaviors are recorded by a camera. In paired-mice experiments, the two mice receive the same 3-phase stimulations: 1) 5-min baseline (pretest, no stimulation), 5-min red LED stimulation, 5-min free interaction (no stimulation), and 5-min red LED stimulation; 2) 2-hour rest in individual housing; 3) 5-min baseline (pretest, no stimulation), 5-min blue LED stimulation, 5-min free interaction (no stimulation), and 5-min blue LED stimulation. For triple-mice interactions, the three mice receive the following stimulations: 5-min baseline (pretest, no stimulation), 5-min LED stimulation (red, red, blue), 5-min free interaction (no stimulation), and 5-min LED stimulation (blue, blue, red). The tests are repeated for three times, by alternating the order. Experiment groups express stGtACR2 + ChrimsonR, and control groups express EGFP + mCherry. The videos are analyzed by Etho Vision XT 15 and social behaviors including sniff and attack are manually labeled and scored.

”

We add a new Figure 4, including the results of bidirectional modulations of social interactive activities.

Figure 4

Figure 4. Bidirectional, *in vivo* optogenetic control of mice social interactions with wirelessly operated, dual-color micro-LED probes. **a**, Photograph of two *DAT-cre* mice with wireless probes in an arena (30 × 30 cm²). Experiment groups (Exp) express stGtACR2 + ChrimsonR, and control groups (Ctrl) express EGFP + mCherry. **b**, Representative behavioral sequences recorded for one mouse receiving red or blue stimulations during social interactions. Here sniffing and attacking behaviors are highlighted and labeled, and other behaviors include locomotion, resting and escaping. **c**, Summary of changes in social interactions under red or blue stimulations. Here we compare the time spent in sniff and attack during the 5 mins light stimulation with the 5 mins resting state. $n = 4$ groups for both Exp and Ctrl. **d**, Photograph of three *DAT-cre* mice with wireless probes in an arena. **e**, Representative behavioral sequences recorded for the three mouse (#A, #B, #C) receiving red or blue stimulation during social interactions. **f**, Summary of changes in social interactions under red or blue stimulations for each mouse. $n = 4$ groups for both Exp (left) and Ctrl (right). All data are represented as mean ± s.e.m.. Student's *t* test, * $P < 0.1$, ** $P < 0.01$, *** $P < 0.001$, **** $P < 0.0001$, n.s. $P > 0.1$.

Comment 2: “Application using simultaneous dual-color control. Figs. 3 and 4 show distinct effects of red vs. blue light, but it is not clear that any of the results require dual-color devices. Please suggest and demonstrate an application that cannot be realized using sequential single-color illumination.”

Our response:

We agree with the reviewer that some results in Figs. 3 and 4 in our original manuscript can be conducted in separate animals expressing one opsin (ChrimsonR or stGtACR2). The advantages of using dual-color control include: 1) Excitation and inhibition effects are demonstrated in the same animal, eliminating uncertainties and variations when using two different animals; 2) the capability of up-regulating and down-regulating neurotransmitters (e.g., dopamine) in the same animal controls neural activities in a more precise manner; 3) the applications of two opsins and dual-color probes reduce the animal usage and make the experiments more ethical.

In addition, we further expand the experiments in Fig. 3. By applying both red and blue stimulations in the left and right chambers respectively, the mice’s preference indices are further enhanced, compared to the animals under sequential single-color illumination. This is ascribed to the fact that red illumination induces preference response in the left chamber, and blue illumination induces aversion response in the right chamber, in the same experiment. These new results have been included in Fig. S18. These results further demonstrate the efficacy of bidirectional modulations, which cannot be accomplished with single-color illumination.

Our modification to the manuscript:

In Line 259, we add:

“... Additionally, the application of both red and blue stimulations in opposite chambers further enhances the preference indices of mice compared to those only experiencing single color stimulations, demonstrating the efficacy of bidirectional modulation (Figure S18). ...”

We add Fig. S18.

Figure S18

Figure S18. (a) Patterns used for optogenetic modulations, including a 10-min pretest, a 30-min red LED stimulation (20 Hz, 10-ms pulse, current 7 mA) in the left chamber and blue LED stimulation (continuous, current 5 mA) in the right chamber. (b) Representative heat maps comparing pretest and real-time preference behavior following both red (left chamber) and blue (right chamber) stimulation for mice expressing stGtACR2 + ChrimsonR. (c) Preference indices measured at different times for mice under only red stimulations ($n = 4$ mice), or red stimulations in the left chamber and blue stimulations in the right chamber ($n = 3$ mice), and the black line show the result of control groups which only express EGFP and mcherry ($n = 3$ mice). (d) Summary of preference indices (the ratio of the time that mice spend in the left chamber to the whole recorded time) for mice under only red stimulations ($n = 4$ mice), or red stimulations in the left chamber and blue stimulations in the right chamber ($n = 3$ mice). Student's t test, ** $P < 0.01$.

Comment 3: “*Damage to local circuitry. In cellular terms, the probe has huge dimensions: 320 x 120 micro-meters, similar to the cross-section of 200 micro-meter diameter fibers that have been used in some optogenetic studies. However, even from the authors own work (Figure S9) it is clear that the damage that such devices incur is enormous. Even 100 micro-meter diameter fibers preclude neuronal recordings for at least 200 micro-meters away from the tip (Kravitz et al., 2010, Nature). Thus, using a 200 micro-meter fiber as a reference for comparison is inadequate. Furthermore, histological tools are insufficient, since local pressure may induce neuronal ischemia with no or minor structural changes. Please quantify the extent of damage functionally, for instance using ex-vivo recordings after implantation.*”

Our response:

We agree with the reviewer that all kinds of implantable probes, either our micro-LED probes or conventional glass fibers, inevitably create damage to local circuitry in the brain. We apply standard histological analysis to evaluate the biocompatibility of our probe, and indicate that the micro-LED probe and the fiber have similar inflammatory responses occurring after implantation. Despite the tissue lesion, the implantable probe is still functional, and effectively activates and inhibits neural activities at both behavioral (place preference) and circuitry (VTA to NAc) levels, as demonstrated in Figs. 3, 4 and 5.

We also agree that functionally evaluating the living tissues after implantation will be very beneficial. Therefore, we perform additional experiments to record *ex vivo* cell activities by taking calcium fluorescence images in acute brain slices 14 days after probe implantation. As the reviewer indicates, there are no active neuronal activities observed surrounding the lesion area, in agreement with the histological analysis. For cells more than 100 μm away from the tip, calcium dynamics can be effectively recorded.

Our modification to the manuscript:

In Line 243, we add:

“... In addition, we perform dynamic calcium imaging on acute brain slices for AAV-hsyn-GCaMP6m expressing mice 2 weeks post implantation (Figure S15). Cell activities can be effectively recorded in neurons $\sim 100 \mu\text{m}$ away from the lesion region, in agreement with the histological analysis. ...”

In Line 649, the Methods section, we add:

“

Calcium imaging in acute brain slice

Mice are injected with 0.4 μL of Raav2/9-hsyn-GCaMP6m into the S1. After 2 weeks of recovery, acute brain slices are prepared. Ca^{2+} fluorescent signals in GCaMP6m expressing cells are captured with a 20 \times water immersion objective on a confocal microscope (FV1000, Olympus). For cell activation, 1 μM α -amino-3-hydroxy-5-methyl-4-isoxazolepropionic acid (AMPA, No.A6816, Sigma-Aldrich) is added in ACSF to enhance the Ca^{2+} signals. Control experiments are performed by applying pure ACSF. Fluorescence images are analyzed via ImageJ. Normalized fluorescence changes are calculated as $dF/F = (F - F_0) / F_0$, where F_0 is the baseline intensity (before 10 s). ...”

We add Fig. S15:

Figure S15

Figure S15. Imaging cellular calcium dynamics in acute brain slice. (a) Representative fluorescence image showing GCaMP6m-expressing cells near the probe region after 14 days implantation. Red circles mark active cells followed by AMPA administration. (b) Averaged calcium signal traces (dF/F_0) for all marked neurons after applying AMPA (red line) or pure ACSF (grey line) at frame 11, the sampling rate is 1.1 s / frame. (c, d) Heatmaps showing fluorescence variations in all 21 neurons after (c) applying AMPA or (d) pure ACSF.

A movie clip can be seen here:

<https://youtu.be/wt8dU30Fdy0>

Comment 4: “Stimulation with arbitrary waveforms. The driving circuit seems to be a modification of a previously reported device (Supplementary information, reference 5). Does the circuitry support the use of non-square waveforms, e.g. sinusoids? If not, can an upgrade be designed and demonstrated?”

Our response:

The reviewer is correct. The wireless circuit is modified based on our previously reported work (Y. Zhao, et al, IEEE Transactions on Electron Devices 2019, 66(1): 785-792.). The blue and red micro-LEDs are independently controlled by two separate LED drivers ZLED7012, which provides programmable constant current levels ranging from 1.8 mA to 20 mA. By programming the circuit, the system also supports non-square waveforms, like sinusoids or triangular waves. See the photographs below:

Sinusoidal wave (1 Hz, amplitude 10 mA)

Triangular wave (1 Hz, amplitude 10 mA)

Movie clips can be seen here:

<https://youtu.be/tn3iSaHQwW8>

<https://youtu.be/u1q5tS78NVE>

Our modification to the manuscript:

In Line 515, we add:

“... The driver provides programmable constant current levels ranging from 1.8 mA to 20 mA and also supports arbitrary waveforms. ...”

Comment 5: “Independent channels. While it is clear that ChrimsonR responds to both red and blue light, and that stGtACR2 responds specifically to blue light, it is not clear that the combination of ChrimsonR and stGtACR2 is truly effective. Specifically, Fig. S18a and S18b show similar time courses for the combined protocols (green curves). Why is the fluorescence signal decreased when blue light is added to red light when only ChrimsonR is expressed? Regardless of the specific mechanism, this critical observation limits the interpretational power of some of the experiments.”

Our response:

Our paper reports:

- (1) At the cellular level, neurons co-expressing ChrimsonR and stGtACR2 can be deterministically evoked or silenced under red or blue irradiations;
- (2) At the circuitry level, colocalized red or blue stimulations up-regulate or down-regulate dopamine levels in the VTA-to-NAc projection in mice co-expressing ChrimsonR and stGtACR2;
- (3) At the behavioral level, red or blue stimulations generate rewarding or aversive behaviors of freely moving animals co-expressing ChrimsonR and stGtACR2 in a place preference test;
- (4) At the social level, red or blue stimulations modulate aggressive and affective behaviors among multiple mice co-expressing ChrimsonR and stGtACR2 in a social interactive test.

These collective results clearly demonstrate the efficacy of bidirectional stimulations combining with ChrimsonR and stGtACR2 co-expression.

We thank the reviewer for carefully reading our paper and pointing out the interesting result in Fig. S18b (now Fig. S25b) (green curve), which shows that “the fluorescence signal decreased when blue light is added to red light when only ChrimsonR is expressed”. It should be noted that these data plots the averaged signal traces among multiple mice ($n = 3$). If we check the raw data in Fig. 5k, we can see that different mice (with only ChrimsonR) show varied responses upon red + blue illumination, and their trends are not as significant as other cases.

We acknowledge the fact that applying both red and blue illuminations simultaneously may generate various responses, depending on the optical irradiance, time duration, viral load, etc. Our results only explore a small fraction of such effects. We agree with the reviewer that the bidirectional optogenetic modulations with combined opsins definitely worth further investigation. Some explorations, we believe, are beyond the scope of the current manuscript.

Our modification to the manuscript:

In Line 411, we add:

“... It should be also noted that these dual-color or multi-color modulations should be carefully designed, and their efficacies are highly dependent on the spectral overlap among opsins, irradiance of different colors, levels of viral expressions, etc. ...”

Comment 6: “Combination with sensing electrodes. The choice of using fiber photometry from a remote region (Fig. 4) suggests that extracellular electrophysiology has either not been attempted or has been attempted and failed. If the former, please attempt; if the latter, please report the conclusions so device limitations would be crystal clear.”

Our response:

The reviewer’s suggestion is excellent. We agree that combining optogenetic stimulations with simultaneous electrophysiological sensing capability is very useful for neural interrogation, as demonstrated in previous works, for example, in Kampasi, et al., *Microsystems & Nanoengineering* 4:10 (2018), and Noked, et al., *IEEE Trans Biomed Eng* 68:416-427 (2021).

To evaluate the utility of bidirectional modulations, we attempt extracellular electrophysiological recordings in the cortex (Fig. S11). We implant our dual-color micro-LED probe together with metallic sensing electrodes, and perform optical stimulations and electrical recording in head-fixed mice expressing ChrimsonR and stGtACR2. Recording results clearly show enhanced or suppressed cell activities during red or blue illuminations, respectively.

Here the recordings are obtained with home-made metallic wires. We admit that simultaneous optical stimulations and electrical recordings in the deep brain region like VTA will be more challenging and require more sophisticated probe design and fabrication. An outstanding example is the integration of patterned micro-LEDs and electrodes on silicon probes (Wu, et al., *Neuron*, 88, 1136 (2015)).

To alternatively monitor neural activities in the deep brain, we record dopamine release in the NAc with fluorescence photometry, thereby demonstrating all-optical based modulations and sensing (Fig. 5). This method also owns several advantages over direct electrophysiological recordings in the VTA: 1) it reduces damage in the VTA caused by inserting additional electrodes; 2) it avoids photoelectric artifacts within the electrodes; 3) it further verifies the efficacy of bidirectional modulations in the VTA-to-NAc circuitry level.

Our modification to the manuscript:

In Line 211, we add a new section:

“

Bidirectional optogenetic modulation and simultaneous electrophysiological recording *in vivo*

Neural activities associated with dual-color stimulations are further examined with simultaneous electrophysiological recordings *in vivo* (Figure S11). Two opsins (ChrimsonR and stGtACR2) are co-expressed in the primary somatosensory cortex of *CaMKII-Cre* mice (Figure S11a). Followed by two weeks of recovery and viral expression, extracellular electrodes and dual-color micro-LED probes are inserted in the head-fixed mice, with a separation of ~500 μm (Figure S11b). Spontaneous and red-light stimulated waveforms are recorded and sorted, showing typical extracellular spike events (Figure S11c). Multiple trials collected for a sample unit indicate that cell spike rates can be enhanced or decreased by applying red or blue illuminations (Figures S11d and S11e). Results summarized for multiple cells show that the basal

spike firing rate (13 Hz) increases to 28 Hz during red illumination and decreases to 2 Hz during blue illumination (Figures S11f and S11g). These results showcase the capability for *in vivo* bidirectional modulations and are in good accordance with the previous report based on a different opsin combination (ChR2 and Jaws)³⁶.

In Line 390, we add:

“... While arrays of blue InGaN micro-LEDs can be directly grown on silicon and integrated with recording electrodes⁴¹, the combination simultaneous dual-color stimulations and electrophysiological recordings in the deep brain region like VTA require more sophisticated device design and fabrication and have not been attempted here. ...”

In reference list, we add the paper:

41. Wu F, Stark E, Ku P-C, Wise Kensall D, Buzsáki G, Yoon E. Monolithically Integrated μ LEDs on Silicon Neural Probes for High-Resolution Optogenetic Studies in Behaving Animals. *Neuron* 2015, **88**(6): 1136-1148.

In Line 625, we add:

“

In vivo extracellular electrophysiology recordings

We perform the extracellular single unit recording as our previously described procedure⁸¹. Adult CamkIIa-Cre mice are anesthetized with an intraperitoneal injection of Avertin (250 mg/Kg). A bolus of 1 μ L of virus (AAV2/9-CAG-DIO-ChrimsonR-tdTomato and AAV2/9-EF1a-DIO-stGtACR2-EGFP-Kv2.1, 1:1) is injected into neocortex and hippocampus at different depths (AP -1.6 mm, ML 1.1 mm, DV 0.4 to 2.0 mm increments; 200 nL/site). A silver wire (127 μ m diameter, A-M system) is attached to a skull-penetrating M1 screw above olfactory bulb to serve as the ground reference. For head-fixed preparations, a custom-made titanium head-plate is secured to the skull with dental acrylic. Recordings are performed 2 weeks after viral injection. Mice are held in place with a head bar cemented to the skull. The dual-color micro-LED probe is lowered into the neocortex with a 15° angle from medial to lateral and another 10° angle from anterior to posterior. 8 stereotrodes with Pt-Ir wires (diameter 17.5 μ m, platinum 10% iridium, California Fine Wire, USA) are used for extracellular electrophysiology recordings. The distance between the stereotrodes and the micro-LEDs is \sim 500 μ m. To reduce the noise induced by the stray capacitance coupling from LEDs to electrodes, we use a tinfoil shield to encompass the electric circuit of the LED. Voltage signals are digitized and recorded by an Open-Ephys board (<http://www.open-ephys.org/>). Signals from each recording electrode are band-pass-filtered between 300 and 6000 Hz and sampled at 25 kHz. Single units are discriminated with principle component analysis (Offline sorter, spike2, CED, UK). Recorded units smaller than 2.5 times the noise band or in which interspike intervals are shorter than 2 ms are excluded from further analysis. Red light (pulse width 0.2 s, interval 3 s, 50–100 pulses, LED current 10 mA) and blue light (pulse width 1 s, interval 3 s, 50–100 pulses, LED current 1 mA) are used to activate or inhibit neurons. Data are analyzed and plot with a custom-developed MATLAB program.

”

We add a new Fig. S11.

Figure S11

Figure 11. Bidirectional, *in vivo* optogenetic modulation of neural activities with dual-color illuminations in the cortex, combining with electrophysiological recordings. (a) Schematic strategy for co-expressing ChrimsonR and stGtACR2 in the primary somatosensory cortex of *CamkIIa-Cre* mice. (b) Illustration of the setup for simultaneous optogenetic stimulation and electrophysiological recordings by implanting the micro-LED probe and metal electrodes into the cortex of head-fixed mice. (c) Waveforms of a single unit recorded during baseline period before the red LED illumination (left) and during the red LED illumination period (right). Correlation coefficient between the two sets of waveforms is 0.9965. (d, e) Raster plots (top) and peri-stimulus time histogram (PSTH) plots (bottom) recorded for an example unit ($n = 72$ trials) for a sample unit during (d) red illumination (pulse width 0.2 s, LED current 10 mA) and (e) blue illumination (pulse width 1 s, LED current 1 mA). (f, g) Summarized results (top: heatmaps; bottom: PSTH plots) for multiple cells collected during (f) red illumination and (g) blue illumination ($n = 13$ cells from 2 mice, 50–100 trials for each cell).

Minor points

Comment 7: “Background (lines 24-26) is incorrect. Previous work has shown activation and silencing of the same neuron in freely-moving rats about a decade ago, and more recently using mice (reference 33).”

Our response:

We thank the reviewer for the correction. The background parts in the abstract and the introduction have been revised accordingly.

Our modification to the manuscript:

In Line 24, we modify the sentence to:

“... the ability of simultaneous neural activation and inhibition in a same brain region of freely moving animals is highly desirable and being actively researched. ...”

In Line 64, we add:

“... Recent efforts also remarkably demonstrate bidirectional optogenetic modulations with implantable waveguides coupled to extra-cranial dual-color laser sources and their combination with *in vivo* electrophysiological recordings^{23, 36, 37}, but wireless operation has not achieved. ...”

Comment 8: “*The novelty of the present work with respect to the authors’ own work should be clarified. the thin film technology has already been reported (references 34-36), and a single-color version of the micro-LED probe along with the control circuitry has already been reported (Supplementary Information, reference 5).*”

Our response:

Here we would like to articulate the technical and scientific advances in our work. Technically, for the first time, we report a wireless operated, dual-color optoelectronic probe for colocalized stimulations in freely moving animals. Scientifically, for the first time, we accomplish the simultaneous up-regulating and down-regulating dopamine levels, bidirectionally affecting behaviors of freely moving animals. These efforts have been based our expertise in thin-film, microscale device design and fabrication, but not been accomplished in prior works.

Our modification to the manuscript:

In Line 90, we add:

“...Our previous works establish wirelessly operated microscale optoelectronic device strategy for *in vivo*, single-color optogenetic stimulations^{37, 38, 39, 40}, here these concepts are applied for dual-color, bidirectional modulations. ...”

In Line 521, we add:

“... Different from our previous design⁸², here a polyimide based flexible circuit board is implemented for reduced weight. ...”

Comment 9: “The novelty of the present work with respect to the work of other authors should be clarified. The combination of optical stimulation with wireless control and power delivery has been reported much earlier than mentioned in the Introduction (Wentz et al., 2011, *Journal of Neural Engineering*). Same-cell activation, silencing, and extracellular recordings in intact mice have been reported earlier than mentioned in the Introduction (Kampasi et al., 2016, *Scientific Reports*), and much earlier in freely-moving rats.”

Our response:

We thank the reviewer for referring these earlier pioneering works to us. These papers have been updated in the reference list.

Here we would like to articulate the technical and scientific advances in our work. Technically, for the first time, we report a wireless operated, dual-color optoelectronic probe for colocalized stimulations in freely moving animals. Scientifically, for the first time, we accomplish the simultaneous up-regulating and down-regulating dopamine levels, bidirectionally affecting behaviors of freely moving animals. These achievements have not been accomplished in prior works.

Our modification to the manuscript:

In Line 63, we modify the sentences:

“... There are also reports on dual-color optogenetic activation and inhibition by co-expressing different opsins in the same cells^{33, 34, 35, 36}. Recent efforts also remarkably demonstrate bidirectional optogenetic modulations with implantable waveguides coupled to extra-cranial dual-color laser sources and their combination with *in vivo* electrophysiological recordings^{23, 36, 37}, but wireless operation has not been achieved....”

In the references we add:

25. Wentz CT, Bernstein JG, Monahan P, Guerra A, Rodriguez A, Boyden ES. A wirelessly powered and controlled device for optical neural control of freely-behaving animals. *J. Neural. Eng.* 2011, 8(4): 046021.
35. Kampasi K, Stark E, Seymour J, Na K, Winful HG, Buzsáki G, et al. Fiberless multicolor neural optoelectrode for in vivo circuit analysis. *Sci. Rep.* 2016, 6(1): 1-13.

Comment 10: “The LEDs themselves are very large (125 x 180 micrometers) compared to a neuron. Is light emitted uniformly over the emitter, or are there “hot spots”? In particular, are any hot spots aligned between the red and blue LEDs?”

Our response:

The micro-LEDs we use have a lateral dimension of $125 \times 180 \mu\text{m}^2$. They are not designed to target a single neuron, but instead to target a large nucleus like VTA. Meanwhile, the micro-LEDs have a thickness of 7–8 μm , much thinner than commercial LEDs (e.g., CREE TR2227, ~50 μm thick) used in most recent publications (e.g., Montgomery, et al., Nature Methods, 12, 969, 2015; Yang, et al., Nature Neuroscience, 24, 1035, 2021).

The blue and red micro-LEDs are formed by high-quality, single crystalline III-V semiconductor materials (InGaN and InGaP), which are similar to industrial standard devices for lighting and displays. The devices have desirable performance for light emission. Their emissions are uniform.

In the near field, we can see the images below:

In the far field, we can see Fig. S2d. Both the blue and red LEDs have an ideal, nearly Lambertian pattern.

Our modification to the manuscript:

In Line 129, we add:

“...In addition, both devices exhibit a uniform, near Lambertian emission profile. ...”

Comment 11: *“stGtACR2 can be activated with arbitrary waveforms, not only “pulsed” (lines 109-110).”*

Our response:

We thank the reviewer for correcting this statement, which has been accordingly revised.

Our modification to the manuscript:

In Line 123, we delete the sentence:

“... as well as their different operational modes (pulsed red light activation for ChromsonR, and continuous blue light inhibition for stGtACR2), ...”

Comment 12: “Electrophysiological assessment of dual-color control is incomplete. Figure 4c shows a dual-color protocol, but the electrophysiological experiments (Figure 2) only used one color at a time (Fig. 2e-k) or simultaneous application of two colors without a single-color internal control (Fig. 2l-m). Please show electrophysiological results for the exact protocol used in the intact mice (Fig. 4c).”

Our response:

The electrophysiological assessment (Fig. 2) was performed before *in vivo* experiments (Fig. 5), and it is used to evaluate the photo response for cells co-expressing ChrimsonR and stGtACR2. We find that these cells can be selectively excited or inhibited by red or blue light, which establishes the foundation for *in vivo* demonstrations. Moreover, we find that introducing blue irradiances orders of magnitude lower than red irradiances can effectively suppress cell spikes and cause activity inhibition. In our view, these protocols applied in Fig. 2 are sufficient to demonstrate the feasibility and capability for bidirectional modulation.

We can also see that the *in vitro* electrophysiological results (Fig. 2) are not identical to the *in vivo* results in Fig. 5. In fact, higher irradiances for blue LEDs are needed for effective inhibition *in vivo*. This may be due to the fact that blue light has higher absorption in tissue and stronger illumination is required to affect enough cells in the VTA. Therefore, we believe protocols applied during *in vitro* and *in vivo* experiments are not necessarily needed to be identical.

We acknowledge the fact that applying both red and blue illuminations simultaneously may generate various responses, depending on the optical irradiance, time duration, viral expression level, etc. Our results only explore a small fraction of such effects. We also agree that the bidirectional optogenetic modulations with combined opsins and combined red/blue illumination definitely worth further investigation. Some explorations, we believe, are beyond the scope of the current manuscript.

Our modification to the manuscript:

In Line 345, we add:

“... It should be noted that the *in vivo* results in Figure 5g are not quantitatively identical to the *in vitro* electrophysiological ones in Figure 2m. In fact, higher irradiances for blue LEDs are needed for effective inhibition *in vivo*. This may be due to the fact that blue light has higher absorption in tissue and stronger illumination is required to affect enough cells in the VTA. ...”

Comment 13: “Clarify the advantage relative to extra-cranial light sources. One of the potential advantages of wireless devices is the capability to conduct experiments with multiple subjects simultaneously, as suggested by Movie S4. Can experiments be designed to that effect?”

Our response:

We thank the reviewer for this excellent suggestion. The unique potential application with our wireless systems is to study interactive social activities among multiple animals. We have performed additional experiments and explore the social activities among two and three mice, and demonstrated bidirectional modulations of their social preference. These experiments have not been reported before and will be difficult to perform using a single-color emitter or a wired stimulation system.

Our modification to the manuscript:

In Line 64, we add:

“... Recent efforts also remarkably demonstrate bidirectional optogenetic modulations with implantable waveguides coupled to extra-cranial dual-color laser sources and their combination with *in vivo* electrophysiological recordings^{23, 36, 37}, but wireless operation has not been achieved. ...”

In Line 276, we add a new section:

“

Bidirectional optogenetic modulation of social interactions among multiple mice, with wireless dual-color micro-LED probes

Modulating dopaminergic neurons in the VTA region also influences the social preference, in particular, affective and aggressive behaviors^{49, 50}. Compared with tethered stimulation systems, our wirelessly operated micro-LED probes with independently addressable illuminations present prominent advantages in studying social interactions among multiple objects (Movie S4). Additionally, the colocalized, dual-color emission capability allows the bidirectional regulation of social activities in real time, which is difficult to access with single color stimulations^{49, 51}. In paired-object interactions (Figure 4a), two mice co-expressing ChrimsonR and stGtACR2 in the VTA are placed in an open-field arena, and synchronized red or blue stimulations are provided for 5 mins each. Mice behaviors are video recorded throughout the test period, with social activities (including sniffing and attacking) labeled and scored (Figure 4b). The total time spent in these activities are summarized and compared among different scenarios (red light on, blue light on, and resting) in Figure 4c. In comparison to the resting state, red or blue LED stimulations strongly promote or suppress social interactions between two objects in the experiment group, while no significant effects are observed in the control group (expressing EGFP and mCherry).

The wireless micro-LED probes can further be applied to interrogate more complicated interaction behaviors among multiple (> 2) objects. In Figure 4d, we place three mice in the same social chamber, and provide these animals with alternating red or blue LED stimulations. In such a testing paradigm, two mice receive red stimulations while the third one receives blue stimulation, then vice versa (Figure 4e). The mice with red stimulations spend more time in interacting with each other compared to the individual under blue illuminations, and the social preference can be reversed by alternating the stimulation patterns among these three mice (Figure 4f).

By contrast, summarized data among control groups present no significant difference under red or blue stimulations. Collectively, these social interaction explorations clearly demonstrate the unique advantage of our wireless and bidirectional LED stimulators in the interrogation of complex animal behaviors.

”

In Line 672, we add associated description of the method:

“

Social interaction test

After 2 weeks of recovery and AAV expression, social interaction tests are performed with *DAT-cre* mice. Two or three mice are placed in an open-field arena ($30 \times 30 \text{ cm}^2$), wireless dual-color micro-LED probes provide red (20 Hz, 20-ms pulse, 7 mA) and blue (continuous, 5 mA) stimulations, and their behaviors are recorded by a camera. In paired-mice experiments, the two mice receive the same 3-phase stimulations: 1) 5-min baseline (pretest, no stimulation), 5-min red LED stimulation, 5-min free interaction (no stimulation), and 5-min red LED stimulation; 2) 2-hour rest in individual housing; 3) 5-min baseline (pretest, no stimulation), 5-min blue LED stimulation, 5-min free interaction (no stimulation), and 5-min blue LED stimulation. For triple-mice interactions, the three mice receive the following stimulations: 5-min baseline (pretest, no stimulation), 5-min LED stimulation (red, red, blue), 5-min free interaction (no stimulation), and 5-min LED stimulation (blue, blue, red). The tests are repeated for three times, by alternating the order. Experiment groups express stGtACR2 + ChrimsonR, and control groups express EGFP + mCherry. The videos are analyzed by Etho Vision XT 15 and social behaviors including sniff and attack are manually labeled and scored.

”

We add a new Figure 4, including the results of bidirectional modulations of social interactive activities.

Figure 4

Figure 4. Bidirectional, *in vivo* optogenetic control of mice social interactions with wirelessly operated, dual-color micro-LED probes. **a**, Photograph of two *DAT-cre* mice with wireless probes in an arena (30 × 30 cm²). Experiment groups (Exp) express stGtACR2 + ChrimsonR, and control groups (Ctrl) express EGFP + mCherry. **b**, Representative behavioral sequences recorded for one mouse receiving red or blue stimulations during social interactions. Here sniffing and attacking behaviors are highlighted and labeled, and other behaviors include locomotion, resting and escaping. **c**, Summary of changes in social interactions under red or blue stimulations. Here we compare the time spent in sniff and attack during the 5 mins light stimulation with the 5 mins resting state. $n = 4$ groups for both Exp and Ctrl. **d**, Photograph of three *DAT-cre* mice with wireless probes in an arena. **e**, Representative behavioral sequences recorded for the three mouse (#A, #B, #C) receiving red or blue stimulation during social interactions. **f**, Summary of changes in social interactions under red or blue stimulations for each mouse. $n = 4$ groups for both Exp (left) and Ctrl (right). All data are represented as mean ± s.e.m.. Student's *t* test, * $P < 0.1$, ** $P < 0.01$, *** $P < 0.001$, **** $P < 0.0001$, n.s. $P > 0.1$.

Comment 14: “The manuscript could benefit from English grammar editing. For instance, line 138 – “to wirelessly operate”; line 271, “which further confirms with the”.”

Our response:

We thank the reviewer for helping on the grammar editing. These sentences have been corrected.

Our modification to the manuscript:

In Line 153, we modify to:

“... to wirelessly operate ...”

In Line 344, we modify to:

“... which further conforms with ...”

Comment 15: “Details about the optical model are missing (Supp. Lines 57-62).”

Our response:

We have provided more details about the optical model in the revised manuscript.

Our modification to the manuscript:

In Line 473, the Methods part, we modify the section of “Optical modeling”:

“

Optical modeling

A ray tracing method based on Monte Carlo simulations (TracePro free trial version) is used to simulate the light propagation in the brain tissue. In the model, the tissue (size $5 \times 5 \times 5 \text{ mm}^3$) has absorption coefficients of 0.2 /mm and 0.08 /mm, and scattering coefficients of 47 /mm and 35 /mm, for blue (475 nm) and red (630 nm) wavelengths, respectively. The tissue has an anisotropy factor of 0.85 and a refractive index of 1.36.⁷⁸ Planar sources similar to the surface of micro-LEDs (size $125 \times 180 \mu\text{m}^2$) emits 10^6 rays for each monochromic wavelength (475 nm and 630 nm), assuming a Lambertian distribution. For the stacked micro-LEDs, the blue and red sources are overlaid. For the lateral case, the two sources are adjacent to each other.

”

Comment 16: “Differences between *in vitro* (electrophysiology, Fig. 2m) and *in vivo* (dopamine readout, Fig. 4g) should be minimized/discussed.”

Our response:

The results in the *in vitro* part (Fig. 2m) and the *in vivo* part (Fig. 5g) basically show similar trends. Both of them reveal that red or blue illuminations can deterministically activate or inhibit neural activities, clearly demonstrating the capability of bidirectional modulations.

Quantitatively, it is difficult to directly compare the numbers in the two cases (*in vitro* in Fig. 2m and *in vivo* in Fig. 5g). For *in vitro* experiments (Fig. 2m), thin 2D brain slices are placed in the culture dish with a uniform illumination, so the optical irradiance (unit: mW/mm^2) can be well calibrated. By contrast, *in vivo* experiments (Fig. 5g) are performed in the 3D mouse brain, and optical irradiances vary in different locations. Therefore, we only provide the currents applied for micro-LEDs in Fig. 5g. Corresponding irradiances on the LED surfaces can be determined to be about 0–30 mW/mm^2 for the red LED (current 0–6 mA) and 0–20 mW/mm^2 for the blue LED (current 0–1 mA), and the optical intensity decays as light propagates within the tissue.

We can also see that the *in vitro* electrophysiological results (Fig. 2) are not quantitatively identical to the *in vivo* results in Fig. 5. In fact, higher irradiances for blue LEDs are needed for effective inhibition *in vivo*. This may be due to the fact that blue light has higher absorption in tissue and stronger illumination is required to affect enough cells in the VTA. Therefore, we believe protocols applied during *in vitro* and *in vivo* experiments are not necessarily needed to be identical.

Our modification to the manuscript:

In Line 345, we add:

“... It should be noted that the *in vivo* results in Figure 5g are not quantitatively identical to the *in vitro* electrophysiological ones in Figure 2m. In fact, higher irradiances for blue LEDs are needed for effective inhibition *in vivo*. This may be due to the fact that blue light has higher absorption in tissue and stronger illumination is required to affect enough cells in the VTA. ...”

Comment 17: *“Animals’ is a bit misleading, as only a mouse model was tested.”*

Our response:

We have used ‘mice’ or ‘mouse’ to replace ‘animals’ in the manuscript title and throughout the revised manuscript.

Comment 18: “Fig. 1d-1e - why is there a difference in the size of the blue and the red LEDs? This difference is not seen in fig 1c and is not mentioned in the given dimensions in line 94.”

Our response:

This is an interesting observation. The blue and red micro-LEDs in the probe structure are designed to have the same lateral dimension ($125\ \mu\text{m} \times 180\ \mu\text{m}$), which is precisely defined by the mask layout during photolithography. However, red LEDs are based on GaAs/InGaP material systems that absorb visible light and look black, while blue LEDs are based on GaN/InGaN material systems that is transparent for visible. Therefore, waveguide effects occur within the blue LED film, which make the device edge look brighter.

Microscopic images of individual red and blue LEDs are shown below (on and off states). We can see that the two LEDs have similar dimensions, but the blue LED appears to be “larger” than the red one, when it is turned on.

Comment 19: “Fig. 1g – it is not clear that the first three columns are the new presented technology. Is the last column another probe you tried? If so, I would like to see if this probe could not achieve all the results you have achieved with the presented technology, thus establishing the unique results the new probe offers.”

Our response:

In Fig. 1g, the first three columns present the experimental photographs and simulated results illustrating blue and red light propagations for the probe with stacked blue and red LEDs. The last (4th) column shows the result for a different probe with laterally arranged red and blue micro-LEDs adjacent to each other, as a comparison.

We agree that a probe with laterally arranged red and blue micro-LEDs still shows overlapped dual-color emissions and can enable bidirectional optogenetic stimulations in certain applications. However, it is noted that the overlapping volume in such a design is much smaller than that in a vertically stacked micro-LED design (Fig. 1g). It means that more currents need to be injected into the LEDs to create a larger illuminating region, which inevitably increases tissue heating as well as the power consumption. By contrast, our unique vertical stacking design makes sure that the overall device has a smaller footprint and generates the optimal overlapping optical profile for dual-color stimulation.

Our modification to the manuscript:

In Line 138, we add:

“... Compared to a probe with laterally arranged red and blue micro-LEDs displaying highly misaligned emissions, the vertically assembled device provides a smaller footprint and generates the optimal overlapping profile for dual-color stimulation. ...”

In the caption of Fig. 1g, we add:

“... Experimental photographs and simulated results illustrating blue and red light propagations for a micro-LED probe embedded into a brain phantom (column 1–3). Results for a probe with laterally assemble blue and red LEDs are also presented (column 4), ...”

Comment 20: “Fig. 1h - how does the chip connect to the probe? A photo would clarify this.”

Our response:

We have added a new Figure S8 in the supplement, to illustrate the probe and the circuit connector.

Our modification to the manuscript:

We add Fig. S8

Figure S8

Figure S8. Optical images of a micro-LED probe and a control circuit (a) before and (b) after connection via a standard 4-pin connector.

Comment 21: “Fig. 1i - very low resolution of the photos.”

Our response:

High quality photos have been updated in Fig. 1i.

Our modification to the manuscript:

We update Fig. 1i:

Comment 22: “Fig. 2d - it is written in the text (line 161) that ‘> 83% of cells co-express both opsins’. I believe you meant ‘> 83% of the labeled cells’. Is that the case? What are the precise percentages?”

Our response:

We apologize for the confusion. About 83% of these labeled cells co-express both opsins.

Our modification to the manuscript:

In Line 178, we modify it to:

“... about 83% of these labeled cells co-express both opsins in average (Figure 2d). ...”

Comment 23: “Fig. 2 (and lines 176-179) - bidirectional activation in the same cell was shown in a previous paper (reference 33). This was done in a transgenic mouse injected with a mix of two viral vectors. Is the viral cocktail you used the novel procedure to be emphasized here? Calibration of the light intensity needed to use with these viral vectors? Otherwise, I am not sure how this figure is adding new information, especially when lasers are used, and not the probe your paper is about”

Our response:

As the reviewer mentioned, bidirectional activation in the same cell was shown previously, by mixing two opsins (ChR2 and Jaws) [Noked, et al., IEEE Trans Biomed Eng 68:416-427 (2021)]. We agree that there could be many possibilities for different viral mixtures. Here we adapt a different combination of opsins (ChrimsonR and stGtACR2), which has not been explored in prior works. Results in Fig. 2 demonstrate the efficacy of such a new combination. We use collimated laser sources for illumination here, mainly because they provide a uniform irradiation with well calibrated power densities. These results can provide a basis and guideline for *in vivo* experiments.

Our modification to the manuscript:

In Line 401, we add:

“... Other combinations of opsins, like ChR2 and Jaws³⁶, have also been proved to be feasible. ...”

Comment 24: “Fig. 3b - on line 200 it says that this panel illustrates the locations of the micro-LEDs in the VTA. Does this mean this is an illustration? I am not sure how were these photos created.”

Our response:

The photos were taken by placing a dual-color micro-LED probe underneath the brain slice, with a confocal fluorescence microscope. The relative position of the micro-LEDs is around the location for *in vivo* injection.

Our modification to the manuscript:

In Line 235, we modify it to:

“... Figure 3b presents fluorescence images of a brain slice with a micro-LED probe underneath it, showing the co-expression of ChrimsonR and stGtACR2 in the VTA, ...”

Comment 25: *“Fig. S10 – what are “functional samples”? Are the five sample five mice? five micro-LEDs? five couples of probes (10 micro LEDs, five red and five blue)?”*

Our response:

In this study, 5 dual-color probes are separately implanted into 5 behaving mice. Probes with both red and blue micro-LEDs operating in the normal condition are defined as "functional probes".

Our modification to the manuscript:

In Fig. S16, we modify the legend:

“... 5 probes are separately implanted into 5 behaving mice. Probes with both red and blue micro-LEDs operating in the normal condition are defined as "functional probes". ...”

Comment 26: “Fig. 3f – I am not sure why the significance test was done once in reference to the pretest and once in reference to the Red+. what is the difference between pretest and Blue +?”

Our response:

The real-time place preference / avoidance (RTPP) test consists of 3 phases: 10-min baseline (pretest), 30-min red LED stimulation (RED+), 30-min rest (no stimulation), and 30-min blue LED stimulation (Blue+). During the pretest phase, individual mice are allowed to freely explore the entire apparatus for 10 min, then a 10-min red stimulation is given. So the influence of the red LED is compared with the case in pretest. After 30-min rest, a 10-min blue stimulation is given, so the influence of the blue LED is compared to the phase Red+.

We have updated Fig. 3f, and also included the significant test between pretest and Blue+, the *P* value is 0.0182 (* *P* < 0.1).

Our modification to the manuscript:

We modify Fig. 3f:

Comment 27: “*Fig. 4g - what do the stars mean?*”

Our response:

In Fig. 5g, the stars indicate results for five traces shown in Fig. 5f.

Our modification to the manuscript:

In the caption of Fig. 5g, we add:

“...the stars indicate results for five traces shown in Fig. 5f. ...”

Comment 28: *“Lines 213-215 - I suggest it would be added in the main text that the stimulation only occurred in the left chamber.”*

Our response:

We have updated the text according to this comment.

Our modification to the manuscript:

In Line 254, we add:

“... The red or blue LED stimulations are only provided once the mice appear in the left chamber. ...”

Comment 29: *“Movie S3, S5 - does the driving chip support the multi-channel and multi-color modes?”*

Our response:

The multi-channel and multi-color probes shown in Movies S3 and S5 are driven by a wired circuit, not by the same circuit mentioned in the main text.

In our current module, the 24-pin Bluetooth chip (nRF24LE1) can control two LED drivers (ZLED7012), each of which can drive 4 LEDs. So the circuit can drive 2×4 micro-LEDs in total. By upgrading the Bluetooth chip to a 32-pin nRF24LE1 with a similar size, the circuit can drive as many as 10×4 micro-LEDs in total.

Our modification to the manuscript:

In the captions of Movies S4 and S5, we add:

“... These micro-LEDs are driven by a wired external power source. ...”

Comment 35: *“Fig. 2f, 2i, 2l – why were these values chosen?”*

Our response:

In Fig. 2, we choose the intensity values for both red and blue light that can cover the spike activation/inhibition probability from 0 to 1. The values of irradiance (unit: mW/mm^2) are calculated based on the power of light source (0–10 mW) and the area of the laser spot (diameter ~ 1 mm).

Comment 36: “Fig. S8 – small DC response unnoticed.”

Our response:

The neuron activity was recorded at current-clamp patch recording mode in brain slice. The small DC response in Fig. S8 (now Fig. S10) is caused by the red-light depolarization of ChrimsonR, but not fully activated because the cell is below the threshold or the spikes are suppressed by the blue light.

Similar “DC response” can be seen in the literature, for example:

Fig. 3d, in Vierock et al., Nature Comm., 12:4527 (2021).

Fig. 2a, in Klapoetke et al., Nature Methods, 11:338 (2014).

Comment 41: “Fig. 4h – I suggest to add a graphical legend so it would be easier to understand the panel without referring to the legend text.”

Our response:

We have updated Fig. 5h with more detailed information.

Our modification to the manuscript:

We modify Fig. 5h:

Other minor points

Comment 30: “Line 88 – I suggest the word “enlarged” instead of “exploded””

Comment 31: “Line 90 – I suggest to add “of” after “comprises””

Comment 32: “Fig. 1f – I suggest to print the headlines in a different order: red LED and ChrimsonR one below the other on the right (above the corresponding curve), and blue LED and stGtACR2 on the left.”

Comment 33: “Line 143 - I suggest “commands” instead of “commends””

Comment 34: “Line 183 – I suggest “are suppressed” instead of “suppressions””

Comment 37: “Line 203 – I suggest “assess” instead of “access””

Comment 38: “Line 211 – I suggest “implantation” instead of “injection””

Comment 39: “Fig. 3d – I suggest to fix shaded areas to match exactly 30 min in the graph.”

Comment 40: “Line 251 – I suggest “Other” instead of “More””

Comment 42: “Supplementary line 171 – I suggest “finally” instead of “at last”.”

Our response:

We thank the reviewer for these detailed suggestions. All of these points have been addressed in the revised text accordingly.

Reviewer #3

The manuscript reports on the application of dual-color optoelectronic probes in the VTA of freely behaving animals. The optrodes are used in place preference test controlling the μ LEDs using a smaller wireless headstage. The authors demonstrate by increasing or decreasing dopamine levels excited through colocalized red and blue stimulations rewarding and aversive behaviors of freely moving animals. The probes combine existing technologies to realize blue and red μ LEDs (cf. Refs 1&2 of the supplementary information) and – for the first time – their combination as a stacked μ LED sandwich with a filter placed in-between both μ LEDs. The μ LED probe is validated in vitro and in vivo with respect to electrical, optical, and thermal properties. The in vivo application targets dopaminergic neurons in the VTA – increasing and decreasing the dopamin level by optical stimuli at different wavelengths.

While the neuroscientific experiments are convincing, a couple of aspects with respect to the optoelectronic probe fabrication and characterization need to be clarified as further detailed.

Our response:

We thank the reviewer for these positive comments and valuable suggestions. The technical questions have been addressed in the following discussions.

Dual color optoelectronic probe fabrication

Comment 1: “The μ LEDs are relatively large compared to the state of the art, cf. Yoon Lab at Univ. of Michigan, Ann Arbor, Mathieson Lab at Strathclyde Univ., Glasgow Scotland, and Schwarz Lab at Fraunhofer Institute IAF, Freiburg, Germany (Gossler et al.), Rogers Lab at Univ. of Illinois at Urbana-Champaign, USA: these probes are thinner (when realized on polymeric substrates provide an improved flexibility), provide smaller LEDs; the statement that the probes are flexible is a relative statement at a documented thickness of 120 μm – it has to be taken into account that the stiffness increases with the thickness to the power of 3.”

Our response:

The choice of device dimensions does not only depend on the lesion or mechanical properties, but also depends on their specific applications. The micro-LEDs we use have a lateral dimension of $125 \times 180 \mu\text{m}^2$. They are not designed to target a single neuron or a small nucleus, but instead to target a large nucleus like VTA.

Meanwhile, the micro-LEDs have a thickness of 7–8 μm , much thinner than commercial LEDs (e.g., CREE TR2227, ~50 μm thick) often used in some recent publications including Rogers Lab (e.g., Yang, et al., Nature Neuroscience, 24, 1035, 2021) and Deisseroth Lab (e.g., Montgomery, et al., Nature Methods, 12, 969, 2015;).

While very small InGaN blue micro-LEDs (~20 μm) can be directly grown and patterned on silicon, for example, by Yoon Lab [Wu, et al., Neuron, 88, 1136 (2015)], the combination of very small InGaP red and InGaN blue micro-LEDs is much more challenging and has not been accomplished in prior works.

We agree that our micro-LED probe does not have the smallest dimension. However, the goal of this research is not to break the world record for probe dimension. Instead, we design and fabricate a dual-color micro-LED probe that is suitable and works properly for bidirectionally interrogating the VTA region, and successfully activate and inhibit neural activities in the same mice, and deterministically evoke reward and aversion based behaviors, respectively. These results have not been demonstrated in previous works.

Comment 2: “It would be interesting to see how small the LEDs can be made; which smallest pitch can be achieved in case of multiple LED probes; how small the LED substrates can be made? How precisely can the μ LEDs be positioned on the polyimide substrate?”

Our response:

The blue and red micro-LEDs are formed by high-quality, single crystalline III-V semiconductor materials (InGaN and InGaP), which are similar to industrial standard devices for lighting and displays. The devices have desirable performance for light emission. The sizes of these micro-LEDs are defined by standard photolithography, with a resolution of $\sim 2 \mu\text{m}$.

The transfer process is performed with a customized setup based on a mask aligner for positioning and alignment. The precision of positions for these micro-LEDs is $\sim 2 \mu\text{m}$.

Below show an array of blue and green micro-LEDs (size $\sim 20 \times 30 \mu\text{m}^2$) transferred on glass substrates, fabricated in our lab (unpublished data).

As the reviewer mentioned, similar small InGaN blue micro-LEDs ($\sim 20 \mu\text{m}$) can be directly grown and patterned on silicon probes as well, for example, by Yoon Lab [Wu, et al., *Neuron*, 88, 1136 (2015)]. However, the combination of very small InGaP red and InGaN blue micro-LEDs on flexible probes is much more challenging and has not been accomplished in prior works.

The choice of device dimensions does not only depend on the lesion or mechanical properties, but also depends on their specific applications. The micro-LEDs we use have a lateral dimension of $125 \times 180 \mu\text{m}^2$. They are not designed to target a single neuron or a small nucleus, but instead to target a large nucleus like VTA.

Even though smaller micro-LEDs can be fabricated, it is known that reducing LED size causes dramatic efficiency decreases for all types of LEDs, particularly InGaP based red LEDs that are more susceptible to sidewall defects.

For example:

InGaN blue LED:

Fig. 5. EQE curves showing experimental (solid circles) and expected maximum (empty circles).

F. Olivier et al., Influence of size-reduction on the performances of GaN-based microLEDs for display application. *J. Lumin.* 191, 112–116 (2017).

AlGaInP red LED:

Fig. 1. (a) EQE curves for six different LEDs as a function of current density. (b) Normalized EQEs at 5 A/cm² for the two sets of different-size LEDs.

J. T. Oh et al., Light output performance of red AlGaInP-based light emitting diodes with different chip geometries and structures. *Opt. Express* 26, 11194–11200 (2018).

Therefore, the use of smaller LEDs will inevitably require higher injection currents to reach sufficient irradiance for stimulating a large nucleus like VTA, which will cause more tissue heating.

We agree that our micro-LED probe does not have the smallest dimension. However, the goal of this research is not to break the world record for probe dimension. Instead, we design and fabricate a dual-color micro-LED probe that is suitable and works properly for bidirectionally interrogating the VTA region, and successfully activate and inhibit neural activities in the same mice, and deterministically evoke reward and aversion based behaviors, respectively. These results have not been demonstrated in previous works.

Our modification to the manuscript:

In Line 390, we add:

“...While blue InGaN micro-LEDs can be directly grown on silicon and integrated with recording electrodes⁴¹, the combination simultaneous dual-color stimulations

and electrophysiological recordings in the deep brain region like VTA require more sophisticated device design and fabrication and have not been attempted here. ...”

In Line 104, we add:

“... The dimensions of our LEDs are designed to target a large nucleus like VTA. Although smaller devices down to 10–20 μm can be fabricated^{41, 42}, reducing LED size decreases luminescence efficiencies^{43, 44} and may cause additional tissue heating. ...”

In Line 451, we add:

“... (with a lateral alignment accuracy of $\sim 2 \mu\text{m}$) ...”

In the reference list, we add:

43. Olivier F, Tirano S, Dupré L, Aventurier B, Largeton C, Templier F. Influence of size-reduction on the performances of GaN-based micro-LEDs for display application. *J. Lumin.* 2017, 191: 112-116.3

44. Oh JT, Lee SY, Moon YT, Moon JH, Park S, Hong KY, et al. Light output performance of red AlGaInP-based light emitting diodes with different chip geometries and structures. *Opt. Express* 2018, 26(9): 11194-11200.

Comment 3: *“How are the μ LEDs/filter fixed on the polyimide substrate and on each other?”*

Our response:

The transfer process is performed with a customized setup based on a mask aligner for positioning and alignment. The LEDs and filter are transferred on the polyimide with a spin-coated SU-8 epoxy layers as bonding interfaces.

Our modification to the manuscript:

In Line 449, we highlight:

“... From bottom to top, a red LED, an optical filter and a blue LED are sequentially transfer printed on the film by PDMS in a vertical stack, via a customized alignment setup (with a lateral alignment accuracy of $\sim 2 \mu\text{m}$). Between each device layer, spin-coated SU-8 based epoxy (thickness $\sim 2\text{--}5 \mu\text{m}$) serves an optically transparent and electrically insulating bonding layer. ...”

Comment 4: “Are the 2 copper layers needed on the polyimide substrate? Aren’t these layers increasing the probe stiffness too much? What about the biocompatibility of the applied copper? Can the authors guarantee that copper does not leak out?”

Our response:

The copper layers are added to enhance the thermal conductance of the probe, which greatly reduces the tissue heating effects [Li L, *Adv. Mater. Technol.* 2018, **3**(1): 1700239].

The thin-film Cu/PI/Cu probe has a measured Young’s modulus of ~15 GPa and a bending stiffness of $\sim 9 \times 10^4$ pNm². It is slightly harder than a PI probe with the same thickness (~2 GPa), but still much softer than silicon (~180 GPa) and tungsten (~400 GPa).

Our immunoreactivity results (Fig. S13) indicate that our micro-LED probe shows biocompatibility comparable to a glass fiber. Chronic stability tests (Fig. S16) show that the device can work properly after long term operation (more than ~200 days).

In addition, we perform mass spectrometric analysis and show that the brain tissue has minimal accumulation of dissolved Cu element 5 weeks post implantation (Fig. S14). The measured Cu concentration is 18.2 ± 2.3 µg/g in dried brain tissue for mice with probe implantation, compared to the result (14.4 ± 0.2 µg/g) for the control group without probe. Both results are in the normal range (< 20 µg/g in mice brain¹, and 30–100 µg/g in human brain²).

Reference 1: Chen Y, Wang L, Geng J-H, Zhang H-F, Guo L. Apolipoprotein E deletion has no effect on copper-induced oxidative stress in the mice brain. *Biosci Rep* 2018, **38**(5).

Reference 2: Harrison WW, Netsky MG, Brown MD. Trace elements in human brain: Copper, zinc, iron, and magnesium. *Clinica Chimica Acta* 1968, **21**(1): 55-60.

Our modification to the manuscript:

In Line 100, we add:

“... The Cu coating on PI serves as an efficient heat spreader to reduce the probe’s operation temperature³⁷. ...”

In Line 114, we add:

“...The Cu coated PI substrate has a measured Young’s modulus of ~15 GPa, softer than silicon (~180 GPa) and tungsten (~400 GPa), but still much harder than the brain tissue (~1 kPa). The formed probe has a bending stiffness of $\sim 9 \times 10^4$ pNm², similar to metal electrodes used for electrophysiological studies⁴⁵. ...”

We add a new reference to compare the mechanical properties of probes made by different materials:

45. He F, Lycke R, Ganji M, Xie C, Luan L. Ultraflexible neural electrodes for long-lasting intracortical recording. *iScience* 2020: 101387.

In Line 242, we add:

“... Also, mass spectrometric analysis shows that the brain tissue has minimal accumulation of dissolved Cu element 5 weeks post implantation (Figure S14). ...”

We add Fig. S14,

Figure S14

Figure S14. Measured Cu concentration in the brain tissue. The brain tissue is dried and analyzed with inductively coupled plasma mass spectrometry (Thermo ICP-MS iCAPQ, ThermoFisher, USA). The measured Cu level is 18.2 ± 2.3 µg/g for mice 5 weeks post probe implantation (experiment group). The result for the control group without probe is 14.4 ± 0.2 µg/g. The Cu levels in both cases are in the normal range (< 20 µg/g in mice brain, and 30–100 µg/g in human brain). $n = 3$ mice for each group.

References:

Chen Y, Wang L, Geng J-H, Zhang H-F, Guo L. Apolipoprotein E deletion has no effect on copper-induced oxidative stress in the mice brain. *Biosci Rep* 2018, 38(5).

Harrison WW, Netsky MG, Brown MD. Trace elements in human brain: Copper, zinc, iron, and magnesium. *Clinica Chimica Acta* 1968, 21(1): 55-60.

Comment 5: “How is the PDMS layer applied? By dip coating?”

Our response:

Yes. The PDMS layer is applied by dip coating.

Our modification to the manuscript:

In Line 456, we modify:

“...A bilayer of PDMS (~20 μm , by dip coating) and parylene (~15 μm , by CVD) is used for waterproof encapsulation...”

Comment 6: *“Have the lateral LED probes been applied in this study?”*

Our response:

We have not applied the lateral LED probe for biological studies.

We agree that a probe with laterally arranged red and blue micro-LEDs still shows overlapped dual-color emissions and can enable bidirectional optogenetic stimulations in certain applications. However, it is noted that the overlapping volume in such a design is much smaller than that in a vertically stacked micro-LED design (Fig. 1g). It means that more currents need to be injected into the LEDs to create a larger illuminating region, which inevitably increases tissue heating as well as the power consumption. By contrast, our unique vertical stacking design makes sure that the overall device has a smaller footprint and generates the optimal overlapping optical profile for dual-color stimulation.

Our modification to the manuscript:

In Line 138, we add:

“... Compared to a probe with laterally arranged red and blue micro-LEDs displaying highly misaligned emissions, the vertically assembled device provides a smaller footprint and generates the optimal overlapping profile for dual-color stimulation. ...”

Comment 7: “LED dimensions: A general question to be answered is on whether the larger LED size of 125x180 μm is requested to optically stimulate enough brain tissue to elicit behavioral changes; could the same effect be achieved with smaller LEDs as used for instance by the Yoon Lab?”

Our response:

As the reviewer mentioned, similar small InGaN blue micro-LEDs ($\sim 20 \mu\text{m}$) can be directly grown and patterned on silicon probes as well, for example, by Yoon Lab [Wu, et al., *Neuron*, 88, 1136 (2015)]. However, the combination of very small InGaP red and InGaN blue micro-LEDs on flexible probes is much more challenging and has not been accomplished in prior works.

The choice of device dimensions does not only depend on the lesion or mechanical properties, but also depends on their specific applications. The micro-LEDs we use have a lateral dimension of $125 \times 180 \mu\text{m}^2$. They are not designed to target a single neuron or a small nucleus, but instead to target a large nucleus like VTA.

Even though smaller micro-LEDs can be fabricated, it is known that reducing LED size causes dramatic efficiency decreases for all types of LEDs, particularly InGaP based red LEDs that are more susceptible to sidewall defects.

For example:

InGaN blue LED:

Fig. 5. EQE curves showing experimental (solid circles) and expected maximum (empty circles).

F. Olivier et al., Influence of size-reduction on the performances of GaN-based microLEDs for display application. *J. Lumin.* 191, 112–116 (2017).

AlGaInP red LED:

Fig. 1. (a) EQE curves for six different LEDs as a function of current density. (b) Normalized EQEs at 5 A/cm² for the two sets of different-size LEDs.

J. T. Oh et al., Light output performance of red AlGaInP-based light emitting diodes with different chip geometries and structures. *Opt. Express* 26, 11194–11200 (2018).

Therefore, the use of smaller LEDs will inevitably require higher injection currents to reach sufficient irradiance for stimulating a large nucleus like VTA, which will cause more tissue heating.

We agree that our micro-LED probe does not have the smallest dimension. However, the goal of this research is not to break the world record for probe dimension. Instead, we design and fabricate a dual-color micro-LED probe that is suitable and works properly for bidirectionally interrogating the VTA region, and successfully activate and inhibit neural activities in the same mice, and deterministically evoke reward and aversion based behaviors, respectively. These results have not been demonstrated in previous works.

Our modification to the manuscript:

In Line 390, we add:

“...While blue InGaN micro-LEDs can be directly grown on silicon and integrated with recording electrodes⁴¹, the combination simultaneous dual-color stimulations and electrophysiological recordings in the deep brain region like VTA require more sophisticated device design and fabrication and have not been attempted here. ...”

In Line 104, we add:

“...The dimensions of our LEDs are designed to target a large nucleus like VTA. Although smaller devices down to 10–20 μm can be fabricated^{41, 42}, reducing LED size decreases luminescence efficiencies^{43, 44} and may cause additional tissue heating. ...”

In the reference list, we add:

43. Olivier F, Tirano S, Dupré L, Aventurier B, Largeron C, Templier F. Influence of size-reduction on the performances of GaN-based micro-LEDs for display application. *J. Lumin.* 2017, 191: 112-116.3

44. Oh JT, Lee SY, Moon YT, Moon JH, Park S, Hong KY, et al. Light output performance of red AlGaInP-based light emitting diodes with different chip geometries and structures. *Opt. Express* 2018, 26(9): 11194-11200.

Thermal characterization

Comment 8: “How have the authors calibrated the temperature measurements taking the emissivity of the LED materials into account? How do the measurements taken with the IR camera compare to the measurements using the thermocouples? This could be done in the tissue phantom. Are the extracted temperatures those of the LED surface or the surrounding tissue?”

Our response:

During *in vitro* tests using a brain phantom, the IR camera captures the temperature on the phantom surface, not on the LED semiconductor materials. The emissivity of the phantom is similar to that of human tissue (~0.97, very close to 1.0), due to the high water content. This emissivity has been taken into account during measurement.

As the reviewer suggested, we perform additional experiments by inserting a thermocouple into the phantom (~0.3 mm above the LED probe tip), and measure the temperature inside the phantom. The results are similar to those via IR camera and consistent with those measured inside the mice brain.

Our modification to the manuscript:

In Line 486, we add:

“...The temperature distributions on the phantom surface are mapped with an infrared camera (FOTRIC 228). For comparison, the temperature inside the phantom is also measured by inserting an ultra fine flexible micro thermocouple (Physitemp, IT-24P) (~0.3 mm above the LED probe tip). ...”

We modify Fig. S3, including the results in phantom measured by a thermocouple.

Figure S3

Figure S3. Thermal behaviors of a dual-color micro-LED probe. (a, b) Images and infrared (IR) thermographs showing a probe embedded into a brain phantom (~0.5 mm below the phantom surface). (a) Red LED on (injection current 7 mA, pulse frequency 20 Hz, pulse duration 10 ms). (b) Blue LED on (injection current 2 mA, continuous operation). (c) Measured temperature rise on the phantom surface by IR camera. (d) Measured temperature rise in the phantom by thermocouple. (e) Measured temperature rise in the brain of living mice by thermocouple. Red LED current 0–10 mA, frequency 10–50 Hz, pulse duration 10 ms). (f–h) Corresponding results when the blue LED is on (continuous mode).

Comment 9: “Are the copper layers on the polyimide substrate taken as a heat spreader?”

Our response:

Yes. The copper layers are added to enhance the thermal conductance of the probe, which greatly reduces the tissue heating effects [Li L, *Adv. Mater. Technol.* 2018, **3**(1): 1700239].

Our modification to the manuscript:

In Line 100, we add:

“... The Cu coating on PI serves as an efficient heat spreader to reduce the probe’s operation temperature³⁷. ...”

Comment 10: “*In vitro tests with thermocouples: How do the thermocouples influence the temperature measurement as their wire represent a further path to conduct heat away from the LED?*”

Our response:

We agree that the insertion of thermocouples may influence the temperature measurement. We perform additional thermal simulations by including the thermocouple within the tissue.

In the model, the thermocouple (Physitemp, IT-24P) consists of two metal wires: copper (thermal conductivity 400 W/m/K, diameter 31 μm) and constantan (45Ni-55Cu alloy, thermal conductivity 21 W/m/K, diameter 31 μm), which are encapsulated by polyester to form a wire with an outer diameter of 127 μm . Compared to the results without the thermocouple in the model, we find that the tissue temperature rise decreases by $\sim 5\%$ or $\sim 20\%$ when the red LED or the blue LED is on, respectively. Therefore, the thermocouple indeed influences the temperature measurement and modeling. The results are less affected when the red LED is on, because the red LED is in the bottom layer of the stacked structure and closer to the Cu layer in the probe substrate.

Our modification to the manuscript:

In Line 502, we add:

“... The thermocouple is also included in the model for comparison, which is 0.3 mm above the LED probe tip. It consists of two metal wires: copper (thermal conductivity 400 W/m/K, diameter 31 μm) and constantan (45Ni-55Cu alloy, thermal conductivity 21 W/m/K, diameter 31 μm), which are encapsulated by polyester to form a wire with an outer diameter of 127 μm”

We modify Fig. S4, including the simulated thermal results by including the thermocouple in the model.

Figure S4

Figure S4. Simulated thermal behaviors of a dual-color micro-LED probe. (a) Simulated steady-state temperature distribution in the brain tissue surrounding the probe, when the red LED is on (current 7 mA, frequency 20 Hz, pulse duration 10 ms). (b) Simulated maximum temperature rise of the brain tissue when the red LED is in different operation conditions. (c) Simulated maximum temperature rise of the brain tissue when the red LED is on (frequency 50 Hz, pulse duration 10 ms). The results for models with and without the thermocouple are compared. (d–f) Corresponding results when the blue LED is on (continuous mode), current = 2 mA in (d).

Comment 11: *“Thermal modeling: which material parameters of the LED probe have been applied for the FE simulation of the temperature distribution?”*

Our response:

We have provided more details about the thermal modeling in the Methods part of the revised manuscript.

Our modification to the manuscript:

In Line 494, we add:

“... Materials and corresponding parameters (density, thermal conductivity and heat capacity) used in the model include: brain tissue (1.1 g/cm³, 0.5 W/m/K, 3.7 J/g/K), parylene (1.2 g/cm³, 0.082 W/m/K, 0.71 J/g/K), PDMS (0.98 g/cm³, 0.16 W/m/K, 1.5 J/g/K), polyimide (1.4 g/cm³, 0.15 W/m/K, 1.1 J/g/K), GaN (6.1 g/cm³, 130 W/m/K, 0.49 J/g/K), Cu (9.0 g/cm³, 400 W/m/K, 0.39 J/g/K), SU-8 (1.2 g/cm³, 0.2 W/m/K, 1.2 J/g/K), GaP (4.4 g/cm³, 110 W/m/K, 0.43 J/g/K). ...”

Comment 12: “In a recent paper by Zgierski-Johnston et al. (doi: 10.1016/j.pbiomolbio.2019.11.004), the temperature increase in cardiac tissue (presumably with similar thermal properties as brain tissue) was measured both for pulsed and continuous optical stimulation (in both cases 10 mA, 50% duty cycle at 10 kHz) using commercial CREE LEDs of comparable size. The extracted temperature increase was in the continuous operation mode up to 8 K after 1.67 sec. Please comment on the pronounced temperature increase extracted directly under the LEDs in comparison to your moderate temperature increase.”

Our response:

We thank the reviewer for introducing this recent paper to us. In this paper by Zgierski-Johnston et al., the CREE LEDs are bonded on a polyimide based flexible probe, without copper coatings as a heat spreader. Therefore, the thermal effect is more pronounced than our results. These results are consistent with our previous publication, which demonstrates that the Cu coated PI substrate can significantly mitigate the heating effect.

Li L, et al. Heterogeneous Integration of Microscale GaN Light-Emitting Diodes and Their Electrical, Optical, and Thermal Characteristics on Flexible Substrates. *Advanced Materials Technologies* 2018, 3(1): 1700239.

(note the experiments in this paper are performed in air, not in tissue)

Figure 4. a) Microscope images of a micro-LED printed and metallized on a PI-based flexible needle with (right) and without (left) illumination. b,c) Measured (left) and simulated (right) temperature distributions for CSS micro-LEDs on Cu b) and PI c), with an injected current of 5 mA. d) Measured and e) simulated maximum temperature rises (located on the LED surface) as a function of injected current for CSS and PSS micro-LEDs on PI and Cu. f) Measured maximum temperature rises for CSS LEDs on PI and Cu under pulsed current injection (3 mA) with different duty cycles.

Probe implantation

Comment 13: “Do you need to remove the dura for optrode implantation? Is the stiffness high enough for a direct implantation through the dura without a shuttle as requested for thinner, polymeric recording probes (cf. Lan Luan et al.)”

Our response:

Yes. We need to remove the dura before probe implantation.

The thin-film probe has a measured Young’s modulus of ~15 GPa and a bending stiffness of $\sim 9 \times 10^4$ pNm². It is softer than silicon (~180 GPa) and tungsten (~400 GPa), but still much harder than the brain tissue (~1 kPa). The probe’s critical buckling load is ~35 mN, which is much larger than the load (~90 μ N) required to puncture the brain. Therefore, no other supporting shuttles are requested to assist the implantation as requested for thinner polymer probes.

Our modification to the manuscript:

In Line 552, we add:

“...A small craniotomy is made, followed by carefully removing the dura with a thin needle. ...”

In Line 114, we add:

“...The Cu coated PI substrate has a measured Young’s modulus of ~15 GPa, softer than silicon (~180 GPa) and tungsten (~400 GPa), but still much harder than the brain tissue (~1 kPa). The formed probe has a bending stiffness of $\sim 9 \times 10^4$ pNm², similar to metal electrodes used for electrophysiological studies⁴⁵. ...”

We add a new reference to compare the mechanical properties of probes made by different materials:

45. He F, Lycke R, Ganji M, Xie C, Luan L. Ultraflexible neural electrodes for long-lasting intracortical recording. *iScience* 2020: 101387.

Long-term optrode testing

Comment 14: “How did the authors test the long-term performance of the LEDs and –in particular– the long-term stability of the PDMS / parylene C based encapsulation layer stack? Was the probe just exposed to the salt solution and tested from time to time or have the authors operated the LEDs continuously while exposing them to the electrolyte (or in vivo to brain tissue)?”

Our response:

We apologize for the confusion. In Fig. S16, we test the long-term performance of the micro-LED probe by separately implanting 5 probes into 5 behaving mice. Probes with both red and blue micro-LEDs operating in the normal condition are defined as "functional probes". These probes are kept within the mouse brain. Their performance is evaluated by measuring LED's forward voltages at a constant current of 0.7 mA. After 365 days, the functional probes are taken out and still operate normally.

Our modification to the manuscript:

We modify Fig. S16, to include more details about the long-term stability test.

Figure S16

Figure S16. (a) Chronic stability for the dual-color micro-LED probe. 5 probes are separately implanted into 5 behaving mice. Probes with both red and blue micro-LEDs operating in the normal condition are defined as "functional probes". (b) Summary of the performance for red and blue micro-LEDs in each probe. These probes are kept within the mouse brain. Their performance are evaluated by measuring LED's forward voltages at a constant current of 0.7 mA. After 365 days, the functional probes are taken out and still operate normally.

To be clarified:

Comment 15: “Line 275-276: “The red illumination triggers the elevated dopamine release, while the use of blue light can effectively and promptly suppress the signal.” From Figure 4h I conclude that red illumination results in an immediate response while the blue stimulus causes a slow reduction in DA concentration starting after 5s of blue light stimulation. The authors are asked to clarify.”

Our response:

We apologize for the confusion. In Fig. 5h, the blue light stimulation (blue curve) is only performed at 6–12 s, not at 0–12 s.

Our modification to the manuscript:

We modify Fig. 5h and include some details about the experiment conditions.

Typos (changes indicated by **)

Comment 16: “Line 42 – Change to: Understanding ** brain functions and treating neurological disorders rely on the continuous development of advanced technologies to interrogate with ** complex nervous systems.”

Comment 17: “Line 222: ... More results are provided in Figures S12 and S13, comparing these animals with other group*s* expressing only ... ”

Comment 18: “Line 250-251: Figure 4c plots temporally resolved dopamine release in respon*se* to stimulations caused by red and blue micro-LEDs in *the” same mouse.”

Our response:

We thank the reviewer for pointing out these typos. All of these points have been addressed in the revised text accordingly.

References

Comment 19: *“Ref. 23 does not use fibers; the tool applies waveguides integrated on the probe shank; the authors are asked to be more specific”*

Our response:

We thank the reviewer for the clarification. We have revised the discussions associated with Ref. 23 accordingly, by changing “fiber-based emitters” to “waveguide-based emitters”.

Our modification to the manuscript:

In Line 51, we modify it to:

“... Apart from various waveguide-based emitters ...”

Comment 20: “The authors are further asked to compare their system with the state of the art (using transfer printing as well as wafer-level based bonding and laser-lift-off) in view of system dimensions, integration density (how many μ LEDs can realistically be integrated using their approach), mechanical stiffness, long-term stability”

Our response:

We agree that our system does not have the smallest dimensions, the highest integration density or the lowest mechanical stiffness, when comparing with the state of the art. In terms of system dimensions, the wireless circuit can be further miniaturized by using inductive coupling based power transfer (e.g., Shin, et al., *Neuron*, 93, 509 (2017), by Rogers Lab). In terms of integration density, the implantable probe can further integrate more LEDs (> 10) with reduced sizes (e.g., Wu, et al., *Neuron*, 88, 1136 (2015), by Yoon Lab). In terms of mechanical stiffness, the probe can be much softer by using thin polymer substrates (e.g., He, et al., *iScience*, 23, 101387 (2020), by Xie and Luan Labs).

However, the goal of this research is not to break the world record for system dimensions, integration density or mechanical stiffness. Instead, we design and fabricate a dual-color probe with stacked blue and red micro-LEDs that is suitable and works properly for bidirectionally interrogating the VTA region, and successfully activate and inhibit neural activities in the same mice, and deterministically evoke reward and aversion based behaviors, respectively. These results have not been demonstrated in previous works.

Our modification to the manuscript:

In Line 390, we add:

“...While arrays of blue InGaN micro-LEDs can be directly grown on silicon and integrated with recording electrodes⁴¹, the combination simultaneous dual-color stimulations and electrophysiological recordings in the deep brain region like VTA require more sophisticated device design and fabrication and have not been attempted here. Considering the system dimensions, wireless circuits can be further miniaturized by employing inductive coupling based power transfer strategies^{24, 26}. In terms of biocompatibility, tissue lesion can be further reduced by using thinner substrates with much lower stiffness⁴⁵. ...”

We add a new reference to indicate possible ways to reduce the probe’s stiffness:

45. He F, Lycke R, Ganji M, Xie C, Luan L. Ultraflexible neural electrodes for long-lasting intracortical recording. *iScience* 2020: 101387.

Reviewers' Comments:

Reviewer #1:

Remarks to the Author:

Authors' response to all my comments were satisfactory. I just have one minor suggestion as written below.

The authors made the following modification in the manuscript.

In Line 138:

"... Compared to a probe with laterally arranged red and blue micro-LEDs displaying highly misaligned emissions, the vertically assembled device provides a smaller footprint and generates the optimal overlapping profile for dual-color stimulation..."

I feel that "highly misaligned" is exaggerated expression because microscale LEDs are small and they can be placed side by side. To tone down, I would recommend to change "highly misaligned emissions" to "misaligned emissions with relatively small overlapping volume of light".

Overall, this is a well-written paper with high quality. I recommend acceptance of the manuscript.

Reviewer #2:

Remarks to the Author:

Overview

The manuscript presents a device for colocalized dual-color operation of "micro"-LEDs. The device is implanted in the VTA of freely moving mice injected with two opsins with low-overlapping activation spectra (ChrimsonR and stGtACR2). The opsins are chosen so that red light would increase the firing rates of the local population and blue light would decrease the firing rates of the same cells. The opsins are validated in vitro. The system operates wirelessly. Three experiments are reported for demonstrating device potential, all involving dual-color manipulations in mouse VTA: (1) real-time place preference/avoidance paradigm; (2) social interactions of multiple mice; (3) recording of dopamine release in the nucleus accumbens.

General

The technology of dual-color probes has great potential. Although this technology already exists, the wireless operation of the reported dual-color device provides an advance. Thus, the novelties in the present manuscript are: (1) adding another color at the same location; (2) wireless operation. Some of the experiments conducted in this paper could have been carried out using other technologies (e.g., without optical co-localization). This is mainly due to the large size of the micro-LEDs and the high light intensity. Nevertheless, if properly validated, the device could open an innovative path for exploring multi-animal behaviors.

Following the previous round of review, the authors have done an enormous amount of outstanding work in a relatively short time. Most issues have been dealt with satisfactorily. However, four critical issues remain. Without addressing these issues head-on, the interpretation of the results reported in Figures 3, 4, and 5 is limited, and the proposed device cannot be declared as suitable for intact subjects.

In summary, the manuscript has been improved. However, there are several outstanding issues that limit the validity and should be addressed; some require experiments (comments 1 and 2 below), whereas others may be addressed via either experiments or text (comments 3 and 4 below).

Major issues

(1) The social interaction results (the new Fig. 4) are very impressive. However, it seems that the authors mixed the input from two different comments. A request was made to "suggest and demonstrate an application that cannot be realized using extra-cranial illumination." The emphasis was on extra-cranial illumination, not on wireless control. Extra-cranial illumination literally means

that the light is emitted outside the cranium, without any waveguide into the cranium. In other words, that the photons would enter the neural tissue without a physical implant in the skull. This has been done with Jaws when initially reported (Chuong et al., 2014, Nature Neuroscience) and several times since. All of the experiments reported in the revised MS can be carried out using extracranial illumination.

Please suggest and demonstrate an application that cannot be realized using extra-cranial illumination (tethered or wireless).

(2) The new red + blue place preference experiments (Fig. S18) are interesting. Although red light and blue light are applied simultaneously, these experiments could not be carried out without connecting and disconnecting a new physical device (e.g., wirelessly controlled light source) every few seconds/minutes.

However, no-opsin controls (as in Fig. 3e) are missing. Without those, the validity of the new experiments cannot be assessed.

(3) There is something very weird in the plots of Fig. S25 (previous Fig. S18), which are the only evidence for altering neuronal activity using the proposed device in deep brain structures in the intact animal. The new Fig. S11 gives some confidence, but the issues in Fig. S25 must be resolved. There are two oddities in Fig. S25. First, when blue light is added, fluorescence decreases (compare green curve to red curve, Fig. S25b). Second, when blue light is applied alone, the fluorescence increases (step in blue curve, Fig. S25b). While the second issue may be conciliated by stating that Chrimson is also sensitive to blue light (Klapoetke et al., 2014, Nature Methods), the first cannot. On the other hand, while the first issue may be conciliated by stating that Chrimson changes conformation by the addition of blue photons, the second cannot. Thus, by Occam's razor, we are led to believe that there may be a different explanation. Differential heating in the brain during the two wavelengths may cause part of the effect: some heating can lead to depolarization, but more heating can lead to hyperpolarization. However, the lack of such effects in no-opsin controls (Fig. S25d) complicates the picture. Perhaps there is an interaction between the effect of activating Chrimson and heating? Or a circuit effect? In any case, the data in the figure cast doubt on the validity of the "VTA-NA/Chrimson/blue+red light" toolbox, and without resolving this, all experiments that use that toolbox (even without the NA) cannot be interpreted as simply as the authors suggest.

As is, we get the feeling of a problem being hidden, which is clearly not what the authors intend. There are two options: (1) understand (and ideally resolve) the source of the two issues; (2) present these issues and limitations head-on, by incorporating Fig. S25b in a main figure and dedicating text to discussing the unresolved issues.

(4) After reading the authors response to the comments, it has become clear that the "micro-LED" device is designed for regional application, and not for single-unit use. For such purposes, the size of the LEDs is satisfactory, as stated explicitly on line 104. Thus, the issue is one of terminology: the term "micro-LED" has already been coined and used in other papers, for much smaller size LEDs. Objectively, the term "micro" refers to micro-meters, whereas the LEDs in this work are 125 x 180 micrometers – which is closer to milli-meters, not micro-meters.

Thus, the term "microscale light-emitting diodes (micro-LEDs)" (used in the abstract, and throughout the text) is misleading. There are two options: (1) change the terminology, e.g., to "mini-LEDs" or "milli-LEDs"; or (2) give exact measurements of the LEDs in the Abstract.

Minor issues

(5) Grammar and typographical mistakes remain (e.g., Fig. 4e legend: "three mouse", should read "three mice").

(6) Details are still missing from multiple figures (e.g., Fig. 4e: time axis is missing).

(7) Fig. S15a - please mark the location of the lesion

Reviewer #3:

Remarks to the Author:

I thank the authors taking into account my comments and suggestions. The reply to my comments is satisfactory and I strongly support the publication of the manuscript.

Reviewer #1

Authors' response to all my comments were satisfactory. I just have one minor suggestion as written below.

The authors made the following modification in the manuscript.

In Line 138:

"... Compared to a probe with laterally arranged red and blue micro-LEDs displaying highly misaligned emissions, the vertically assembled device provides a smaller footprint and generates the optimal overlapping profile for dual-color stimulation..."

I feel that "highly misaligned" is exaggerated expression because microscale LEDs are small and they can be placed side by side. To tone down, I would recommend to change "highly misaligned emissions" to "misaligned emissions with relatively small overlapping volume of light".

Overall, this is a well-written paper with high quality. I recommend acceptance of the manuscript.

Our response:

We thank the reviewer for these positive comments and the recommendation for publication.

Our modification to the manuscript:

In Line 139, we modified the sentence to:

"... displaying misaligned emissions with a relatively small overlapping volume of light, ..."

Reviewer #2

Overview

The manuscript presents a device for colocalized dual-color operation of “micro”-LEDs. The device is implanted in the VTA of freely moving mice injected with two opsins with low-overlapping activation spectra (ChrimsonR and stGtACR2). The opsins are chosen so that red light would increase the firing rates of the local population and blue light would decrease the firing rates of the same cells. The opsins are validated in vitro. The system operates wirelessly. Three experiments are reported for demonstrating device potential, all involving dual-color manipulations in mouse VTA: (1) real-time place preference/avoidance paradigm; (2) social interactions of multiple mice; (3) recording of dopamine release in the nucleus accumbens.

General

The technology of dual-color probes has great potential. Although this technology already exists, the wireless operation of the reported dual-color device provides an advance. Thus, the novelties in the present manuscript are: (1) adding another color at the same location; (2) wireless operation. Some of the experiments conducted in this paper could have been carried out using other technologies (e.g., without optical co-localization). This is mainly due to the large size of the micro-LEDs and the high light intensity. Nevertheless, if properly validated, the device could open an innovative path for exploring multi-animal behaviors.

Following the previous round of review, the authors have done an enormous amount of outstanding work in a relatively short time. Most issues have been dealt with satisfactorily. However, four critical issues remain. Without addressing these issues head-on, the interpretation of the results reported in Figures 3, 4, and 5 is limited, and the proposed device cannot be declared as suitable for intact subjects.

In summary, the manuscript has been improved. However, there are several outstanding issues that limit the validity and should be addressed; some require experiments (comments 1 and 2 below), whereas others may be addressed via either experiments or text (comments 3 and 4 below).

Our response:

We thank the reviewer for these positive comments and valuable suggestions. The technical questions have been addressed in the following discussions as well as the revised manuscript.

Major Issues

Comment 1: “The social interaction results (the new Fig. 4) are very impressive. However, it seems that the authors mixed the input from two different comments. A request was made to “suggest and demonstrate an application that cannot be realized using extra-cranial illumination.” The emphasis was on extra-cranial illumination, not on wireless control. Extra-cranial illumination literally means that the light is emitted outside the cranium, without any waveguide into the cranium. In other words, that the photons would enter the neural tissue without a physical implant in the skull. This has been done with Jaws when initially reported (Chuong *et al.*, 2014, *Nature Neuroscience*) and several times since. All of the experiments reported in the revised MS can be carried out using extracranial illumination.

Please suggest and demonstrate an application that cannot be realized using extra-cranial illumination (tethered or wireless).”

Our response:

We apologize for the confusion we made previously, by misunderstanding the term “extra-cranial illumination”. As the reviewer pointed out, super sensitive opsins have recently been rapidly developed and enabled non-invasive, transcranial optogenetics for certain applications.

Representative examples are summarized:

Jaws:

Chuong A S, Miri M L, Busskamp V, *et al.* Noninvasive optical inhibition with a red-shifted microbial rhodopsin. *Nat. Neurosci.* 2014, **17**(8): 1123-1129

ReaChR:

Lin J Y, Knutsen P M, Muller A, *et al.* ReaChR: a red-shifted variant of channelrhodopsin enables deep transcranial optogenetic excitation. *Nat. Neurosci.* 2013, **16**(10): 1499-1508.

ChRmine:

Chen R, Gore F, Nguyen Q-A, Ramakrishnan C, Patel S, Kim SH, *et al.* Deep brain optogenetics without intracranial surgery. *Nat. Biotechnol.* 2021, **39**(2): 161-164.

SOUL:

Gong X, Mendoza-Halliday D, Ting JT, Kaiser T, Sun X, Bastos AM, *et al.* An ultra-sensitive step-function opsin for minimally invasive optogenetic stimulation in mice and macaques. *Neuron* 2020, **107**(1): 38-51.

These opsins are very sensitive to light and can be activated via extra-cranial illumination. Nevertheless, most of them are designed to have a peak response in the red range for optimal light penetration in the tissue. These opsins have highly overlapping spectra and difficult to be combined for bidirectional excitation/inhibition modulations.

Figure. Summarized response spectra from references.

Another possible candidate is SOUL, which is a step-function opsin that is turned on with blue light (473 nm) and turned off with orange light (589 nm) (Gong X, et al., *Neuron* 2020). It has not been combined with other opsins for bidirectional modulations, either.

Attenuation of light with different colors in the brain tissue, from Figure E3a, in Chen R, et al., *Nat. Biotechnol.* 2021.

Some demonstrations are performed in the very shallow brain region of mice, for example, vM1 at 1~2 mm depth for ReaChR (Lin, et al., *Nat. Neurosci.* 2013), and mFPC at 1~3 mm depth for Jaws (Chuong, et al., *Nat. Neurosci.* 2014). Due to the strong tissue attenuation, very high illumination intensity is required to penetrate into the deep brain (e.g. VTA at ~5 mm depth), even at 635 nm. For example, red light $> 200 \text{ mW/mm}^2$ is used to stimulate VTA in mice (Chen, et al., *Nat. Biotechnol.* 2021),

and blue light $> 400 \text{ mW/mm}^2$ is used to stimulate LH in mice (Gong, et al., *Neuron* 2020). In these papers, such a high power is delivered via an optical fiber cemented on skull. In other words, wireless operations have not been realized, which makes the studies on social interactions among multiple mice impossible (as we demonstrated in Fig. 4).

In order to achieve wireless operations for extra-cranial activation, larger laser or LED chips are needed to provide strong illumination ($> 200 \text{ mW/mm}^2$, $10\times$ more than our micro-LED), and heavier batteries are required to supply high power. It will be very challenging to miniaturize the system that can be head-mounted on freely behaving mice. Alternatively, one can envision a remote laser scanning system that can automatically track animal movement and enables wireless extra-cranial illumination. Such a system would also be very complicated and has not yet been reported either. After all, the bidirectional modulations of place preference (Fig. 3) as well as the social interactions (Fig. 4) reported in our paper have not been carried out using these state-of-the-art super sensitive opsins with extra-cranial illumination.

Overall, we agree that these super sensitive opsins are very remarkable developments and will have important utilities for certain applications. Nonetheless, we have to admit the fact that these techniques are not mature enough for broad optogenetic applications. Neuroscientists and engineers still rely on more conventional opsins activated via implantable waveguides or miniaturized light sources, which is definitely worth continuing efforts for future improvements. We believe our wireless dual-color LED probe will attract broad interest within the community.

Our modification to the manuscript:

In Line 415, we add:

“... Moreover, the combination of super sensitive opsins (Jaws¹⁵, ChRmine⁷¹, ReaChR⁷², SOUL¹³, etc.) could be explored for bidirectional modulations via extra-cranial illumination without implants. Since relatively large irradiance ($> 200 \text{ mW/mm}^2$) is required to penetrate into the deep tissue, corresponding light sources and control systems have to be miniaturized for wireless operation in mice. ...”

We add references:

71. Lin J Y, Knutsen P M, Muller A, et al. ReaChR: a red-shifted variant of channelrhodopsin enables deep transcranial optogenetic excitation. *Nat. Neurosci.* 2013, **16**(10): 1499-1508.

72. Chen R, Gore F, Nguyen Q-A, Ramakrishnan C, Patel S, Kim SH, et al. Deep brain optogenetics without intracranial surgery. *Nat. Biotechnol.* 2021, **39**(2): 161-164.

Comment 2: “The new red + blue place preference experiments (Fig. S18) are interesting. Although red light and blue light are applied simultaneously, these experiments could not be carried out without connecting and disconnecting a new physical device (e.g., wirelessly controlled light source) every few seconds/minutes.

However, no-opsin controls (as in Fig. 3e) are missing. Without those, the validity of the new experiments cannot be assessed.”

Our response:

We thank the reviewer for the suggestion and have performed additional experiments to test the same protocol on control mice (with EGFP + mCherry). Similar to results obtained in Fig. 3, the control group shows no obvious place preference, further validating the efficacy of dual-color modulations.

Our modification to the manuscript:

In Line 260, we modify the sentence to:

“... Additionally, the application of both red and blue stimulations in opposite chambers further enhances the preference indices of mice compared to those only experiencing single color stimulations as well as the control group (co-expressing EGFP and mCherry), ...”

We update Fig. S18, to include the results for the control group:

Figure S18

Figure S18. (a) Patterns used for optogenetic modulations, including a 10-min pretest, a 30-min red LED stimulation (20 Hz, 10-ms pulse, current 7 mA) in the left chamber and blue LED stimulation (continuous, current 5 mA) in the right chamber. (b) Representative heat maps comparing pretest and real-time preference behavior following both red (left chamber) and blue (right chamber) stimulation for mice expressing stGtACR2 + ChrimsonR (experiment group) and EGFP + mCherry (control group). (c) Preference indices measured at different times for mice under only red stimulations (red line, $n = 4$ mice), or red stimulations in the left chamber and blue stimulations in the right chamber (exp, purple line, $n = 3$ mice), and the control group with the red stimulation in the left chamber and blue light in the right chamber (ctrl, black line, $n = 4$ mice). (d) Summary of preference indices (the ratio of the time that mice spend in the left chamber to the whole recorded time) for mice under only red stimulations ($n = 4$ mice), or red stimulations in the left chamber and blue stimulations in the right chamber ($n = 3$ mice) for both experiment and control groups. Student's t test, ** $P < 0.01$, **** $P < 0.0001$.

Comment 3: “There is something very weird in the plots of Fig. S25 (previous Fig. S18), which are the only evidence for altering neuronal activity using the proposed device in deep brain structures in the intact animal. The new Fig. S11 gives some confidence, but the issues in Fig. S25 must be resolved. There are two oddities in Fig. S25. First, when blue light is added, fluorescence decreases (compare green curve to red curve, Fig. S25b). Second, when blue light is applied alone, the fluorescence increases (step in blue curve, Fig. 25b). While the second issue may be conciliated by stating that Chrimson is also sensitive to blue light (Klapoetke et al., 2014, Nature Methods), the first cannot. On the other hand, while the first issue may be conciliated by stating that Chrimson changes conformation by the addition of blue photons, the second cannot. Thus, by Occam’s razor, we are led to believe that there may be a different explanation. Differential heating in the brain during the two wavelengths may cause part of the effect: some heating can lead to depolarization, but more heating can lead to hyperpolarization. However, the lack of such effects in no-opsin controls (Fig. S25d) complicates the picture. Perhaps there is an interaction between the effect of activating Chrimson and heating? Or a circuit effect? In any case, the data in the figure cast doubt on the validity of the “VTA-NA/Chrimson/blue+red light” toolbox, and without resolving this, all experiments that use that toolbox (even without the NA) cannot be interpreted as simply as the authors suggest.

As is, we get the feeling of a problem being hidden, which is clearly not what the authors intend. There are two options: (1) understand (and ideally resolve) the source of the two issues; (2) present these issues and limitations head-on, by incorporating Fig. S25b in a main figure and dedicating text to discussing the unresolved issues.”

Our response:

We thank the reviewer for this comment. We agree that results in Fig. S25b are interesting, and admit that understanding the spectral and temporal response of ChrimsonR requires additional explanations and investigations that would be beyond the scope of our current paper.

Nevertheless, for bidirectional modulations, it is very rare to apply red+blue light simultaneously, and more commonly used strategies are alternating different colors. See our *in vivo* behavioral studies like Figs 3 and 4, as well as those reported in other papers (e.g., Kampasi, et al., Microsystems & Nanoengineering, 2018; Noked, et al., IEEE Trans Biomed Eng, 2021). Therefore, we believe that the difficulties in interpreting the unexpected results in Fig. S25b will not affect the key findings and major conclusions in our paper.

As the reviewer suggests, we move results in Fig. S25 into the main figure (Fig. 5h–5k), provide associate discussions on these phenomena and consider them as open questions for further explorations.

Our modification to the manuscript:

In Line 365, we add:

“...It is also noted that fluorescence signals experience a drop when the blue LED is operating during red stimulation for the mice only expressing ChrimsonR (Figure 5i). Such a response is unexpected considering the fact that blue light also activates ChrimsonR. One possible explanation is that when both pulsed red light and continuous blue light are imposed simultaneously, their activation effects on ChrimsonR are not additive but competitive. On the other hand, the application of both LEDs possibly induces additional heating effects, complicating the optical and thermal responses of ChrimsonR. Nevertheless, understanding spectral and temporal properties for these opsins requires further explorations in the future work. ...”

We modify Fig. 5, including the original results in Fig. S25 as now Fig. 5h–5k

Comment 4: “After reading the authors response to the comments, it has become clear that the “micro-LED” device is designed for regional application, and not for single-unit use. For such purposes, the size of the LEDs is satisfactory, as stated explicitly on line 104. Thus, the issue is one of terminology: the term “micro-LED” has already been coined and used in other papers, for much smaller size LEDs. Objectively, the term “micro” refers to micro-meters, whereas the LEDs in this work are 125 x 180 micrometers – which is closer to milli-meters, not micro-meters.

Thus, the term “microscale light-emitting diodes (micro-LEDs)” (used in the abstract, and throughout the text) is misleading. There are two options: (1) change the terminology, e.g., to “mini-LEDs” or “milli-LEDs”; or (2) give exact measurements of the LEDs in the Abstract.”

Our response:

We have accepted the reviewer’s suggestion and provided the exact dimensions for our LEDs in the abstract.

Our modification to the manuscript:

In Line 30, we add:

“... with a lateral dimension of $125 \times 180 \mu\text{m}^2$...”

Minor points

Comment 5: *“Grammar and typographical mistakes remain (e.g., Fig. 4e legend: “three mouse”, should read “three mice”).”*

Our response:

We thank the reviewer for helping edit the paper. We have proofread the manuscript and corrected these grammar and typo errors.

Comment 6: “Details are still missing from multiple figures (e.g., Fig. 4e: time axis is missing).”

Our response:

We have provided more details in several figures.

Our modification to the manuscript:

We add the time axis in Fig. 4e:

We add the scale bar in Fig. 1i:

We add scale bars in Figs. S3a and S3b:

Comment 7: “Fig. S15a - please mark the location of the lesion.”

Our response:

We have updated Fig. S15a and marked the lesion. It should be noted that the preparation of the living tissue slice is different from that for the fixed immune stained slices in Fig. S13. The living tissue slices are more fragile, tend to deform and break into pieces around the lesion part. Therefore, it is very difficult to locate the “lesion area” with the exact size of the probe, as in Fig. S13.

Our modification to the manuscript:

We updated Fig. S15a with a marked border of the lesion area:

Reviewer #3

I thank the authors taking into account my comments and suggestions. The reply to my comments is satisfactory and I strongly support the publication of the manuscript.

Our response:

We thank the reviewer for these positive comments and the recommendation for publication.

Reviewers' Comments:

Reviewer #2:

Remarks to the Author:

The authors have addressed all of my concerns satisfactorily. I would like to congratulate the authors on this work and recommend publication of the manuscript.

Reviewer #2

The authors have addressed all of my concerns satisfactorily. I would like to congratulate the authors on this work and recommend publication of the manuscript.

Our response:

We thank the reviewer for the positive comment and the recommendation for publication.